# A Fourier perspective on the learning dynamics of neural networks: from sample complexities to mechanistic insights

**Fabiola Ricci** [1]   **Claudia Merger** [1]   **Sebastian Goldt** [1]

## Abstract

Neural networks trained with gradient-based methods exhibit a strong simplicity bias: they learn simpler statistical features of their data before moving to more complex features. Previous analyses of this phenomenon have largely focused on settings with (quasi-)isotropic inputs. In this work, we study the simplicity bias from a Fourier perspective, which allows us to include two key features of natural images in the analysis: approximate translation-invariance and power-law spectra. We first show experimentally that simple neural networks trained on image classification tasks first rely on amplitude information – related to pair-wise correlations between pixels – before exploiting phase information, which encodes edges and higher-order correlations. In view of this, we introduce a synthetic data model for translation-invariant inputs that allows precise control over amplitudes and phases while remaining tractable. We rigorously establish that for isotropic and high-dimensional inputs, classification based on phase information alone is a genuinely hard task: online stochastic gradient descent (SGD) cannot distinguish the structured inputs from noise within $n \ll N^3$ steps, but needs at least $n \gg N^3 \log^2 N$ steps. In contrast, we show both experimentally and theoretically that power-law spectra can dramatically accelerate the speed of learning phase information, even if the spectra do not help with classification. Simulations with two-layer networks trained on textures and with deep convolutional networks on ImageNet and CIFAR100 confirm this non-trivial interaction between amplitudes and phases, providing mechanistic insights into how deep neural networks can learn natural image distributions efficiently.

[1]International School of Advanced Studies, Trieste, Italy. Correspondence to: Fabiola Ricci <fricci@sissa.it>, Claudia Merger <cmerger@sissa.it>, Sebastian Goldt <sgoldt@sissa.it>.

*Proceedings of the 43rd International Conference on Machine Learning*, Seoul, South Korea. PMLR 306, 2026. Copyright 2026 by the author(s).

## 1. Introduction

A universal theme in the learning dynamics of neural networks trained with gradient-based methods are *simplicity biases*: for example, neural networks reliably learn simpler statistical features of their data before moving to more complex features, both in image classification (Kalimeris et al., 2019; Ingrosso & Goldt, 2022; Refinetti et al., 2023) and in next-token prediction (Rende et al., 2024; Belrose et al., 2024; Favero et al., 2025; Garnier-Brun et al., 2025). A rigorous characterisation of these biases is essential to understand how neural networks can learn "natural" data distributions like text and images efficiently, and which features they learn from their data. From a theoretical point of view, simplicity biases were first analysed in simple models of neural networks learning a target function over Gaussian inputs (Saad & Solla, 1995; Saxe et al., 2014; 2019; Abbe et al., 2021; 2023; Dandi et al., 2024a; Berthier et al., 2025), in auto-encoders (Kögler et al., 2024), and in the kernel regime (Farnia et al., 2018; Rahaman et al., 2019). More recently, a *distributional simplicity bias*, whereby neural networks first rely on pair-wise input statistics before exploiting higher-order input correlations was analysed in non-Gaussian models of inputs (Ingrosso & Goldt, 2022; Merger et al., 2023; Bardone & Goldt, 2024; Ricci et al., 2025b). However, the input models in these works do not account for several important properties of natural images, such as their (approximate) translation-invariance or the power-law decay of their power spectra (van der Schaaf & van Hateren, 1996; Hyvärinen et al., 2009).

Here, we introduce a different perspective on modelling inputs for the theoretical analysis of neural networks by taking a Fourier point of view. Natural images are approximately translation-invariant, so frequency-space representations are both a simple and a natural choice for this data. The Fourier representation is particularly useful for investigating distributional simplicity biases because it provides a clean separation of pair-wise and higher-order statistics: for translation-invariant inputs like image patches, the amplitudes of the Fourier coefficients determine pair-wise correlations, while higher-order correlations are encoded only by the phases (Hyvärinen et al., 2009); see Section 2 for precise definitions.

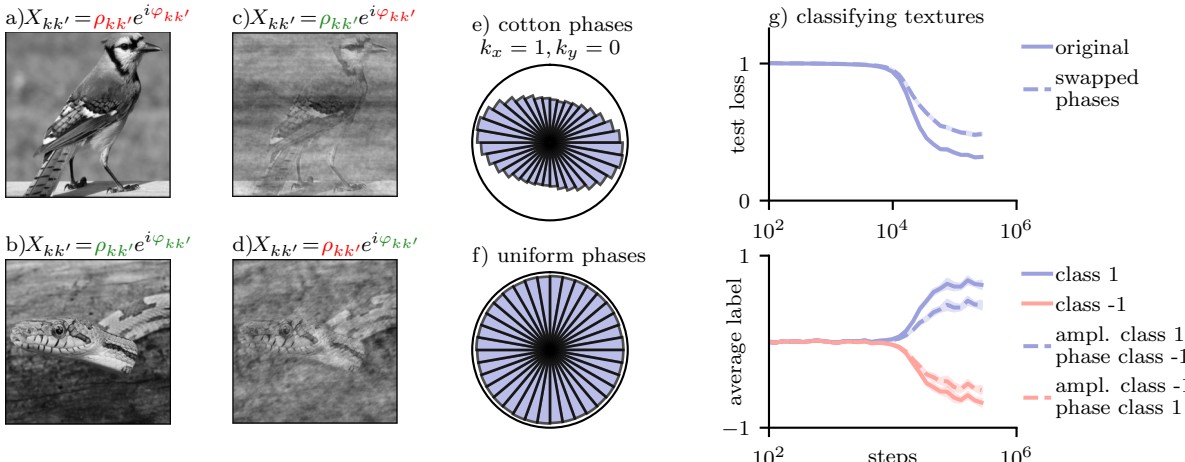

*Figure 1.* **Learning phase vs amplitude information. a-b)** Pictures of a bird and a snake from ImageNet. **c-d)** Fourier image reconstruction with phases $\varphi_{kk'}$ from the "bird" and amplitudes $\rho_{kk'}$ from the "snake" and vice versa. **e)** Phases of images from the "cotton" class of the ALOT texture dataset for patches of size $16 \times 16$ along the first Fourier mode in the $x$-direction. **f)** Uniform phase distribution. **g)** Performance of a classifier trained to distinguish between two texture classes: "cotton" vs. "lace" on patches of size $16 \times 16$ of the ALOT dataset. **Top:** test loss over training steps, on the original data (—) and on data where phase information has been swapped between classes (- ·). **Bottom:** average label assigned to samples from each class (—, —) and samples with mixed amplitude/phase information (- ·, - ·). All curves are averages over 8 random initializations of the classifier.

The different roles of amplitudes and phases can also be seen directly: in Figure 1 a-b), we show images of a bird and a snake, taken from ImageNet. Next to them, we show two images that we obtained by combining the amplitudes of one image with the amplitudes of the other in Fourier space, and then transforming back into pixel space. In both cases, the resulting images look like noisy versions of the image whose phases we reused. This is a classic illustration of the importance of phase information for human perception, which has long been established in cognitive science (Oppenheim & Lim, 1981; Piotrowski & Campbell, 1982).

In this work, we analyse the learning dynamics of simple neural networks trained on image classification tasks from a Fourier perspective. We ask: how hard is it to learn from amplitudes and from phases, respectively? Are they learnt sequentially? And what is the impact of other salient features of images, like the power-law decay of their spectrum (Hyvärinen et al., 2009), on their learning dynamics?

Our **first main contribution** is to demonstrate that neural networks sequentially learn to exploit first amplitude, then phase information. To this end, we train a fully connected two-layer classifier to distinguish between image patches containing textures "cotton" and "lace" from the ALOT dataset (Burghouts & Geusebroek, 2009). These textures are approximately translation-invariant, while the phases are non – uniform (Figure 1 e)). We evaluate the performance of the classifiers on a held-out test set and on the same test set with the image phases swapped between classes, analogous to the bird and snake example. We compare the test performances in Figure 1 g); details on the training in Section B. We find that initially, in the first phase of learn-

ing after a search phase, the network achieves comparable performance on both test sets, indicating that it is using only the amplitude information to distinguish the two classes. As training progresses, the performance on the phase-swapped test set deteriorates in comparison to the original data, meaning that the network has learnt to exploit phase information. This demonstrates sequential learning of first amplitude, then phase information in neural networks. In what follows, we build a theoretical model that elucidates that amplitude and phase information correspond to lower and higher-order statistics of a dataset, respectively, and quantifies the sample complexity at which this information becomes available to a neural network. Our **further main contributions** are as follows:

- We introduce a synthetic model for non-Gaussian, translation-invariant inputs which allows us to precisely control pair-wise and higher-order statistics via amplitude and phase manipulations (Section 2).

- We prove that when the inputs have isotropic covariance, weakly recovering information carried exclusively by the phases requires a sample complexity of the order of $n \gg N^3$ for online SGD, making this a hard task in high dimensions (Section 3).

- For non-isotropic inputs, we show experimentally that a power-law spectrum speeds up the recovery of phase information in shallow and deep convolutional neural networks trained on a variety of datasets (Section 4).

- We show theoretically that neural networks can achieve quasi-linear sample complexity for weak recovery of phase information when inputs have a power-law spectrum (Section 5).

From a technical point of view, much of our results rely on an analysis of the dynamics of SGD by identifying suitable invariants of the loss, most notably the *information exponent* introduced and developed by Ben Arous et al. (2021; 2022) for isotropic inputs with a single non-trivial direction. A key innovation in the present work is an analysis of the population loss when the non-Gaussian inputs have a general non-isotropic covariance matrix, which is essential to establish our results on the speed-up of learning due to the distribution of amplitudes in real images.

**Further related work** A series of works study the **impact of power-law spectra on SGD dynamics**. Paquette et al. (2024) studied scaling laws in a power-law random feature model. Braun et al. (2025) studied phase retrieval on Gaussian inputs with power-law covariances. Ben Arous et al. (2025a); Ren et al. (2026); Defilippis et al. (2026) studied scaling laws of the loss when training on isotropic Gaussian data with labels given by a two-layer teacher with power-law second-layer weights. The key novelty of our work is to include the power-law in the spectrum of *non-Gaussian* inputs, which do not reduce to a Gaussian additive model, see Lemma D.8, and to look at SGD in the feature learning regime. We discuss further related work in Section A.

## 2. The Fourier data model

Our goal is to establish rigorously how phases are learnt by a simple neural network, and to clarify the role of the amplitudes in shaping the learning dynamics. To that end, we consider the standard setting of a binary discrimination task with high-dimensional inputs $\mathcal{D} = (x^\mu, y^\mu)^n_\mu \subseteq \mathbb{R}^N \times \mathbb{R}$ on which we train a single neuron or student $\sigma$ with weights $w \in \mathbb{R}^N$ using the correlation loss

$$L(w; x^\mu, y^\mu) = 1 - y^\mu \sigma(w \cdot x^\mu), \quad (1)$$

see Ben Arous et al. (2021); Damian et al. (2023); Dandi et al. (2024b); Bardone & Goldt (2024) for examples of similar setups.

### 2.1. The standard path to structured data

A natural approach to characterise the difficulty of learning certain data structures is to let the student distinguish noise from structured inputs in the form of hypothesis test. One can assume that inputs with label $y^\mu = -1$ are pure noise sampled from a multivariate normal distribution $\mathbb{P}_0$. Meanwhile, inputs with label $y^\mu = +1$ are drawn from a distribution $\mathbb{P}$ that has a single preferred, or planted, direction. The inputs are typically built by sampling $z^\mu$ from $\mathbb{P}_0$ and adding a "spike" $u \in \mathbb{R}^N$, e.g.

$$x^\mu = \beta g^\mu u + z^\mu, \quad (2)$$

with signal-to-noise ratio $\beta > 0$ and a scalar latent variable $g^\mu$. By discriminating these two input classes, we

essentially perform noise contrastive estimation (Gutmann & Hyvärinen, 2010) and the student ideally recovers the planted direction $u$.

### 2.2. The Fourier data model

Here, we propose a new data model that captures some salient structural properties of images and enables us to surgically manipulate amplitudes and phases.

Our **baseline distribution** is again the multivariate Gaussian, except that we impose a non-trivial, translation-invariant covariance matrix. We set $\mathbb{P}_0 = \mathcal{N}(0, \Sigma)$, where the real covariance matrix $\Sigma \in \mathbb{R}^{N \times N}$ is stationary with periodic boundaries conditions, i.e. $\Sigma_{ij} = c_{(i-j) \bmod N}$, for some real coefficients $(c_0, \cdots, c_{N-1})$. Equivalently, $\Sigma$ is a circulant matrix (see Lemma C.4). In what follows, we denote the inputs sampled from $\mathbb{P}_0$ as *translation-invariant*.

Inputs under the **planted distribution** $\mathbb{P}$ are obtained by a suitable modification, in Fourier space, of inputs drawn from $\mathbb{P}_0$. We could see $\mathbb{P}$ as a non-Gaussian counterpart of $\mathbb{P}_0$, since by construction they share the same low-order statistics – mean and covariance matrix – but the signal distinguishing the two classes of inputs is encoded in the higher-order cumulants of $\mathbb{P}$. Hence, the inputs sampled from $\mathbb{P}$ are also translation-invariant. We stress that $\mathbb{P}_0$ and $\mathbb{P}$ are indistinguishable based on their cumulants up to second-order.

We build $\mathbb{P}$ by first sampling $(z^\mu)^n_{\mu=0} \subseteq \mathbb{R}^N$ from the baseline distribution $\mathbb{P}_0$ and considering their Discrete Fourier Transform (DFT) denoted by $Z^\mu = \text{DFT}(z^\mu)$. The Fourier coefficients can be decomposed into amplitudes $\rho^\mu_k$ and phases $\varphi^\mu_k$, i.e. $Z^\mu_k = \rho^\mu_k e^{i\varphi^\mu_k} \in \mathbb{C}$ for the frequencies $k = 0, \ldots, N - 1$. In particular, the phases are independent and uniform in $[-\pi, \pi)$ (cf. Lemma C.11). We can now change the distribution of either the amplitudes or the phases. Amplitude manipulations change both the covariance matrix and higher-order correlations of the inputs, phase manipulations do not. Indeed, as long as the data remain translation-invariant, the phases control their higher-order correlations (HOCs) leaving the covariance unchanged (cf. Lemma C.7), so we focus on that. We are going to alter the distribution of the phases as follows: for any $\varepsilon > 0$, we choose a frequency $k_0 \neq \{0, N/2\}$ define the new inputs $X^\mu \in \mathbb{C}^N$ in Fourier space as

$$X^\mu = \begin{pmatrix} \rho^\mu_0 \\ \vdots \\ \rho^\mu_{N-1} \end{pmatrix} \exp \begin{pmatrix} i\,\varphi^\mu_0 \\ \vdots \\ i\,\psi^\mu_{k_0} \\ \vdots \\ i\,\varphi^\mu_{N-1} \end{pmatrix}, \psi^\mu_{k_0} = \varphi^\mu_{k_0} + \varepsilon \underbrace{f(\varphi^\mu_{k_0})}_{\text{HOCs}} + U^\mu,$$

$$(3)$$

where $f = f(\varphi^\mu_{k_0})$ is an arbitrary function of the phase

$\varphi_{k_0}^\mu$ and $U^\mu$ is sampled from a discrete random variable, independent from $\varphi_k^\mu$, which takes values uniformly among the angles $\{0, \pi/2, \pi, 3\pi/2\}$ associated to the fourth roots of unity. To ensure that the new inputs are still real in pixel space, we ask that $X_{N-k_0}^\mu = \overline{X}_{k_0}^\mu$. We define the planted distribution $\mathbb{P}$ as the one of the Inverse Discrete Fourier Transform (IDFT) of $X^\mu$, i.e. $x^\mu = \text{IDFT}(X^\mu)$.

We now discuss the motivations behind the choices of $f(\varphi_{k_0}^\mu)$ and $U^\mu$. First of all, the presence of the phase-dependent modification leads to a non-Gaussian distribution of inputs by introducing non trivial higher-order correlations in pixel space. No such effect occurs when $f$ is independent of the phases, meaning that a simple "spike" would not serve our purpose (cf. Lemma D.1). For concreteness, we perform our theoretical analysis with $f(\varphi_{k_0}^\mu) = \sin(\varphi_{k_0}^\mu)$. On the other hand, the presence of $U^\mu$ enforces the covariance matrix of $X^\mu$ to stay diagonal, so that the inputs are still translation-invariant after the alteration of the phases. In absence of the corrector $U^\mu$, an alteration of the phases in general breaks translation-invariance of the inputs, as proved in the next lemma.

**Lemma 2.1.** *If $X^\mu$ is sampled from the Fourier data model* (3) *with $\psi_{k_0}^\mu = \varphi_{k_0}^\mu + \varepsilon f(\varphi_{k_0}^\mu)$, then $x^\mu = \text{IDFT}(X^\mu)$ is not translation-invariant.*

*Proof.* For a generic function of the phases $f$, the entry $(k_0, N - k_0)$ of the covariance matrix of $X^\mu$ is given by

$$\mathbb{E}[X_{k_0}^\mu \overline{X}_{N-k_0}^\mu] = \mathbb{E}[Z_{k_0}^{\mu\,2}]\,\mathbb{E}[e^{2i\varepsilon f(\varphi_{k_0}^\mu)}] \neq 0,$$

which contradicts $x^\mu$ being translation-invariant (see Lemma C.7 for a reminder that the covariance matrix of translation-invariant inputs is diagonal in Fourier space). □

Therefore, one can actually prove (cf. Lemma D.2) that the modified inputs $x^\mu$ are still translation-invariant and have the same covariance matrix of the original inputs drawn from $\mathbb{P}_0$. However, the new data points $x^\mu$ are non-Gaussian and have a "signal" in their fourth-order statistics (cf. Lemma D.7). We can note that, by construction of $U^\mu$, we get a non-vanishing factor $\mathbb{E}[e^{i\,4U^\mu}] = 1$, which is the reason why the new inputs $x^\mu$ are allowed to exhibit non-trivial fourth-order cumulants.

We finally note that manipulating the phases means that, in pixel space, there is a two-dimensional subspace of $\mathbb{R}^N$ along which the projections of the inputs drawn from $\mathbb{P}$ have non-Gaussian statistics (cf. Lemma D.7) and hence differ from the projections of inputs drawn from $\mathbb{P}_0$. This subspace is spanned by the pair of DFT basis vectors $(u^k)_k$ and sine DFT basis vectors $(v^k)_k$ corresponding to the modified mode $k_0$, where, for the frequencies $k = 1, \ldots, \lfloor \frac{N-1}{2} \rfloor$, we have defined

$$u_n^k = \sqrt{\frac{2}{N}} \cos\left(\frac{2\pi kn}{N}\right), \quad v_n^k = \sqrt{\frac{2}{N}} \sin\left(\frac{2\pi kn}{N}\right), \quad (4)$$

for any $n = 0, \ldots, N-1$ (cf. Figure 7 for a visualisation). The "DFT basis" $(u^k, v^k)_k$ diagonalises the covariance matrix of any translation-invariant input, see Lemma C.6. In what follows, we will drop the explicit dependence on the frequency $k_0$, write $u$ and $v$ - instead of $u^{k_0}$ and $v^{k_0}$ - and call them "**DFT phase vectors**".

## 3. Sample complexity to classify isotropic data

In this section we rigorously quantify the sample complexities of online SGD on the correlation loss (1), in the case of isotropic inputs. Recall that, by construction of the Fourier data model (3), to successfully distinguish the two classes of inputs, SGD has to learn the subspace of $\mathbb{R}^N$ spanned by the DFT phase vectors $u$ and $v$, along which the projections of the inputs drawn from $\mathbb{P}_0$ and $\mathbb{P}$ differ in distribution. Additionally, in this section we assume that the inputs of both the classes are isotropic, so with identity covariance. We prove that the choice of having whitened inputs, together with the fact that the DFT phase vectors are encoded in the higher-order cumulants of the data (cf. Lemma D.7), leads to a genuinely hard computational task in high dimensions.

Online SGD is implemented by sampling a new data point at each step from $\mathcal{D} = (x^\mu, y^\mu)_{\mu=1}^n$. For a suitable learning rate $\delta_N > 0$, each iteration of the spherical online SGD, for $w_0 \sim \text{Unif}(\mathbb{S}^{N-1})$, is defined as

$$\begin{cases} \widetilde{w}_t = w_{t-1} + \delta_N \left. \nabla_{\text{sph}} L(w; x^t, y^t) \right|_{w=w_{t-1}} & t \geq 1, \\ w_t = \widetilde{w}_t / \|\widetilde{w}_t\|, \end{cases} \quad (5)$$

where the spherical gradient is given by $\nabla_{\text{sph}} L(w, \cdot) = (\mathbb{1} - ww^\top)\nabla_w L(w, \cdot)$. To tackle this classification problem, we ask: how fast does SGD recover a signal encoded in the phase of high-dimensional translation-invariant inputs? To answer, a key object to look at is the *population correlation loss* defined as

$$L(w) = \mathbb{E}[L(w; x^\mu, y^\mu)], \quad (6)$$

where the expectation is taken over the data distribution. A fundamental invariant is the *information exponent*, which is the order $k^*$ of the first term having a non-zero coefficient in a suitable expansion of the loss (6) (cf. Definition C.16). It was introduced by Ben Arous et al. (2021) and governs the sample complexity of online SGD in single-index models. They prove that for $k^* \geq 3$, no recovery of the signal is possible within $n \ll N^{k^*-1}$ samples, and a partial alignment occurs when $n \gg N^{k^*-1} \log^2 N$. Conversely, the signal is recovered at linear sample complexity - up to logarithmic factors - when $k^* = 2$. Other useful invariants are the *leap index* of Abbe et al. (2023); Dandi et al. (2024a), and the *generative exponent* of Damian et al. (2024). Using the approach of Ben Arous et al. (2021), we prove in Theorem 3.1 that SGD requires a cubic sample

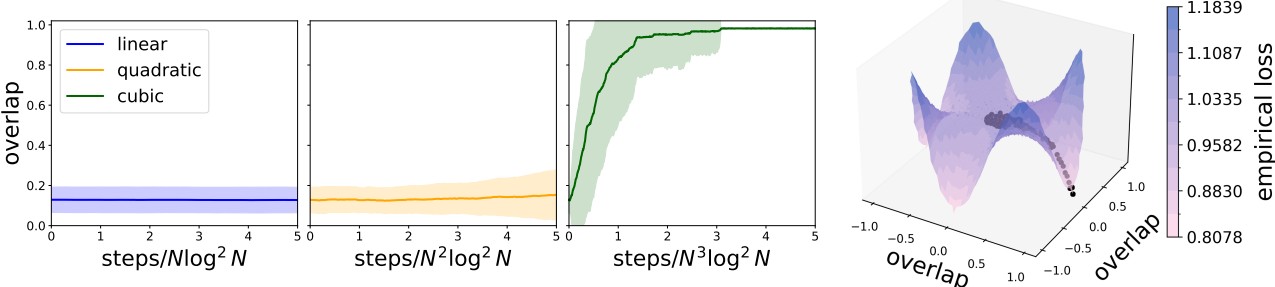

*Figure 2.* **Performance of SGD in classifying isotropic inputs on the Fourier data model. (Left)** We plot the norm of the projection of the perceptron weight on the "non-Gaussian" subspace during online SGD on the correlation loss (1) with isotropic inputs drawn from the Fourier data model (3). SGD does not weakly recover the signal at linear (—) or quadratic (—) sample complexity, whereas it converges to the subspace spanned by the DFT phase vectors in the cubic (—) regime. **(Right)** Empirical loss landscape as a function of the overlaps between the weight vector and the DFT phase vectors, i.e. $w \cdot u$ and $w \cdot v$. Additionally, we display one run of SGD (•) converging to a global minimum of the loss only after a long search phase around the origin, as predicted by Theorem 3.1.

complexity to align with the subspace spanned by the DFT phase vectors. This hardness in learning is due to the fact that the phase modification is encoded only in fourth-order cumulants (cf. Lemma D.7), leading to a high information exponent $k^* = 4$. Note that the same scalings are needed to solve the tensor PCA problem (Richard & Montanari, 2014) for a fourth-order tensor and to perform Independent Component Analysis (ICA) (Ricci et al., 2025a;b).

Assume that the activation function $\sigma : \mathbb{R} \rightarrow \mathbb{R}$ is continuously differentiable and even. The parity is a technical assumption, which does not change qualitatively the theoretical analysis, but simplifies parts of it. Theorem 3.1 is proved in Appendix E.

**Theorem 3.1.** *Sample* $(x^\mu)_{\mu=1}^n \subseteq \mathbb{R}^N$ *from the Fourier data model* (3) *with* $\Sigma = \mathbb{1}$. *Apply online SGD* (5) *to the correlation loss* (1) *and define the overlaps* $\alpha_{u,n} = w_n \cdot u$ *and* $\alpha_{v,n} = w_n \cdot v$, *where* $(u, v)$ *are the DFT phase vectors. Then, if* $1/(N^2 \log^2 N) \ll \delta_N \ll 1/(N \log N)$ *and* $\boldsymbol{n} \gg \boldsymbol{N^3 \log^2 N}$, *there is* **weak recovery** *for* $(u, v)$ *i.e., for some* $\eta > 0$,

$$\lim_{N \to \infty} P(|\alpha_{u,n}| \geq \eta) = 1 \quad and \quad \lim_{N \to \infty} P(|\alpha_{v,n}| \geq \eta) = 1.$$

*Conversely, when* $\delta_N \ll 1/(N \log N)$ *and* $\boldsymbol{n} \ll \boldsymbol{N^3}$, *there is* **no recovery** *for* $(u, v)$ *i.e., in probability,*

$$|\alpha_{u,n}|, |\alpha_{v,n}| \xrightarrow[N \to +\infty]{} 0.$$

*Sketch of the proof.* The correlation population loss (6) is essentially the sum of $\mathbb{E}_{\mathbb{P}_0}[\sigma(w \cdot z)]$ and $\mathbb{E}_{\mathbb{P}}[\sigma(w \cdot x)]$. Since $\|w\| = 1$, the former term is constant and equals the zeroth-order Hermite coefficient $c_0^\sigma$ of the activation function (cf. Definition C.15). The latter term can be expanded by using the *likelihood ratio* $\ell(s) = d\mathbb{P}/d\mathbb{P}_0(s)$ of the planted distribution $\mathbb{P}$ with respect to the baseline distribution $\mathbb{P}_0$ (see Definition C.17), which depends only on the projections of the inputs along the DFT phase vectors. Hence, we get that

$$\mathbb{E}_{\mathbb{P}}[\sigma(w \cdot x)] = \mathbb{E}_{\mathbb{P}_0}[\sigma(w \cdot x)\ell(v \cdot x, u \cdot x)]. \quad (7)$$

We now expand the activation function $\sigma$ and the likelihood ratio $\ell$ in Hermite polynomials. Since $\mathbb{P}$ and $\mathbb{P}_0$ share the same first and second-order statistics, the Hermite coefficients $c_{ij}^\ell$ of the likelihood ratio are such that $c_{ij}^\ell = 0$ if $i + j \leq 2$, with the only exception of $c_{00}^\ell = 1$ which, multiplied by $c_0^\sigma$, cancels with $\mathbb{E}_{\mathbb{P}_0}[\sigma(w \cdot z)]$. Since any odd Hermite coefficient of $\sigma$ vanishes, we get, for $c_1, c_2 \in \mathbb{R}$,

$$L(w) = 1 - c_1(\alpha_v^4 + \alpha_u^4) + c_2 \alpha_v^2 \alpha_u^2 + \text{h.o.t.}. \quad (8)$$

Then, the information exponent is $k^* = 4$. $\qquad\square$

## 4. Power-law Fourier amplitudes accelerate learning from the phases

We have established that learning phase information is computationally hard in high dimensions, requiring at least a cubic sample complexity of SGD. However, in neural networks trained on real data, no such huge gap appears to exist: in our experiments with simple networks in Figure 1 g) we find no large gap between the point where the network's performance initially improves and where it starts to exploit phase information. So how neural networks access phase information quickly on real images, when it is hard to learn from phase information alone?

A key difference between Theorem 3.1 and real images is the distribution of Fourier amplitudes of the data. For Theorem 3.1, we assumed an identity covariance matrix, which corresponds to flat amplitudes. However, images typically have a covariance whose power-spectrum is not flat, but instead exhibits a power-law decay, with amplitudes decaying roughly as $1/f$, where $f$ is the frequency of the mode. We illustrate this for the "cotton" class of the ALOT texture dataset in Figure 3 a) and we see that the diagonal of the Fourier covariance follows an approximate power-law behavior, increasing in magnitude with the size of the image patches. Those diagonal entries are the second moments of the amplitudes or, equivalently, the eigenvectors of the covariance in pixel space (cf. Remark D.3).

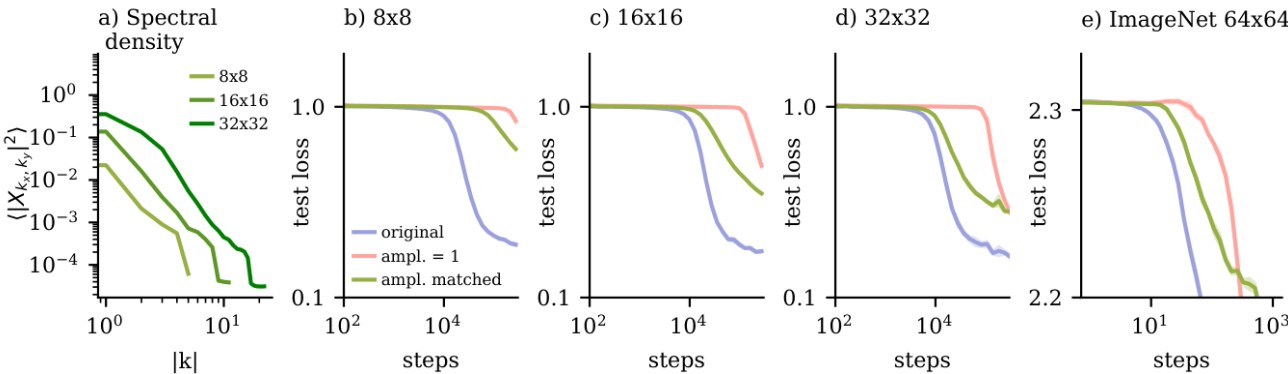

*Figure 3.* **Shared principal subspace speeds up learning. a)** Average squared Fourier amplitudes of image patches of "cotton" class from ALOT texture dataset, averaged over wave vectors of equal length $|k| = \sqrt{k_x^2 + k_y^2}$, for patches of increasing size. **b-d)** Test losses of classifiers trained on distinguishing "cotton" vs. "lace" on original data (—), data where all Fourier amplitudes of both classes have been set to one (—) and data where the amplitudes of the second class are replaced by the amplitudes of the first class (—). Test loss curves are averages over 8 random initializations of the classifiers. **e)** The same experiment as b-d, but using a Resnet18 on a super-classed version of ImageNet with ten classes. Cf. Figure 6 in the appendix to visualise the same behaviour on classes from CIFAR-100.

To investigate the effect of the power-law amplitudes on the learning dynamics, we train two-layer fully connected networks on the same textures (cf. Figure 3 b-d)), but modify their *amplitudes*. First, we show the test loss of classifiers trained on the original images without any modification (blue curves (—)). We then train classifiers on the same images after setting their amplitudes to one (pink curves (—)). This results in a considerable slow-down in the onset of learning, as the networks can now only rely on phase information, illustrating the idea of Theorem 3.1 on real data. Finally, we construct a third dataset, where we keep one class of the images unchanged, but replace the amplitudes of the second class with the amplitudes of the first one. At the level of the amplitude statistics, the two classes are hence exactly identical: they differ in their phases. However, in contrast to before, their Fourier spectra are no longer flat, but follow the curves in Figure 3 a). We show the corresponding test loss curves in green (—) in Figure 3 b-d). Even though the network can only distinguish the two classes based on the phase information, we observe a significant speed-up with respect to the case of flat Fourier spectra. Hence, the mere amplification of specific directions belonging to the principal subspace leads to a considerable speed-up in how fast the network can detect and exploit phase information.

This effect is not limited to fully-connected networks: Figure 3 e) shows the same experiment on a Resnet18 trained on a super-classed version of ImageNet with ten classes (see also Figure 6 for CIFAR-100). Even though convolutions are designed to be responsive to edges and other features encoded in the phases, we see again a slow-down of the onset of learning when amplitudes are flattened, while setting the amplitudes of all the training images to the one of one class speeds up learning. Note that the images are normalised; full details on training in Section B. In conclusion, the speed-up emerges consistently across various datasets (tex-

tures, ImageNet and CIFAR-100), architectures (Resnet18 and two-layer classifiers), optimisers (mini-Batch SGD and Adam) and losses (mean-squared and cross-entropy).

In what follows, we explain theoretically how the presence of a leading principal subspace can speed up phase learning, even if it is shared among classes and does therefore not provide additional information for discriminating them.

## 5. Theoretical analysis for non-isotropic inputs

In Section 5, we extend our theory to account for the effect of power-law spectra found experimentally. We consider inputs sampled from the Fourier data model (3) with a translation-invariant, non-isotropic covariance. We first expand the population loss, and analyse how the spectrum interacts with the phase to yield an effective signal-to-noise ratio (Section 5.1). We then discuss how power-law amplitudes can speed up learning from the phase (Section 5.2).

### 5.1. Expansion of the population loss for non-isotropic inputs

Sample inputs from the Fourier data model (3), with a circulant covariance. The key difficulty in analysing the population loss (6) here is due to the interaction between the non-Gaussianity and the anisotropy of the inputs. As in the isotropic case, the population loss can be written as the expectation over the baseline distribution $\mathbb{P}_0$ of the product of the activation $\sigma$ and the likelihood ratio $\ell$. Due to anisotropy, after expanding $\sigma$ and $\ell$ in Hermite polynomials, one first needs to rescale (23) and resum (24) our formulas before applying the orthogonality properties (Lemma C.14); otherwise, one cannot write the expectation (so the loss) in terms of the relevant overlaps. Note that we compute the Hermite coefficients of a rescaled likelihood ratio (Equation (26)).

While previous work has considered inputs with a rank-one perturbation of the identity for the covariance (Mousavi-Hosseini et al., 2023; Bardone & Goldt, 2024), a full-rank anisotropic covariance has not yet been treated in this framework, to the best of our knowledge. Technically, $\mathbb{P}_0$ is not the normal distribution as in (Bardone & Goldt, 2024), since an extensive number $O(N)$ of eigenvectors cannot be explicitly treated in the likelihood ratio.

**Proposition 5.1.** *Consider $w \in \mathbb{R}^N$, $\alpha_u = w \cdot u$ and $\alpha_v = w \cdot v$, where $(u, v)$ are the DFT phase vectors. Let $\lambda_{k_0}$ be the $k_0$-th eigenvalue of a circulant matrix $\Sigma$ and sample the inputs $x^\mu$ from the Fourier data model (3) with covariance $\Sigma$. Then, the population loss (6) is such that*

$$L(w) = 1 - \lambda_{k_0}^2 c_4^\sigma \left( c_{04} \alpha_u^4 + c_{22} \alpha_u^2 \alpha_v^2 + c_{04} \alpha_v^4 \right) \quad (9)$$
$$+ \lambda_{k_0}^2 c_6^\sigma \frac{(\sigma_\Sigma^2 - 1)}{2} \left( c_{04} \alpha_u^4 + c_{22} \alpha_u^2 \alpha_v^2 + c_{04} \alpha_v^4 \right)$$
$$+ \text{h.o.t.},$$

*with $\sigma_\Sigma = \sqrt{w^\top \Sigma w}$. Here, $c_{04}, c_{22}$ and $c_{04}$ depend on the likelihood ratio between $\mathbb{P}$ and $\mathbb{P}_0$, $c_4^\sigma$ and $c_6^\sigma$ are the 4-th and 6-th Hermite coefficients of the activation $\sigma$.*

We prove Proposition 5.1 in Section F. By construction, $\lambda_{k_0}$ is an eigenvalue of the DFT phase eigenvectors $u$ and $v$ of the covariance matrix. Note that $\lambda_{k_0}$ is task-relevant since it corresponds to the frequency of the modified phase and plays the role of an effective signal-to-noise ratio induced by the structure of the inputs. The eigenvalues $(\lambda_m)_m^{N-1}$ of the DFT eigenvectors $(u^m, v^m)_m^{N-1}$ appear in the quadratic form $\sigma_\Sigma$ induced by the covariance through the overlaps $\alpha_{u_m} = w \cdot u^m$ and $\alpha_{v_m} = w \cdot v^m$, i.e.

$$\sigma_\Sigma^2 - 1 = (\lambda_{k_0} - 1)(\alpha_u^2 + \alpha_v^2) + \sum_{m=1}^{N-1} (\lambda_m - 1)(\alpha_{u_m}^2 + \alpha_{v_m}^2). \quad (10)$$

### 5.1.1. LOSS SCALINGS FOR A LOW-RANK EIGENSTRUCTURE OF THE INPUTS

Even if Proposition 5.1 allows for any scaling of $\lambda_m$, we focus on the case where some top eigenvalues are much larger than the bulk ones, as occurs for power-law-decaying spectra, i.e. $\lambda_m \propto m^{-\alpha}$, for some $\alpha > 0$. Indeed, it is known that, in covariances with power-law spectra, different measures of dimensionality like the *effective dimension* or the *Inverse Participation Ratio* (IPR) of the eigenvalues concentrate on an $O(1)$ quantity for power-law decays with exponent $\alpha > 1$ (Wortsman & Loureiro, 2025; Bartlett et al., 2020; Cheng & Montanari, 2024). Note that this is the typical regime of natural images (van der Schaaf & van Hateren, 1996; Field, 1987; Hyvärinen et al., 2009). Motivated by this, we consider an effective low-rank eigenstructure that simplifies the analysis and maintains the correct effective

dimension. Precisely, we model a finite set $O(1)$ of leading eigenvalues as extensive, i.e. $O(N)$, while the tail of the spectrum stays $O(1)$. We contrast this realistic setting with the one where also the leading eigenvalues remain $O(1)$, as for flat amplitudes, by gaining a key insight into the impact of power-law amplitudes on the learning dynamics.

Due to the uniform initialisation of SGD in high dimensions, the following scalings close to initialisation are appropriate: $\alpha_u, \alpha_v \approx 1/\sqrt{N}$, as well as $\alpha_{u_m}, \alpha_{v_m} \approx 1/\sqrt{N}$. Hence, from Equation (9), the population loss scales as

$$L(w) \approx \frac{\lambda_{k_0}^2}{N^2} + \frac{\lambda_{k_0}^2}{N^2} (\sigma_\Sigma^2 - 1). \quad (11)$$

**Near-isotropic inputs:** If $\lambda_{k_0}, \lambda_m = O(1)$, then $L(w) \approx 1/N^2$, scaling as $\alpha_u^4$ and $\alpha_v^4$ at initialisation. It corresponds to information exponent $k^* = 4$ – also found in Theorem 3.1 for isotropic covariances. Hence, online SGD is expected to require a cubic sample complexity to distinguish structured inputs from noise (cf. Remark G.6).

**Power-law decaying inputs:** If $\lambda_{k_0} \approx \sqrt{N}, 1 \lesssim \lambda_m \lesssim N$, then $L(w) \approx 1/N$. Here, $\lambda_{k_0}$ is a top eigenvalue whose role is to reduce the information exponent from $k^* = 4$ to $k^* = 2$. In the setting of Ben Arous et al. (2021), this means recovering the signal at quasi-linear sample complexity. Note that the low-dimensional eigenstructure does not reduce the dynamics to a finite exploration, as it is often the result of common preprocessing procedure (like PCA). Instead, the effect of the power-law decaying spectrum is to induce an extensive signal-to-noise ratio $\lambda_{k_0}^2 \approx N$ in the loss which "couples" the signal and the quadratic form $\sigma_\Sigma^2$.

### 5.2. Dynamics of online SGD for a low-rank eigenstructure of the inputs

We now analyse the effect of power-law decaying amplitudes on the dynamics of online SGD, building on Ben Arous et al. (2022; 2025b). Motivated by the discussion in Section 5.1.1, we consider here an effective low-rank eigenstructure.

Let $M + 1 \in \mathbb{N}$ be a (finite) number of non-trivial, i.e. non-unit, top eigenvalues $((\lambda_m)_{m=1}^M, \lambda_{k_0})$, whose eigenvectors $((u^m, v^m)_{m=1}^M, u, v)$ span the principal subspace of the data. For simplicity, we set the remaining eigenvalues to one. We track the evolution during training of the summary statistics $\boldsymbol{\alpha} = (\alpha_u, \alpha_v, (\alpha_{u_m}, \alpha_{v_m})_{m=1}^M, \omega_\perp)$, where $\alpha_u = w \cdot u, \alpha_v = w \cdot v, \alpha_{u_m} = w \cdot u^m, \alpha_{v_m} = w \cdot v^m$ and $\omega_\perp = w \cdot w_\perp$, with $w_\perp$ being the projection of the weight vector onto the orthogonal of the principal subspace.

Within this framework, we consider non-normalised online SGD (29) initialised at a measure $\mu_0$ (cf. Section G) and add to the population loss a quartic penalisation term for the

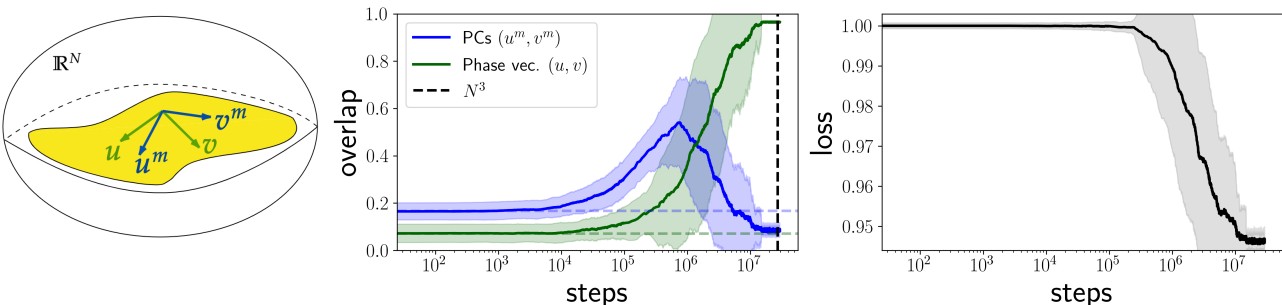

*Figure 4.* **Performance of SGD in classifying non-isotropic inputs on the Fourier data model. (Left)** Cartoon of the principal subspace of power-law-decaying inputs sampled from the Fourier data model (3), spanned by the DFT phase vectors $(u, v)$ and a finite number of other principal components $(u^m, v^m)$. **(Middle)** At first, online SGD quickly recovers the whole principal subspace, including the DFT phase vectors. Then, the other principal components PCs (—), which are not useful for classification, are forgot and SGD converges within $n \lesssim N^3$ steps to the subspace spanned by the DFT phase vectors (—). Compare with Figure 2, where no recovery happens within cubic sample complexity. Here, $N = 300$. **(Right)** Empirical loss function during training (—).

norm of the weights, i.e. $L'(w) = L(w) + \beta\|w\|^4, \beta \geq 0$. We choose non-normalised online SGD in order to match the conventions of Theorem G.3, originally from Ben Arous et al. (2022). Even if we expect that the normalisation does not change the predicted dynamics, proving a normalised counterpart of Theorem G.3 is beyond the scope of this work. On the same line, the penalisation term mimics the effect of the previously considered normalisation without dominating the dynamics; the presence of the penalty does not impact the time scales over which the phases are learnt (see computations of Lemmas G.5 and G.7).

At the population level, Conjectures 5.2 and 5.3 rely on the computation of the *population drift* $A_{\boldsymbol{\alpha}}$, i.e. the operator $A_{\boldsymbol{\alpha}}(\boldsymbol{\alpha}) = \lim_{N \to +\infty} \mathcal{A}_N \boldsymbol{\alpha}$, with $\mathcal{A}_N$ defined in Equation (30). The population drifts are rigorously computed in Lemmas G.5 and G.7. In the latter case, we compute the drift for rescaled statistics to avoid a diverging $A_{\boldsymbol{\alpha}} = +\infty$.

Beyond the population dynamics, we take a step towards sample complexities: we study the *effective dynamics* for infinitely many linear steps of online SGD (cf. Theorem G.3). Among the ingredients needed by this theorem, namely the population drift, the *population corrector* and the *effective volatility*, we do not treat the effective volatility. To our knowledge, it has been only computed in simpler Gaussian additive models e.g. single-index models or Gaussian mixtures (Ben Arous et al., 2022; 2025b). Hence, our results are stated as conjectures and not theorems.

We describe learning in the regimes of "small" (Conjecture 5.2), i.e. $O(1)$, and "large" (Conjecture 5.3), i.e. $O(N)$, top eigenvalues. We first claim that if the leading eigenvalues are $O(1)$, online SGD fails to recover phase information at linear sample complexity.

**Conjecture 5.2** (Near-isotropic inputs). *Consider* $\lambda_{k_0}, \lambda_m = O(1)$. *Then, for* $\delta_N = O(1/N)$ *and* $\mu_0 = \mathcal{N}(0, \mathbb{1}/N)$, *online* SGD *on the penalised loss* $L'$ *fails to weakly recover both the* DFT *phase vectors and the other principal components at linear sample complexity.*

*Derivation.* By Theorem G.3, the effective dynamics of the summary statistics, at linear sample complexity, is due to the population drift $A_{\boldsymbol{\alpha}}(\boldsymbol{\alpha})$ (computed in Lemma G.5), the effective volatility (which is expected to be zero like for Gaussian mixtures and the tensor PCA model (Ben Arous et al., 2025b; 2022)) and a vanishing population corrector (the statistics are linear in the weights, then $\mathcal{L}_N \boldsymbol{\alpha} = 0$ (30)). The dynamics has to be initialised deterministically at $\nu = \lim_{N \to +\infty} \boldsymbol{\alpha}_{\#} \mu_0 = \delta_0 \otimes \cdots \otimes \delta_0 \otimes \delta_1$. Then, $\boldsymbol{\alpha}(t) = \boldsymbol{\alpha}_N(t)$ converges for $N \to +\infty$ to the solution of $\dot{\boldsymbol{\alpha}}(t) = -A_{\boldsymbol{\alpha}}(\boldsymbol{\alpha}_t)$, initialised at $\boldsymbol{\alpha_0} = 0$, so SGD does not escape the initialisation (at linear sample complexity). $\square$

Note that the initialisation of the effective dynamics is non-vanishing anymore if we zoom into a microscopic neighbourhood of the summary statistics of the form

$$\boldsymbol{m} = (\sqrt{N}\alpha_u, \sqrt{N}\alpha_v, \ (\sqrt{N}\alpha_{u_m}, \sqrt{N}\alpha_{v_m})_{m=1}^M, \ \omega_\perp). \tag{12}$$

Indeed, e.g., $\lim_{N \to +\infty} m_{v\#}\mu_0 = \mathcal{N}(0, 1)$. However, if $\lambda_{k_0}, \lambda_m = O(1)$, the population drift $A_{\boldsymbol{m}}$ for the rescaled statistics turns out to be identically zero (cf. Remark G.9). Conversely, if $\lambda_{k_0} \approx \sqrt{N}$ – corresponding to a signal-to-noise $\lambda_{k_0}^2 \approx N$ in the loss (cf. Equation (11)) – the population drift for the rescaled statistics is non-vanishing anymore. We therefore turn to the case of extensive eigenvalue $\lambda_{k_0}$.

Note that, if the mode that carries the phase modification is the one with the largest eigenvalue, it could be identified with simple spectral methods like PCA. Instead, we consider a more interesting and realistic case: the mode $k_0$ is part of the leading subspace without having the largest eigenvalue. We conjecture that if the leading eigenvalues are $O(N)$, online SGD recovers phase information at quasi-linear sample complexity.

**Conjecture 5.3** (Power-law decaying inputs). *Consider* $\lambda_{k_0} \approx \sqrt{N}$ *and* $\sqrt{N} \lesssim \lambda_m \lesssim N$. *Then, for* $\delta_N = \Theta(1/N)$ *and* $\mu_0 = \mathcal{N}(0, \mathbb{1}/N)$, *online* SGD *on the penalised loss* $L'$ *weakly recovers all the principal components, including the* DFT *phase vectors, at quasi-linear sample complexity.*

*Derivation.* Due to the extensive eigenvalue $\lambda_{k_0}$, the population drift of the rescaled dynamics is not identically zero (cf. Lemma G.7). Moreover, due to the rescaling of the summary statistics, the limiting measure at initialisation $\nu$ is no longer deterministically zero. The fact that the summary statistics exhibit non-vanishing dynamics under $\sqrt{N}$ rescaling suggests that, in the original scale, online SGD escapes the search phase in quasi-linear time. $\qquad\square$

We leave it to future work to make our conjectures fully rigorous by also computing the diffusive terms and studying the resulting Ornstein-Uhlenbeck process (34). The key insight of Conjecture 5.3 is that online SGD recovers the leading subspace, which reflects the concentration of variance in the principal subspace of real images and it is shared among the classes of inputs, in quasi-linear sample complexity. Once this subspace is recovered, it is easier for SGD to converge to the mode that carries the phase modification.

We illustrate this effect with a small experiment on data sampled from the Fourier data model where we train a perceptron on a discrimination task where the covariance of both inputs has 6 modes with large eigenvalues (with same scalings of Conjecture 5.3), of which one carries the phase modification. In the middle of Figure 4, we show how the overlap of the perceptron weight with the principal subspace, excluding the mode $k_0$, increases (blue line (—)) before the perceptron converges to the mode with the non-trivial phase (green line (—)), and consequently forgets the principal components which are not task-relevant. Note that online SGD weakly recovers the whole principal subspace ((—) and (—)) at quasi-linear sample complexity and then converges to the signal within $n \lesssim N^3$ samples which, in the case of isotropic inputs, are not even sufficient for weak recovery (cf. Figure 2). Also, the loss only starts decaying after $10^6$ steps, rather than at $10^5$ steps when the whole subspace is already weakly recovered.

Importantly, in prior work by Ben Arous et al. (2024), a similar rich behavior emerges only under fine tuned signal-to-noise separation conditions, whereas here it is induced by the scalings of the Fourier amplitudes, leading to an interesting dynamics of online SGD even in the case of a single neuron. Moreover, in contrast to prior work by Mousavi-Hosseini et al. (2023), where SGD fails in presence of anisotropic covariances, we show with the present work that a non-trivial eigenstructure can speed up recovery of the relevant directions, instead of being a limitation.

Here, we show that classifying anisotropic inputs sampled from the Fourier data model (3) can be performed efficiently by SGD in presence of power-law decaying inputs, which incidentally happens to be the case of real images. Indeed, this structure leads to an extensive signal-to-noise ratio $\lambda_{k_0}^2 \approx N$ in the loss. We conclude then that the signal-to-noise ratio of a – by itself, hard to learn – phase recovery problem combined with a power-law on the amplitudes conspire to yield scaling behaviors similar to the ones of a tensor PCA problem (Richard & Montanari, 2014; Ben Arous et al., 2018) with an effective signal-to-noise ratio that makes it learnable at (quasi-)linear sample complexity.

## 6. Conclusions and future perspectives

Starting from the observation that, for translation-invariant inputs, Fourier amplitudes determine second-order statistics while phases encode higher-order correlations, we introduced a synthetic Fourier data model that makes this separation explicit and controllable. Our main interest was the sample complexity of learning from the phases, which are perceptually more important (see Figure 1) and whose non-Gaussian statistics shape neural representations (Olshausen & Field, 1996; Hyvärinen et al., 2009; Ingrosso & Goldt, 2022; Refinetti et al., 2023). Analysing the dynamics of SGD on a classification task where the relevant signal is carried only by the phases, we established sample complexities for extracting phase information in several scenarios. While learning from phases is a hard task when the covariance is (nearly) isotropic, we found both experimentally and theoretically that power-law spectra like the ones found in natural images can dramatically speed up learning from the phases. This speed-up is reminiscent of staircase effects in learning higher-order polynomials of the target functions (Abbe et al., 2021; 2023; Dandi et al., 2024a; Berthier et al., 2025) or higher-order statistics of the data (Ingrosso & Goldt, 2022; Bardone & Goldt, 2024; Mendes et al., 2026). Here, a crucial difference is that we see this speed-up even if the spectra themselves are not discriminative between the classes. We confirmed our theoretical results by simulations in more realistic settings with two-layer and deep convolutional networks trained on textures, ImageNet and CIFAR100.

The present work could be extended by deriving information theoretic thresholds for learning phase information, and by establishing algorithmic thresholds for the Fourier data model for more general classes of algorithms, like low-degree polynomials (Hopkins, 2018). Going further, it would be intriguing to perform a similar analysis in a different basis for images like wavelets (Mallat, 1999). Unlike Fourier modes, wavelets are localized both in space and frequency, and therefore provide a more refined description of edges, corners, and multiscale image features. Likewise, an extension to other texture models with controllable statistics (Victor JD, 2012; Portilla & Simoncelli, 2000; De Paolis et al., 2026) is a promising direction for future research. Extending our approach to these models may lead to a broader understanding of which data structures are easy or hard for gradient-based neural networks to learn from, and why natural images seem to live in a particularly favourable regime.

## Impact statement

This paper presents work whose goal is to advance the field of Machine Learning. There are many potential societal consequences of our work, none which we feel must be specifically highlighted here.

## Acknowledgements

We thank Giulio Biroli, Bruno Loureiro, Jonathan Victor, Gasper Tkačik, and Davide Zoccolan for inspiring discussions, and Matteo Santoro for the conversations on Fourier analysis. SG acknowledges funding from the European Research Council (ERC) under the European Union's Horizon 2020 research and innovation programme, Grant agreement ID 101166056, and funding from the European Union–NextGenerationEU, in the framework of the PRIN Project SELF-MADE (code 2022E3WYTY – CUP G53D23000780001). SG and FR acknowledge funding from Next Generation EU, in the context of the National Recovery and Resilience Plan, Investment PE1 – Project FAIR "Future Artificial Intelligence Research" (CUP G53C22000440006).

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

# A. Further related works

**Frequency bias of deep neural networks**    Previous work established simplicity biases in Fourier space (Rahaman et al., 2019). The crucial difference to our work is that they consider a simplicity bias in terms of the frequencies of the different Fourier modes of the data. Here, we are instead interested in sequential learning of amplitudes vs. phases, irrespective of the frequency of the given mode. For works studying (linear) CNNs from a Fourier perspective, see Pinson et al. (2023); Gunasekar et al. (2018).

**Perceptual role of amplitude vs. phase information**    Cognitive scientists have long been interested in the different perceptual roles played by phases and amplitudes in visual perception. While early work suggested that humans use only pair-wise information to discriminate structured visual stimuli from pure noise (Julesz, 1962), more recent work showed that for humans, phase information is perceptually dominant and determines the structure of an image (Oppenheim & Lim, 1981; Piotrowski & Campbell, 1982). More recently, Tkačik et al. (2010); Caramellino et al. (2021) have investigated the perceptual role of pair-wise and higher-order correlations more systematically. In contrast to humans, Geirhos et al. (2018) empirically validates that ImageNet-trained CNNs are strongly biased towards recognising textures rather than shapes.

# B. Experimental details

In this appendix, we collect detailed information on how we ran the experiments of this paper.

## B.1. Figure 1

We use greyscale images from the "ALOT" dataset (Burghouts & Geusebroek, 2009), which we downsample to image patches of sizes $8 \times 8$, $16 \times 16$ and $32 \times 32$. We then flatten these images into vectors of lengths $N = 8^2, 16^2, 32^2$ and train a two-layer neural network parametrized by

$$h_i = \sum_j (w_1)_{ij} \, x_j + (b_1)_i,$$

$$f_\theta(x) = \sum_i (w_2)_i \sigma(h_i) + b_2,$$

where $w_1 \in \mathbb{R}^{k \times N}$, $w_2 \in \mathbb{R}^k$, $b_1 \in \mathbb{R}^k$ and $b_2 \in \mathbb{R}$ are all trainable parameters, and we choose $k = 30$. We train to classify textures of type "cotton" (label $= 1$) from textures of type "lace" (label$= -1$). For all experiments, we used the mean squared error as objective function, and trained with learning rate $\eta = 10^{-3}$. We show examples of both textures in Figure 5 a-b) and the test accuracies of the classifiers evaluated on a held-out test set of $20\%$ of the data in Figure 5 c). The empirical test loss is shown in Figure 1 g). We also compute the DFT of the test images, swap the phases sample-wise between classes, and evaluate the models' performance on these phase-swapped test set. This is also shown in Figure 1 g).

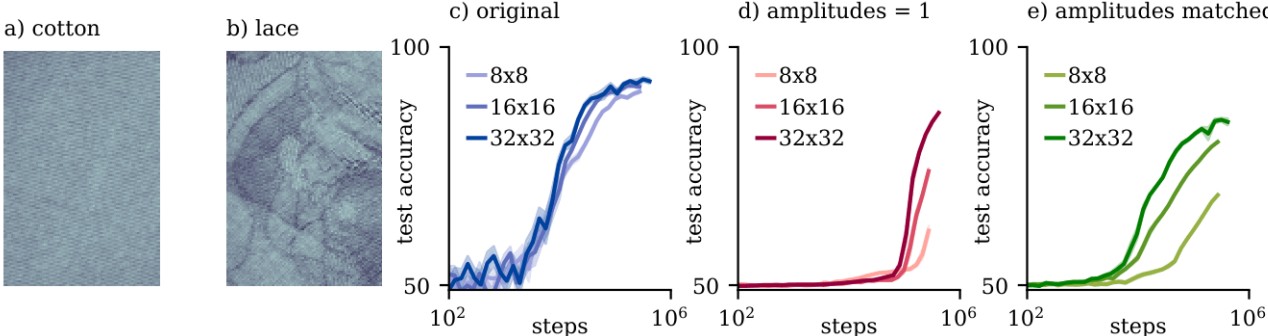

*Figure 5.* **a-b)** Examples of images from ALOT dataset for the classes "cotton" and "lace". **c)** Test accuracy of trained neural networks on original data. **d)** Test accuracies of neural networks on the same data, but where the amplitudes of all images have been set to one. **e)** Test accuracy on the same dataset, but where the amplitudes of class $-1$ are the same as the amplitudes of the class 1. All curves are averages over 8 random initializations.

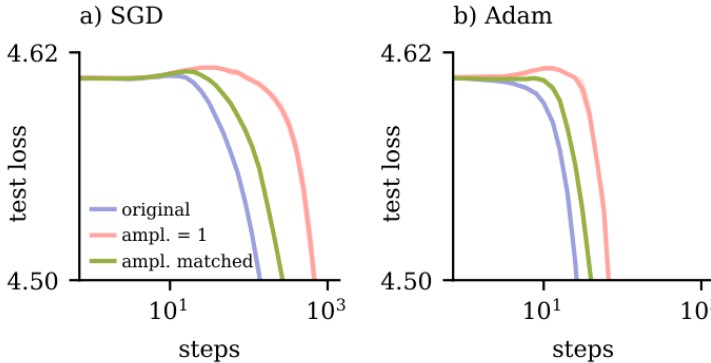

*Figure 6.* **Test loss curves of a Resnet18 trained on CIFAR100** with a) SGD and b) Adam on original data (—), data where all Fourier amplitudes have been set to one (—) and data where the amplitudes of all classes are replaced by the amplitudes of the first class (—). All the curves are averaged over three independent runs. We use the cross-entropy loss. *Parameters:* for SGD, we use learning rate $5 \cdot 10^{-3}$, momentum = 0.9, weight decay of $5 \cdot 10^{-3}$, batch size of 128. For Adam, we used learning rate $10^{-3}$, $\beta = (0.9, 0.99)$, and no weight decay. Cf. Figure 3 to visualise the same trend on textures and ImageNet.

## B.2. Figure 2

**Left:** We show here that online SGD (5) cannot access easily the information encoded in the phase of isotropic translation-invariant inputs drawn accordingly to the Fourier data model (3). We run SGD on inputs of dimension $N = 100$, up to $N^4$ steps, and we perform 60 averages. The parameters of the Fourier data model are $\varepsilon = 1.2$ and the modified entry is $k_0 = 6$. The activation function is the 4-th Hermite polynomial $\sigma(s) = s^4 - 6s^2 + 3$ and the learning rate for SGD is $\delta_N = 10^{-3}/N$.
**Right:** The empirical loss landscape has been computed in the same setting. The level sets on the loss landscape are shown in Figure 9.

## B.3. Figure 3

### B.3.1. PANELS A-D)

We use the same texture dataset as in Figure 1 (see Section B.1) cropped into patches of sizes $8 \times 8$, $16 \times 16$ and $32 \times 32$. We compute the (orthogonal) DFT of all images. In Figure 3 a), we show their average squared Fourier amplitudes. Then, we train classifiers to distinguish between the "cotton" and "lace" texture types, using the same architecture as in Section B.1, in three different cases. First, we use the original texture data. Second, we set all Fourier amplitudes equal to one, and transform back. Finally, we transplant the amplitudes of the "cotton" class onto the "lace" class sample by sample, then transform back. In this last case, we have built two classes of inputs which share the same second order statistics, but have different Fourier phases. The test accuracies of the networks trained on the said image data are displayed in Figure 5 c-e). The corresponding test losses are shown in Figure 3 b-d), with (—) for the original data, (—) for the whitened data and (—) for the data with matched amplitudes.

### B.3.2. PANEL E) IMAGENET EXPERIMENT WITH RESNET18

We trained a standard Resnet18 architecture on a super-classed version of ImageNet with ten classes[1] provided by the `robustness` library of Engstrom et al. (2019). We performed the same amplitude manipulations as in the texture experiments. We trained the network using vanilla SGD with a learning rate of $10^{-3}$, weight decay of $5 \cdot 10^{-4}$, momentum of 0.9, and mini-batch size of 128. In all three data sets, we set the mean image of each class to zero, and we normalised inputs in all three data sets (vanilla images, images with flat amplitudes, images with shared amplitudes) to have the same mean and standard deviation.

---

[1]Dog (n02084071), Bird (n01503061), Insect (n02159955), Monkey (n02484322), Car (n02958343), Cat (n02120997), Truck (n04490091), Fruit (n13134947), Fungus (n12992868), Boat (n02858304)

## B.4. Figure 4

**(Left)** Cartoon of the principal subspace of inputs sampled from the Fourier data model, in case of non-isotropic power-law-decaying inputs. **(Middle)** The ambient dimension is $N = 300$. The parameters for the Fourier data model (3) are $\varepsilon = 1.2$ and $k_0 = 6$. The other $M = 5$ frequencies for which the corresponding eigenvalue $\lambda_m$ is extensive are $k_1 = 15, k_2 = 24, k_3 = 20, k_4 = 9$ and $k_5 = 18$. The extensive eigenvalues are $(\lambda_{k_0}, \lambda_{k_1}, \lambda_{k_2}, \lambda_{k_3}, \lambda_{k_4}) = (N^{1/2}, N^{1.2/2}, N^{1.1/2}, N^{1.3/2}, N^{1.4/2}, N^{0.9/2})$. Normalised online SGD is performed with $\delta_N = 0.03/N$, for 60 averages. The circulant matrix for $\mathbb{P}_0$ and $\mathbb{P}$ has eigenvalues $(\lambda_{k_0}, \lambda_{k_1}, \lambda_{k_2}, \lambda_{k_3}, \lambda_{k_4})$, and all the rest set to one. As always, the principal subspace is shared among the classes $\mathbb{P}_0$ and $\mathbb{P}$. **(Right)** For the same parameters, we show the empirical loss function.

# C. Preliminaries

## C.1. Notation

W use the standard notations $o, O, \Theta, \Omega, \omega$ for asymptotics. We recall them here for sequences, but they generalize in a straight forward way in the case of functions.

Let $(a_k)_{k \in \mathbb{N}}$, $(b_k)_{k \in \mathbb{N}}$ be two real-valued sequences. Then,

$$a_k \in o(b_k) \iff \lim_{k \to \infty} \frac{a_k}{b_k} = 0,$$

$$a_k \in O(b_k) \iff \exists\, C > 0 \in \mathbb{R} \quad \forall k > k_0 \quad |a_k| \leq C|b_k|,$$

$$a_k \in \Theta(b_k) \iff \exists\, C_1, C_2 > 0 \quad \forall k > k_0 \quad C_1|b_k| \leq |a_k| \leq C_2|b_k|,$$

$$a_k \in \Omega(b_k) \iff \exists\, C > 0 \in \mathbb{R} \quad \forall k > k_0 \quad |a_k| \geq C|b_k|,$$

$$a_k \in \omega(b_k) \iff \lim_{k \to \infty} \frac{|a_k|}{|b_k|} = \infty.$$

Occasionally, we use the shorthands $a_k \ll b_k$ for $a_k = o(b_k)$, $a_k \lesssim b_k$ for $a_k = O(b_k)$ and $a_k \asymp b_k$ for $a_k = \Theta(b_k)$.

**Pushforward**   Given a measurable map $T : X \to Y$ and a probability measure $\mu$ on $X$, we denote by $T_{\#}\mu$ the pushforward of $\mu$ by $T$ defined by $T_{\#}\mu(A) = \mu(T^{-1}(A))$, for any measurable set $A \subset Y$.

## C.2. Fourier analysis

**Definition C.1** (DFT). The Discrete Fourier Transform (DFT) is the linear map $\mathcal{F} : \mathbb{R}^N \to \mathbb{C}^N$ such that for $k = 0, \ldots, N-1$ we have

$$\mathcal{F}(x)_k = \sum_{n=0}^{N-1} x_n e^{-2\pi i n \frac{k}{N}},$$

for any $x \in \mathbb{C}^N$. We denote with $F \in \mathbb{C}^{N \times N}$ the DFT matrix defined as $F_{kn} = e^{-i2\pi kn/N}$. The Inverse Discrete Fourier Transform (IDFT) is instead $\mathcal{F}^* : \mathbb{C}^N \to \mathbb{C}^N$ such that for any $X \in \mathbb{C}^N$ we have

$$\mathcal{F}^*(X)_n = \frac{1}{N} \sum_{k=0}^{N-1} X_k e^{2\pi i k \frac{n}{N}}.$$

In what follows, we will use the notations $X = \mathrm{DFT}(x) = \mathcal{F}(x)$ and $x = \mathrm{DFT}(X) = \mathcal{F}^*(X)$. Another possible choice would be the normalised DFT matrix defined as $\widehat{F} = \frac{1}{\sqrt{N}}F$.

Note that, for real inputs $x \in \mathbb{R}^N$, their DFT exhibits conjugate symmetry, that is $X_k = \overline{X}_{N-k}$. As a consequence, the non-redundant frequencies are only $k = 0, \ldots, \lfloor N/2 \rfloor$. The frequencies $k = 0$ and $k = N/2$, if $N$ is even, are called DC and Nyquist frequencies, respectively. They correspond to the real DFT Fourier coefficients $X_0$ and $X_{N/2}$. Being purely real, these entries are not candidates to exhibit a perturbation of the phase in the sense of the Fourier data model (3), for which we consider $k_0 \notin \{0, N/2\}$.

The next definition identifies a set of particular vectors in $\mathbb{R}^N$. Since these vectors are orthonormal, they form an orthonormal basis of $\mathbb{R}^N$, which we refer to as the (real) DFT basis.

**Definition C.2** (DFT basis). The cosine DFT basis vectors $(u^k)_k$ and the sine DFT basis vectors $(v^k)_k$ in $\mathbb{R}^N$, for the frequencies $k = 1, \ldots, \lfloor \frac{N-1}{2} \rfloor$, are defined by

$$u_n^k = \sqrt{\frac{2}{N}} \cos\left(\frac{2\pi kn}{N}\right) \qquad \text{and} \qquad v_n^k = \sqrt{\frac{2}{N}} \sin\left(\frac{2\pi kn}{N}\right),$$

for any $n = 0, \ldots, N - 1$. In addition, we define the constant basis vectors $\psi^0, \psi^{N/2}$ such as $\psi_n^0 = 1/\sqrt{N}$ and $\psi_n^{N/2} = (-1)^n/N$, for $n = 0, \ldots, N-1$. The DFT basis consists of the constant basis vectors $\psi^0, \psi^{N/2}$, the cosine DFT basis vectors $u^k$ and the sine DFT basis vectors $v^k$.

### C.3. Translation-invariant inputs

In many fields like visual processing and statistical modeling, it is very common (Simoncelli & Olshausen, 2001), (Hyvärinen et al., 2009) to assume that natural images and textures exhibit wide-sense stationarity, meaning that their second-order statistics are translation-invariant or, to put it differently, that their covariance matrix is *Toeplitz*, in the sense that each entry of the matrix depends only on the relative distance of the corresponding indices.

**Definition C.3** (Toeplitz matrix). A matrix $\Sigma \in \mathbb{C}^{N \times N}$ is "Toeplitz" if there exists a function $c : \mathbb{Z} \to \mathbb{C}$ such that

$$\Sigma_{ij} = c(i - j).$$

A zero-mean random vector is said to be "wide-sense stationary" if its covariance matrix is Toeplitz.

For matrices, *circularity* is a more restrictive property with respect to the just mentioned notion of invariance. Indeed, circulant matrices form a proper subset of Toeplitz ones, since they impose additional constraints on the boundary conditions. The entries of a Toeplitz matrix are constant along its diagonals, while each row of a circulant matrix is a circular shift of the previous one, in the following sense.

**Definition C.4** (Circulant matrix). A matrix $\Sigma \in \mathbb{C}^{N \times N}$ is said to be "circulant" if each entry is such that $\Sigma_{ij} = c((i - j) \bmod N)$, for a function $c : \mathbb{Z} \to \mathbb{C}$. This means that there exist $N$ coefficients $c_0, \ldots, c_{N-1}$ such as

$$\Sigma = \begin{pmatrix} c_0 & c_{N-1} & c_{N-2} & \cdots & c_1 \\ c_1 & c_0 & c_{N-1} & \cdots & c_2 \\ c_2 & c_1 & c_0 & \cdots & c_3 \\ \vdots & \vdots & \vdots & \ddots & \vdots \\ c_{N-1} & c_{N-2} & c_{N-3} & \cdots & c_0 \end{pmatrix}.$$

In the Fourier data model (3), we ask that the covariance matrix of the original inputs $z \in \mathbb{R}^N$ is circulant, rather than only Toeplitz. We require this in order to have the covariance matrix of the inputs diagonalized by the DFT basis defined in Definition C.2. For this reason, in this work, we say that the inputs are *translation-invariant* if they satisfy Definition C.5. We emphasize that the notion of translation-invariance adopted here is not the standard one, which often refers to random vectors whose distribution is invariant under translation. However, in the classical sense, translation invariance implies wide-sense stationarity, in the sense of Definition C.3. Moreover, in high-dimensional settings, there are some well-established results (e.g. (Zhu & Wakin, 2016)) proving that Toeplitz matrices can be approximated by circulant matrices in a spectral sense, meaning that the eigenvalue distribution of a proper circulant approximation captures the bulk of energy in the Toeplitz covariance matrix. These results are built on classical limit theorems from (Szegö, 1952) and (Böttcher & Silbermann, 1990). In view of these considerations, the proposed definition of translation-invariant inputs is appropriate in the context of modeling high-dimensional natural images or textures.

**Definition C.5** (Translation-invariant inputs). A zero-mean random vector $x \in \mathbb{R}^N$ is said to be "translation-invariant" if its covariance matrix is circulant.

Because of the symmetry of any covariance matrix, a translation-invariant random vector has covariance $\Sigma$ such that $c_{(i-j) \bmod N} = \Sigma_{ij} = \Sigma_{ji} = c_{(j-i) \bmod N}$, for some coefficients $(c_0, \ldots, c_{N-1})$. Hence, given $\tau = (i - j) \bmod N$, it holds $(j - i) \bmod N = N - \tau$, from which it follows that $c_\tau = c_{N-\tau}$ for $\tau = 1, \ldots, N - 1$. Then, the only free entry $c_0$ is on the

main diagonal, while the other entries $(c_1, \ldots, c_{N-1})$ are forced to be symmetric about the center, leading to the following:

$$
\Sigma = \begin{pmatrix}
c_0 & c_1 & c_2 & \cdots & c_2 & c_1 \\
c_1 & c_0 & c_1 & \cdots & c_3 & c_2 \\
c_2 & c_1 & c_0 & \cdots & c_4 & c_3 \\
\vdots & \vdots & \vdots & \ddots & \vdots & \vdots \\
c_2 & c_3 & c_4 & \cdots & c_0 & c_1 \\
c_1 & c_2 & c_3 & \cdots & c_1 & c_0
\end{pmatrix}.
$$

In the next lemma, we prove that it is possible to diagonalise any circulant covariance matrix through the DFT basis. This is the main reason why circulant matrices are fundamental in studying the structure of inputs distributed according to the Fourier data model (3).

**Lemma C.6** (Diagonalisation of circulant matrices). *Any symmetric circulant matrix is diagonalized by the* DFT *basis.*

*Proof.* For any circulant matrix $\Sigma$, it holds that $\Sigma_{ij} = c_{(i-j) \bmod N}$. It is well known that for any $k = 1, \ldots, \lfloor \frac{N-1}{2} \rfloor$, the vectors

$$
w^k = \frac{1}{\sqrt{N}} \begin{pmatrix} 1 \\ \omega^k \\ \omega^{2k} \\ \vdots \\ \omega^{(N-1)k} \end{pmatrix},
$$

where $\omega = e^{i2\pi/N}$ is the fundamental $N$-th root of unity, are (complex) eigenvectors of $\Sigma$. This is proven for example in the standard reference (Davis, 1994). Moreover, $\Sigma$ is symmetric and then its eigenvalues $(\lambda_0, \ldots, \lambda_{\lfloor \frac{N-1}{2} \rfloor})$ are real. Since

$$
w_n^k = \frac{1}{\sqrt{N}} \cos\left(\frac{2\pi k n}{N}\right) + i\frac{1}{\sqrt{N}} \sin\left(\frac{2\pi k n}{N}\right),
$$

the DFT basis vectors defined Definition C.2 are (real) eigenvectors of $\Sigma$ corresponding to the real eigenvalue $\lambda_k$. □

The next lemma is at the basis of the already mentioned disentanglement between low-order and higher-order correlations of translation-invariant inputs or, in Fourier terms, between Fourier amplitudes and Fourier phases.

**Lemma C.7.** *Consider a translation-invariant random vector $z \in \mathbb{R}^N$ and its* DFT*, i.e. $Z = \mathrm{DFT}(z)$, with $Z_k = \rho_k e^{i\varphi_k}$ for $k = 0, \ldots, N - 1$. Then, its covariance matrix $\mathrm{Cov}[Z]$ is diagonal and such that*

$$
\mathrm{Cov}[Z]_{kk} = \mathbb{E}[\rho_k^2].
$$

*Proof.* Since $z$ has circulant covariance, for any $n, m = 0, \ldots, N - 1$, denote its entries by the coefficients $c_{(n-m) \bmod N} = \mathbb{E}[z_n z_m]$. Therefore,

$$
\mathbb{E}[Z_k \overline{Z}_l] = \sum_{n,m=0}^{N-1} e^{-2\pi i(kn+lm)/N} \underbrace{\mathbb{E}[z_n z_m]}_{c_{(n-m) \bmod N}} = \sum_{\tau=0}^{N-1} c_\tau \sum_{m=0}^{N-1} e^{-2\pi i(k(m+\tau)-ml)},
$$

where $\tau = (n - m) \bmod N$. If we define $S_k = \sum_{\tau=0}^{N-1} c_\tau e^{-2\pi i \tau k/N}$, it holds that

$$
\mathbb{E}[Z_k \overline{Z}_l] = S_k \sum_{m=0}^{N-1} e^{-2\pi i m(k-l)/N} = \begin{cases} S_k N & \text{if } k \equiv_N l, \\ 0 & \text{otherwise.} \end{cases}
$$

Now, note that for any $k = 0, \ldots, N - 1$, we can write that

$$
\mathbb{E}[\rho_k^2] = \mathbb{E}[Z_k \overline{Z}_k] = \sum_{n,m=0}^{N-1} e^{-2\pi i k(n-m)} \mathbb{E}[z_n z_m] = \underbrace{\sum_{\tau=0}^{N-1} c_\tau e^{-2\pi i k\tau}}_{S_k} N,
$$

and then $\mathrm{Cov}[Z]_{kl} = \mathbb{E}[Z_k \overline{Z}_l] = \mathbb{E}[\rho_k^2]\delta_{k \equiv_N l}$, which implies the thesis. □

### C.4. Amplitudes and phases of the DFT coefficients

In this small section, we investigate closely the distributions of the amplitudes and phases of a Gaussian-distributed translation-invariant random vector (cf. Definition C.5).

**Definition C.8** (Rayleigh distribution)**.** A scalar random variable $Y$ is said to be Rayleigh-distributed with parameter $\sigma > 0$, and we will write $Y \sim \text{Rayleigh}(\sigma)$, if its probability density function is given by

$$
p_Y(y) = \begin{cases} \dfrac{y}{\sigma^2} \exp\left(-\dfrac{y^2}{2\sigma^2}\right) & y \geq 0, \\ 0 & y < 0. \end{cases}
$$

We can note that the square root of the sum of two squared independent Gaussian random variables is distributed according to a Rayleigh distribution.

*Remark* C.9. Given $Z_1, Z_2 \sim \mathcal{N}(0, \sigma^2)$ independent random variables, $Y = \sqrt{Z_1^2 + Z_2^2}$ follows a Rayleigh distribution with parameter $\sigma$.

*Remark* C.10 (Statistics of Rayleigh distribution)**.** Consider $Y \sim \text{Rayleigh}(\sigma)$. Then,

$$
\mathbb{E}[Y] = \sigma\sqrt{\frac{\pi}{2}}, \qquad \text{Var}(Y) = \frac{4 - \pi}{2}\sigma^2, \qquad \mathbb{E}[Y^k] = \sigma^k 2^{k/2}\Gamma\left(1 + \frac{k}{2}\right),
$$

where $\Gamma(\cdot)$ is the Gamma function and $k \geq 2$.

In the next lemma we prove that the DFT coefficients of translation-invariant Gaussian noise are independent and that their amplitudes and phases are Rayleigh- and uniform-distributed, respectively.

**Lemma C.11.** *Consider a zero-mean translation-invariant Gaussian random vector $z \in \mathbb{R}^N$. Then, its* DFT *coefficients $Z_k$ are independent and*

$$
\rho_k \sim \text{Rayleigh}\left(\sqrt{\frac{N\lambda_k}{2}}\right), \qquad \varphi_k \sim \text{Unif}([-\pi, \pi)),
$$

*where $\lambda_k$ is the k-th eigenvalue of the covariance matrix $\Sigma$ of the inputs.*

*Proof.* Any Gaussian-distributed random vector $z \in \mathbb{R}^N$ has a Gaussian DFT $Z = \text{DFT}(x)$. Hence, each entry, and the corresponding real and imaginary parts are Gaussian-distributed. Since $\Sigma$ is circulant, the entries $Z_k$ are uncorrelated, and then independent. We look now at the statistics of $\text{Re}(Z_k)$ and $\text{Im}(Z_k)$. The real part of the Fourier coefficient is $\text{Re}(Z_k) = \sum_{n=0}^{N-1} x_n \cos\left(\frac{2\pi kn}{N}\right)$. We want to compute

$$
\text{Var}(\text{Re}(Z_k)) = \mathbb{E}\left[\left(\sum_{n=0}^{N-1} z_n \cos\left(\frac{2\pi kn}{N}\right)\right)\left(\sum_{m=0}^{N-1} z_m \cos\left(\frac{2\pi km}{N}\right)\right)\right]
$$

$$
= \sum_{n=0}^{N-1}\sum_{m=0}^{N-1} E[z_n z_m]\cos\left(\frac{2\pi kn}{N}\right)\cos\left(\frac{2\pi km}{N}\right).
$$

Since $E[z_n z_m] = c_{(n-m) \bmod N}$,

$$
\text{Var}(\text{Re}(Z_k)) = \frac{1}{2}\sum_{m,n=0}^{N-1} c_{(n-m) \bmod N}\left[\cos\left(\frac{2\pi k(n-m)}{N}\right) + \cos\left(\frac{2\pi k(n+m)}{N}\right)\right], \tag{13}
$$

where we have applied the trigonometric identity $\cos(A)\cos(B) = 1/2\left[\cos(A - B) + \cos(A + B)\right]$.
We can split Equation (13) into two sums: $\text{Var}(\text{Re}(Z_k)) = S_1 + S_2$. Define $\tau = (n - m) \bmod N$. For any fixed $n$, as $m$ cycles through $0, \ldots, N - 1$, the index $r$ also cycles through $0, \ldots, N - 1$. Then,

$$
S_1 = \frac{1}{2}\sum_{n=0}^{N-1}\left(\sum_{m=0}^{N-1} c_{(n-m) \bmod N}\cos\left(\frac{2\pi k(n-m)}{N}\right)\right).
$$

The inner sum is the definition of the $k$-th eigenvalue $\lambda_k$ for the circulant covariance matrix $\Sigma$, since

$$\sum_{r=0}^{N-1} c_r \cos\left(\frac{2\pi kr}{N}\right) = \mathrm{Re}(\lambda_k) = \lambda_k.$$

Thus, $S_1 = \frac{1}{2}\sum_{n=0}^{N-1} \lambda_k = (N\lambda_k)/2$. For $S_2$, we get that

$$S_2 = \frac{1}{2}\sum_{r=0}^{N-1} c_r \sum_{m=0}^{N-1} \cos\left(\frac{2\pi k(2m+r)}{N}\right)$$

and, by expanding the cosine, it hods that

$$S_2 = \frac{1}{2}\sum_{r=0}^{N-1} c_r \left[\cos\left(\frac{2\pi kr}{N}\right)\sum_{m=0}^{N-1}\cos\left(\frac{4\pi km}{N}\right) - \sin\left(\frac{2\pi kr}{N}\right)\sum_{m=0}^{N-1}\sin\left(\frac{4\pi km}{N}\right)\right].$$

For $0 < k < N/2$, the value $2k$ is not a multiple of $N$. Therefore, the sum of the sinusoids over $N$ points is zero, i.e.

$$\sum_{m=0}^{N-1} e^{j\frac{2\pi(2k)m}{N}} = 0.$$

In conclusion, $S_2 = 0$. Then, $\mathrm{Re}(Z_k) \sim \mathcal{N}(0, N\lambda_k/2)$. Since the same is true for the imaginary part, the first part of the thesis follows from the definition of a Rayleigh-distributed random variable. For a fixed frequency $k$, we denote now $A = \mathrm{Re}(X_k)$, $B = \mathrm{Im}(X_k)$ and $\sigma^2 = (\lambda_k N)/2$. Then, the joint distribution for $(A, B)$ is

$$f_{A,B}(a,b) = \frac{1}{2\pi\sigma^2} e^{-\frac{a^2+b^2}{2\sigma^2}}.$$

In the polar coordinates $A = R\cos(\phi)$ and $A = R\sin(\phi)$, we have $f_{R,\phi}(r,\varphi) = f_{A,b}(r\cos\varphi, r\sin\varphi)|J|$, for $J$ Jacobian of the transformation. Hence,

$$f_{R,\phi}(r,\varphi) = \frac{r}{2\pi\sigma^2}e^{-r^2/2\sigma^2} = \frac{r}{\sigma^2}e^{-\frac{r^2}{2\sigma^2}}\frac{1}{2\pi}$$

and marginalising we obtain the thesis:

$$f_\phi(\varphi) = \int_0^{+\infty} f_{R,\phi}(r,\varphi)dr = \frac{1}{2\pi}\int_0^{+\infty}\frac{r}{\sigma^2}e^{-\frac{r^2}{2\sigma^2}}\,dr = \frac{1}{2\pi}.$$

$\square$

### C.5. Hermite polynomials

In this section, we recall some basic definitions about Hermite polynomials (Szegő, 1939), which happens to be useful tools to study learning dynamics for classical estimation problems. In what follows, $\mathbb{P}_0$ denotes a univariate or multivariate normal Gaussian distribution. However, in some cases, $\mathbb{P}_0$ may admit a non-trivial covariance matrix. This will be specified explicitly when needed.

**Definition C.12** (Gaussian product). Given $f, g : \mathbb{R}^N \to \mathbb{R}$ and $\mathbb{P}_0$ multivariate normal Gaussian distribution, we define the $L^2$-scalar product as

$$\langle f, g \rangle_{\mathbb{P}_0} = \mathbb{E}_{\mathbb{P}_0}\left[f(x)g(x)\right].$$

The space $L^2(\mathbb{R}^N, \mathbb{P}_0)$ contains all the measurable functions $f : \mathbb{R}^N \to \mathbb{R}$ such that $||f||^2 = \langle f, f \rangle_{\mathbb{P}_0} < \infty$.

**Definition C.13** (Hermite polynomials). The (non-normalised) probabilists' Hermite polynomials $(h_k)_k$ are defined by $h_0(x) = 1, h_1(x) = x$ and, for $k \geq 1$, recursively by

$$h_{k+1}(x) = xh_k(x) - kh_{k-1}(x).$$

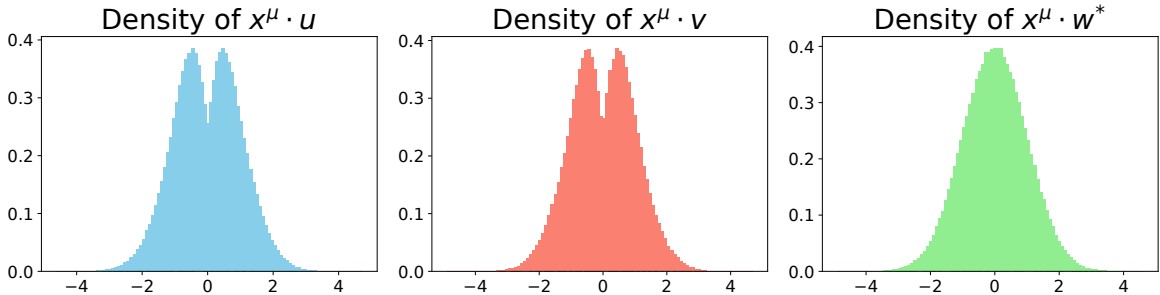

*Figure 7.* **Distributions of the inputs drawn from the Fourier data model along the DFT phase vectors.** Consider the DFT phase vectors $u$ and $v$ (cf. Definition C.2) and a fixed vector $w^*$ uniformly drawn from the sphere in dimension $N = 30$. The DFT phase vectors span the only subspace of $\mathbb{R}^N$ along which the projected inputs are non-Gaussian-distributed. Indeed, (**left**) and (**middle**) are non-Gaussian, whereas (**right**) is Gaussian.

The first Hermite polynomials are

$$
\begin{aligned}
h_0(x) &= 1, \\
h_1(x) &= x, \\
h_2(x) &= x^2 - 1, \\
h_3(x) &= x^3 - 3x, \\
h_4(x) &= x^4 - 6x^2 + 3.
\end{aligned}
$$

**Lemma C.14** (Orthogonality property). *Let $\mathbb{P}_0$ be the multivariate normal Gaussian distribution. Consider $w_1, w_2 \in \mathbb{S}^{N-1}$ and set $\alpha = w_1 \cdot w_2$. Then, for any $i, j \in \mathbb{N}$,*

$$
\mathbb{E}_{\mathbb{P}_0}[h_i(w_1 \cdot x) h_j(w_2 \cdot x)] = i! \, \alpha^i \, \delta_{ij}.
$$

**Definition C.15** (Hermite expansion). Consider a function $f : \mathbb{R} \to \mathbb{R}$ that is square integrable with respect to the univariate Gaussian normal distribution $\mathbb{P}_0$. Then, there exists a unique sequence of real numbers $(c_k)_{k \in \mathbb{N}}$ called Hermite coefficients such that

$$
f(x) = \sum_{k=0}^{\infty} \frac{c_k}{k!} h_k(x) \quad \text{and} \quad c_k(x) = \mathbb{E}_{\mathbb{P}_0}[f(x) h_k(x)],
$$

where $h_k$ is the $k$-th probabilists' Hermite polynomial.

### C.6. Information exponent and likelihood ratios

**Definition C.16** (Information exponent). **Given any function $f$ for which the Hermite expansion exists, its information exponent $k^* = k^*(f)$ is the smallest index $k \geq 1$ such that $c_k \neq 0$.**

In order to compute the information exponent of the population correlation loss (6) for non-Gaussian distributions, we can exploit the *likelihood ratio*. It allows us to rewrite the average over the non-Gaussian inputs as an average over the multivariate Gaussian distribution $\mathbb{P}_0 = \mathcal{N}(0, \Sigma)$.

**Definition C.17** (Likelihood ratio). Let $\mathbb{P}$ and $\mathbb{P}_0$ be two probability distributions on $\mathbb{R}^N$. If $\mathbb{P}$ is absolutely continuous with respect to $\mathbb{P}_0$, then there exists a measurable function

$$
\frac{d\mathbb{P}}{d\mathbb{P}_0} : \mathbb{R}^N \to [0, \infty)
$$

such that for every measurable $A \subseteq \mathbb{R}^N$, we get that

$$
P(A) = \int_A \frac{d\mathbb{P}}{d\mathbb{P}_0} \, d\mathbb{P}_0.
$$

The function $\ell = d\mathbb{P} \backslash d\mathbb{P}_0$ is called "likelihood ratio" (or "Radon–Nikodym derivative") of $\mathbb{P}$ with respect to $\mathbb{P}_0$. If both $\mathbb{P}$ and $\mathbb{P}_0$ admit densities $p$ and $q$ with respect to the Lebesgue measure, then, for any $y \in \mathbb{R}^N$, we have that $\ell(y) = p(y)/q(y)$.

Assume that $\mathbb{P}$ is a non-Gaussian distribution, $\mathbb{P}_0$ is a Gaussian distribution and that the likelihood ratio between $\mathbb{P}$ and $\mathbb{P}_0$ depends only on the projection of the inputs along a fixed number of directions $\ell = \ell(w_1^* \cdot x, \ldots, w_k^* \cdot x)$. Intuitively, the likelihood ratio (or more precisely, its norm) measures how different the distribution of the projection $s = w \cdot x$ is from a standard Gaussian distribution, when the weight vector $w$ belongs to the subspace spanned by $\{w_1^*, \ldots, w_k^*\}$. Given that, we can write

$$\mathbb{E}_{\mathbb{P}}[\sigma(w \cdot x)] = \mathbb{E}_{\mathbb{P}_0}[\sigma(w \cdot x)\ell(w_1^* \cdot x, \ldots, w_k^* \cdot x)]. \tag{14}$$

The likelihood ratio will thus be the key object in our study to determine the number of samples that a given algorithm like SGD requires to find a projection that is distinctly non-Gaussian and yields a large likelihood ratio, akin to how one would proceed in an analysis with the second moment method in hypothesis testing (Kunisky et al., 2019).

## D. Statistical properties of the Fourier data model

Here, we recall what the definition of the Fourier data model (3) is. Assume that $(z^\mu)_{\mu=0}^n \subseteq \mathbb{R}^N$ are $n$ samples drawn from $\mathbb{P}_0 = \mathcal{N}(0, \Sigma)$, with $\Sigma \in \mathbb{R}^{N \times N}$ circulant matrix. Consider $Z^\mu = \mathrm{DFT}(z^\mu)$, with $Z_k^\mu = \rho_k^\mu e^{i\varphi_k^\mu}$ for $k = 0, \ldots, N - 1$. Because of Lemma C.11, we have that $\rho_k^\mu \sim \mathrm{Rayleigh}(\sqrt{\lambda_k N}/2)$, where $\lambda_k$ is the $k$-th eigenvalue of $\Sigma$ and that the phases are independent and $\varphi_k^\mu \sim \mathrm{Unif}([-\pi, \pi))$. Note that, since the inputs are real, we have $Z_{N-k}^\mu = \overline{Z}_k^\mu$. For any $\varepsilon > 0$, choose $k_0 \neq 0, N/2$ and define $X^\mu$ by

$$X^\mu = \begin{pmatrix} \rho_0^\mu \\ \vdots \\ \rho_{N-1}^\mu \end{pmatrix} \exp \begin{pmatrix} i\,\varphi_0^\mu \\ \vdots \\ i\,\psi_{k_0}^\mu \\ \vdots \\ i\,\varphi_{N-1}^\mu \end{pmatrix}, \qquad \psi_{k_0}^\mu = \varphi_{k_0}^\mu + \varepsilon\, \underbrace{f(\varphi_{k_0}^\mu)}_{\text{HOCs}} + U^\mu,$$

where $U^\mu$ is sampled from a discrete random variable, independent from $\varphi_k^\mu$, which takes values with equal probability in the angles corresponding to the 4th roots of unity, i.e. $\{0, \pi/2, \pi, 3\pi/2\}$ with probability $1/4$. In order to have real inputs in pixel space, we ask that $X_{N-k_0}^\mu = \overline{X}_{k_0}^\mu$. We call $\mathbb{P}$ the distribution of $x^\mu = \mathrm{IDFT}(X^\mu)$ in pixel space.

The next lemma clarifies why adding a simple spike independent from the random phase would have not broken the uniformity of the phase, leading to gaussian inputs. Then, the dependence of $f$ from the phase is crucial to guarantees that $\mathbb{P}$ has some non-trivial higher-order statistics and turns out to be, in fact, non-Gaussian.

**Lemma D.1.** *Denote by* $\mu = \mathrm{Unif}(\mathbb{T}^N)$ *the distribution of a random vector on the torus* $\mathbb{T} = \mathbb{R}/(2\pi\mathbb{Z})$. *Then, if* $\varphi \sim \mu$, $w^* \in \mathbb{R}^N$ *and* $\nu$ *is a scalar random variable on* $\mathbb{R}$ *following any distribution, we get that* $\varphi' = \varphi + \nu w^* \sim \mu$.

*Proof.* Consider $K$-measurable sets $A_k \subseteq \mathbb{T}$ and $A = \prod_{k=1}^K \in \mathbb{T}^N$. We have that

$$P(\varphi' \in A \mid \nu) = \prod_{k=1}^K P(\varphi_k + \nu v_k \in A_k \mid \nu) = \prod_{k=1}^K \underbrace{\mu(A_k - \nu v_k)}_{\mu(A_k)},$$

where we have noted that, being $\mu$ a Haar measure on the compact group $(\mathbb{T}^N, +)$, it is translational invariant on the torus. Then, by integrating out $\nu$, we conclude that

$$P(\varphi' \in A) = \prod_{k=1}^K \mu(A_k) = \mu(A).$$

$\square$

We compute now the statistics of the inputs sampled from the Fourier data model (3). For simplicity, we will often drop the dependence on the index for each sample $\mu$. By linearity of the DFT, it is clear that the new inputs have mean zero both in pixel and Fourier space. In Lemma D.2, we prove that the new inputs $x$ and the original inputs $z$ have the same covariance matrix both in pixel space and in Fourier space.

**Lemma D.2** (Second-order cumulants). *Let* $X \in \mathbb{C}^N$ *be a random vector distributed according to the Fourier data model* (3) *and consider* $x = \mathrm{IDFT}(X)$. *Then,* $\mathrm{Cov}[X] = \mathrm{Cov}[Z]$ *and* $\mathrm{Cov}[x] = \Sigma$.

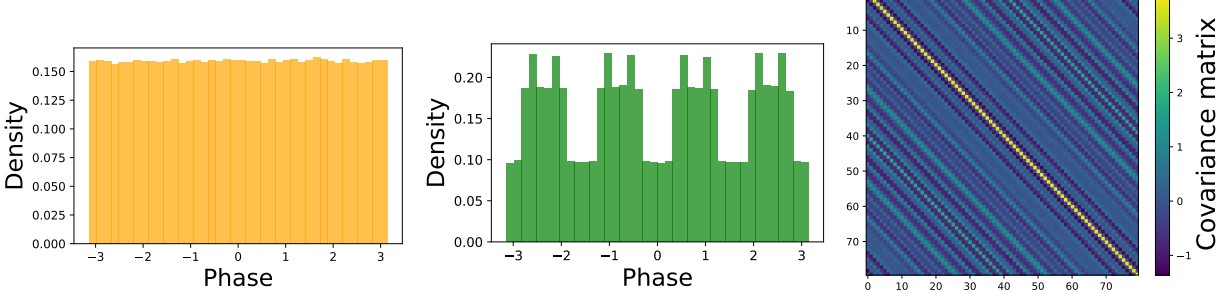

*Figure 8.* **Illustration of the statistics of inputs distributed according to the Fourier data model.** (**Left**) Uniform distribution of the phase of a non-modified frequency. (**Middle**) Spiked distribution of the phase of the modified frequency $k_0 = 6, \varepsilon = 2.5$. (**Right**) Example of a circulant covariance of both baseline $\mathbb{P}_0$ and planted $\mathbb{P}$ distributions, in pixel space.

*Proof.* Given that $x, z \in \mathbb{R}^N$ have mean zero, it holds that

$$\mathbb{E}[xx^\top] = F^{-1}\mathbb{E}[XX^*]F^{*-1} \quad \text{and} \quad \mathbb{E}[zz^\top] = \Sigma = F^{-1}\mathbb{E}[ZZ^*]F^{*-1}.$$

Because of that, it is sufficient to prove that $\mathbb{E}[XX^*] = \mathbb{E}[ZZ^*]$. Note that $U$ has been chosen such that

$$\mathbb{E}[e^{imU}] = \begin{cases} 1 & \text{if } m \equiv_4 0, \\ 0 & \text{otherwise.} \end{cases}$$

We can compute $\mathbb{E}[X_k X_l]$, for $k, l = 0, \ldots, N-1$. The entries for which $k, l \notin \{k_0, N-k_0\}$ are non-modified entries of the original inputs $Z$, and then they equal $\mathbb{E}[Z_k Z_l]$. Else, we have that

$$\mathbb{E}[X_k \overline{X}_l] = \begin{cases} \mathbb{E}[X_{k_0} \overline{X}_{k_0}] = \mathbb{E}[\rho_{k_0}^2] & \text{if } k = l = k_0, \\ \mathbb{E}[X_{N-k_0} \overline{X}_{N-k_0}] = \mathbb{E}[\overline{X}_{k_0} X_{k_0}] = \mathbb{E}[\rho_{k_0}^2] & \text{if } k = l = N - k_0, \\ \boldsymbol{\mathbb{E}[X_{k_0} \overline{X}_{N-k_0}] = \mathbb{E}[X_{k_0}^2] = \mathbb{E}[Z_{k_0}^2 e^{2i\varepsilon \sin \varphi_{k_0}}] \mathbb{E}[e^{i2U}] = 0} & \text{if } k = k_0, l = N - k_0, \\ \mathbb{E}[X_{k_0} X_l] = \mathbb{E}[X_{k_0}]\mathbb{E}[X_l] = 0 & \text{if } k = k_0, l \notin \{k_0, N - k_0\}. \end{cases}$$

In the case of the entry $(k_0, l)$, with $l \notin \{k_0, N - k_0\}$, we have used the independence coming from Gaussian translation-invariant inputs. Moreover, thanks to Lemma C.7, $\text{Cov}[Z]_{kl} = \mathbb{E}[\rho_k^2]\delta_{kl}$ and then we can conclude that $\mathbb{E}[XX^*] = \mathbb{E}[ZZ^*]$. Note that the choice of $U$ guarantees that $\text{Cov}[X]$ stays diagonal by dealing with the case $\boldsymbol{k = k_0, l = N - k_0}$, which instead would have broken translation invariance. $\qquad\square$

*Remark* D.3. Consider $x \in \mathbb{R}^N$ distributed according to $\mathbb{P}$ or $\mathbb{P}_0$. Recall that we denote by $F \in \mathbb{C}^{N \times N}$ the non-normalised DFT matrix and by $\widehat{F} = \frac{1}{\sqrt{N}}F \in \mathbb{C}^{N \times N}$ the normalised DFT matrix. Call $\lambda_k$ the $k$-th eigenvalue of the covariance matrix $\Sigma$ in pixel space. Then, if $X_F = Fx$ and $X_{\widehat{F}} = \widehat{F}x$, because of Lemma C.7, we get that

$$[\text{Cov}(X_F)]_{kl} = \lambda_k N \delta_{kl} \quad \text{and} \quad [\text{Cov}(X_{\widehat{F}})]_{kl} = \lambda_k \delta_{kl}.$$

Note that, in the case of the non-normalised DFT, the amplitudes $\rho_k^F$ are Rayleigh-distributed with parameter $\sigma_k^F = \sqrt{\frac{\lambda_k N}{2}}$. In the normalised case, we have $\sigma_k^{\widehat{F}} = \sqrt{\frac{\lambda_k}{2}}$. Then, thanks to Remark C.10 on the statistics of the Rayleigh distribution,

$$\mathbb{E}[(\rho_k^F)^2] = \lambda_k N \quad \text{and} \quad \mathbb{E}[(\rho_k^{\widehat{F}})^2] = \lambda_k.$$

Moreover, $\mathbb{E}[\rho_k^F] = \sqrt{\frac{\lambda_k \pi N}{4}}$ and $\mathbb{E}[\rho_k^{\widehat{F}}] = \sqrt{\frac{\lambda_k \pi}{4}}$.

The next definition is useful to write explicitly the non-trivial higher-order statistics of the inputs drawn from the Fourier data model in the case of $f(\varphi_{k_0}) = \sin(\varphi_{k_0})$.

**Definition D.4** (Bessel functions). The Bessel function of first kind of order $m$-th is $J_m(z)$ such that

$$\frac{1}{2\pi}\int_0^{2\pi} e^{i(m\varphi + z\sin\varphi)}\mathrm{d}\varphi = J_{-m}(z) = J_m(z)(-1)^m.$$

In the next lemma, we show that the third-order cumulants of the inputs drawn from the Fourier data model (3) are vanishing, and then they equal those of any Gaussian distribution $\mathbb{P}_0$.

**Lemma D.5** (Third-order cumulants). *Consider a random vector $X \in \mathbb{C}^N$ distributed according to the Fourier data model (3) and $x = \text{IDFT}(X)$. Then, for any $t_1, t_2, t_3 \in \{0, \ldots, N-1\}$, we have $\mathbb{E}[x_{t_1} x_{t_2} x_{t_3}] = 0$.*

*Proof.* Since

$$\mathbb{E}[x_{t_1} x_{t_2} x_{t_3}] = \frac{1}{N^3} \sum_{k,l,m=0}^{N-1} \mathbb{E}[X_k X_l X_m] e^{\frac{2\pi i}{N}(t_1 k + t_2 l + t_3 m)},$$

it is sufficient to show that $\mathbb{E}[X_k X_l X_m] = 0$, for any $k, l, m \in \{0, \ldots, N-1\}$. Note that $\mathbb{E}[X_k X_l X_m] = \mathbb{E}[Z_k Z_l Z_m] = 0$ if $k, l, m \notin \{k_0, N - k_0\}$, since $Z$ is Gaussian. The cases in which one or two entries are $X_{k_0}$ or $X_{N-k_0}$ vanish since each entry has mean zero and because of the independence of the entries. To conclude, $\mathbb{E}[X_{k_0}^3] = -J_3(3\varepsilon)\mathbb{E}[\rho_{k_0}^3]\mathbb{E}[e^{i3U}] = 0$ by construction of $U$. Same for $\mathbb{E}[X_{N-k_0}^3]$. $\qquad\square$

The next result, which allows to compute higher-order moments of a Gaussian distribution, will be used to compute the fourth-order statistics of inputs drawn from the Fourier data model (3). It basically tells that any higher-order moment of a zero-mean Gaussian random vector can be written in terms of its covariance matrix.

**Lemma D.6** (Isserlis' theorem (Isserlis, 1918)). *If $x \in \mathbb{R}^N$ is a zero-mean Gaussian random vector, we have*

$$\mathbb{E}[x_1 x_2 \cdots x_k] = \sum_{\text{all pairings of } \{1,\ldots,k\}} \prod_{(a,b) \in \text{ pairing}} \mathbb{E}[x_a x_b].$$

Isserlis' theorem implies that, given $x \sim \mathcal{N}(0, \Sigma), \Sigma \in \mathbb{R}^{N \times N}$ and $t_1, t_2, t_3, t_4 \in \{0, \ldots, N-1\}$, we can write

$$\mathbb{E}[x_{t_1} x_{t_2} x_{t_3} x_{t_4}] = \Sigma_{t_1 t_2} \Sigma_{t_3 t_4} + \Sigma_{t_1 t_3} \Sigma_{t_2 t_4} + \Sigma_{t_1 t_4} \Sigma_{t_2 t_3}.$$

We now compute the fourth-order moments of inputs sampled from the Fourier data model (3). We prove that the fourth-order moments of these inputs projected along a weight vector of unit norm coincide with the ones of a zero-mean Gaussian random vector with covariance $\Sigma$. However, these projections differ in distribution from the Gaussian ones if the weight belongs to the subspace spanned by the DFT phase vectors $u$ and $v$. This is the effect of the phase modification of the mode $k_0$, which results in an additional signal at the level of higher-order interactions for the new data points. Note that choosing $U$ such that $\mathbb{E}[e^{i4U}] = 1$ induces the non-vanishing higher-order correlations to be of order four.

**Lemma D.7** (Fourth-order cumulants). *Consider a random vector $X \in \mathbb{C}^N$ distributed according to the Fourier data model (3) and $x = \text{IDFT}(X)$. Then, for any $t_1, t_2, t_3, t_4 \in \{0, \ldots, N-1\}$, we have that*

$$\mathbb{E}[x_{t_1} x_{t_2} x_{t_3} x_{t_4}] = \Sigma_{t_1 t_2} \Sigma_{t_3 t_4} + \Sigma_{t_1 t_3} \Sigma_{t_2 t_4} + \Sigma_{t_1 t_4} \Sigma_{t_2 t_3} + \frac{2}{N^4} J_4(4\varepsilon)\mathbb{E}[\rho_{k_0}^4] \cos\left(\frac{2\pi k_0}{N}(t_1 + t_2 + t_3 + t_4)\right).$$

*Moreover, for any $w \in \mathbb{R}^N$, it holds that*

$$\mathbb{E}[(w \cdot x)^4] = 3(w^\top \Sigma w)^2 + \frac{2J_4(4\varepsilon)}{N^2} \mathbb{E}[\rho_{k_0}^4] \text{Re}(w_{k_0}^4),$$

*where $\text{Re}(w_{k_0})$ is the real part of*

$$w_{k_0} = \frac{1}{\sqrt{N}} \sum_{t=0}^{N-1} w_t \exp(2\pi i k_0 t/N).$$

*Assume now that $\|w\| = 1$. Consider the case where $\theta \in (0, 2\pi]$ is such that $w = \cos\theta\, u + \sin\theta\, v$. Then, $\text{Re}(w_{k_0}^4) = \cos(4\theta)/4$. Conversely, if $w \cdot u = w \cdot v = 0$, we get that $\text{Re}(w_{k_0}^4) = 0$.*

*Proof.* For any $t_1, t_2, t_3, t_4 \in \{0, \ldots, N-1\}$, we are going to compute

$$\mathbb{E}[x_{t_1} x_{t_2} x_{t_3} x_{t_4}] = \frac{1}{N^4} \sum_{k,l,m,n=0}^{N-1} \mathbb{E}[X_k X_l X_m X_n] e^{\frac{2\pi i}{N}(kt_1 + lt_2 + mt_3 + nt_4)}.$$

We first show that this fourth moment can be written as sum of two contributions: one term containing the fourth moments of Gaussian variables and a second term which involves the modified frequencies, i.e.

$$\mathbb{E}[x_{t_1} x_{t_2} x_{t_3} x_{t_4}] = \frac{1}{N^4} \sum_{k,l,m,n=0}^{N-1} \mathbb{E}[Z_k Z_l Z_m Z_n] \, e^{\frac{2\pi i}{N}(kt_1 + lt_2 + mt_3 + nt_4)} \tag{15}$$

$$+ \frac{1}{N^4} \sum_{\substack{k=l=m=n=k_0, \\ k=l=m=n=N-k_0}}^{N-1} \mathbb{E}[X_k X_l X_m X_n] \, e^{\frac{2\pi i}{N}(kt_1 + lt_2 + mt_3 + nt_4)}. \tag{16}$$

By construction, we know that when $k,l,m,m \notin \{k_0, N - k_0\}$ we have $\mathbb{E}[X_k X_l X_m X_n] = \mathbb{E}[Z_k Z_l Z_m Z_n]$. Also, because of the independence of the entries, we have that $\mathbb{E}[X_{k_0} X_k X_l X_m] = \mathbb{E}[X_{k_0}^2 X_k X_l] = \mathbb{E}[X_{k_0}^3 X_k] = 0$, for $k,l,m \notin \{k_0, N - k_0\}, k \neq l \neq m$, nor their conjugates. What's more, $\mathbb{E}[X_{k_0}^2 X_k^2] = \mathbb{E}[\rho_{k_0}^2 X_k^2 e^{2i\varepsilon \sin \varphi_{k_0}}] \mathbb{E}[e^{i2U}] = 0$, for $k \notin \{k_0, N - k_0\}$. Also, $\mathbb{E}[X_{k_0} X_k^3] = 0$ since $X_{k_0}$ has zero mean. The non-vanishing contributions are

$$\mathbb{E}[X_k X_l X_m X_n] = \begin{cases} \mathbb{E}[X_{k_0} X_{N-k_0} X_m X_{N-m}] = \mathbb{E}[\rho_{k_0}^2] \mathbb{E}[\rho_m^2] & \text{if } k = k_0, l = N - k_0, m \neq \{k, N-k\}, n = N - m, \\ \mathbb{E}[X_{k_0}^2 X_{N-k_0}^2] = \mathbb{E}[X_{k_0}^2 \overline{X}_{k_0}^2] = \mathbb{E}[\rho_{k_0}^4] & \text{if } k = l = k_0, m = n = N - k_0, \\ \mathbb{E}[X_{N-k_0}^4] = \mathbb{E}[\overline{X}_{k_0}^4] = J_4(4\varepsilon) \mathbb{E}[\rho_{k_0}^4] & \text{if } k = l = m = n = N - k_0, \\ \mathbb{E}[X_{k_0}^4] = \mathbb{E}[X_{N-k_0}^4] = J_4(4\varepsilon) \mathbb{E}[\rho_{k_0}^4] & \text{if } k = l = m = n = k_0. \end{cases}$$

Note that the last two cases don't vanish because, by construction of $U$, we have that $E[e^{i4U}] = 1$. We know that $a = \text{Re}(Z_k) \sim \mathcal{N}(0, \mathbb{E}[\rho_k^2]/2)$ and $b = \text{Im}(Z_k) \sim \mathcal{N}(0, \mathbb{E}[\rho_k^2]/2)$ for $k \neq 0, N/2$. Since $\rho_{k_0}^2 = a^2 + b^2$, it follows that

$$\mathbb{E}[\rho_{k_0}^4] = \mathbb{E}[a^4] + \mathbb{E}[b^2] + 2\mathbb{E}[a^2]\mathbb{E}[b^2] = 3\frac{\mathbb{E}[\rho_{k_0}^2]^2}{2} + \frac{\mathbb{E}[\rho_{k_0}^2]^2}{2} = 2\mathbb{E}[\rho_{k_0}^2]^2.$$

Now, recall that thanks to the Isserlis' Theorem D.6 for Gaussian random variables we have

$$\mathbb{E}[Z_k Z_l Z_m Z_n] = \mathbb{E}[\rho_k^2]\mathbb{E}[\rho_m^2](\delta_{k,N-l}\delta_{m,N-n} + \delta_{k,N-m}\delta_{l,N-n} + \delta_{k,N-n}\delta_{l,N-m}),$$

which is non vanishing if and only if the indices can be partitioned in two pairs of conjugates. Then, $\mathbb{E}[X_k X_l X_m X_n] = \mathbb{E}[Z_k Z_l Z_m Z_n]$ for any $(k,l,m,m) \neq (k_0, k_0, k_0, k_0)$ and $(k,l,m,m) \neq (N - k_0, N - k_0, N - k_0, N - k_0)$, and Equation (15) follows. The first addendum of Equation (15), by definition of DFT, is simply equal to $\mathbb{E}[z_{t_1} z_{t_2} z_{t_3} z_{t_4}]$. Now, thanks again to the Isserlis' theorem for $z$, we have that $\mathbb{E}[z_{t_1} z_{t_2} z_{t_3} z_{t_4}] = \Sigma_{t_1 t_2} \Sigma_{t_3 t_4} + \Sigma_{t_1 t_3} \Sigma_{t_2 t_4} + \Sigma_{t_1 t_4} \Sigma_{t_2 t_3}$. It remains to compute the second addendum in Equation (15), which turns out to be

$$\frac{J_4(4\varepsilon)\mathbb{E}[\rho_{k_0}^4]}{N^4}\left[e^{\frac{2\pi i k_0}{N}(t_1+t_2+t_3+t_4)} + e^{\frac{2\pi i(N-k_0)}{N}(t_1+t_2+t_3+t_4)}\right] = \frac{2J_4(4\varepsilon)\mathbb{E}[\rho_{k_0}^4]}{N^4}\cos\left(\frac{2\pi k_0(t_1+t_2+t_3+t_4)}{N}\right).$$

For the second part of the thesis, consider $w \in \mathbb{R}^N$ and hence

$$\mathbb{E}[(w \cdot x)^4] = \sum_{k,l,m,n=0}^{N-1} w_k w_l w_m w_n \mathbb{E}[x_k x_l x_m x_n]$$

$$= 3(w^\top \Sigma w)^2 + \frac{2J_4(4\varepsilon)\mathbb{E}[\rho_{k_0}^4]}{N^4} \sum_{k,l,m,n=0}^{N-1} w_k w_l w_m w_n \cos\left(\frac{2\pi k_0(t_1 + t_2 + t_3 + t_4)}{N}\right).$$

Then, using the fact that $\cos \alpha = (e^{i\alpha} + e^{-i\alpha})/2$, we can write

$$2 \sum_{k,l,m,n=0}^{N-1} w_k w_l w_m w_n \cos\left(\frac{2\pi k_0(t_1+t_2+t_3+t_4)}{N}\right) = \sum_{k,l,m,n=0}^{N-1} w_k w_l w_m w_n \left(e^{\frac{2\pi i k_0}{N}(k+l+m+n)} + e^{-\frac{2\pi i k_0}{N}(k+l+m+n)}\right).$$

Since

$$\sum_{k,l,m,n=0}^{N-1} w_k w_l w_m w_n e^{\frac{2\pi i k_0}{N}(k+l+m+n)} = \underbrace{\left(\sum_{k=0}^{N-1} w_k e^{\frac{2\pi i k_0 k}{N}}\right)^4}_{w_{k_0}(\sqrt{N})}.$$

By defining $w_{k_0} = 1/\sqrt{N} \sum_{k=0}^{N-1} w_k e^{\frac{2\pi i k_0 k}{N}}$, we can conclude that

$$\mathbb{E}[(w \cdot x)^4] = 3(w^\top \Sigma w)^2 + \frac{J_4(4\varepsilon)\mathbb{E}[\rho_{k_0}^4]}{N^2}(w_{k_0}^4 + \overline{w}_{k_0}^4) = 3(w^\top \Sigma w)^2 + \frac{2J_4(4\varepsilon)\mathbb{E}[\rho_{k_0}^4]}{N^2}\mathbb{E}[\rho_{k_0}^4]\mathrm{Re}(w_{k_0}^4).$$

What's more, consider $w$ such that $\|w\| = 1$ and $\theta \in (0, 2\pi]$ with $w = \cos\theta\, u + \sin\theta\, v$, for $u, v$ the DFT phase vectors corresponding to the frequency $k_0$. Then, $w_{k_0}^4 = (\frac{w}{\sqrt{2}}(u + iv))^4 = \frac{1}{4}(\cos\theta + i\sin\theta)^4$ and $\mathrm{Re}(w_{k_0}^4) = \frac{1}{4}\cos 4\theta$. On the other hand, consider $w \cdot u = w \cdot v = 0$. Since

$$\begin{aligned}
w_{k_0} &= \frac{1}{\sqrt{N}} \sum_{t=0}^{N-1} w_t \exp\left(-\frac{2\pi i k_0 t}{N}\right) = \frac{1}{\sqrt{N}} \sum_{t=0}^{N-1} w_t \left[\cos\left(\frac{2\pi k_0 t}{N}\right) - i\sin\left(\frac{2\pi k_0 t}{N}\right)\right] \\
&= \frac{1}{\sqrt{N}}\left[\sum_{t=0}^{N-1} w_t\left(\sqrt{\frac{N}{2}}u_t\right) - i\sum_{t=0}^{N-1} w_t\left(\sqrt{\frac{N}{2}}v_t\right)\right] = \frac{1}{\sqrt{2}}\left(\sum_{t=0}^{N-1} w_t u_t - i\sum_{t=0}^{N-1} w_t v_t\right) \\
&= \frac{1}{\sqrt{2}}\left((w \cdot u) - i(w \cdot v)\right) = 0,
\end{aligned}$$

we have the thesis. $\qquad\square$

In the next lemma, we compute the expression of the inputs sampled from the Fourier data model in pixel space.

**Lemma D.8.** *In pixel space, the inputs* $(x^\mu)_\mu^n \subseteq \mathbb{R}^N$ *sampled from the Fourier data model* (3) *cannot be seen as distributed according to a single-index model in the usual sense (e.g. Equation* (2)). *Instead, for* $k = 0, \ldots, N-1$, *we have that*

$$x_k^\mu = z_k^\mu + \frac{2\rho_{k_0}^\mu}{N}\left[\cos\left(\frac{2\pi k k_0}{N} + \varphi_{k_0}^\mu + \varepsilon f(\varphi_{k_0}^\mu) + U^\mu\right)\right].$$

*Proof.* We know that $X_s = Z_s$ if $s \neq k_0, N - k_0$, and additionally $X_{k_0} = \rho_{k_0}e^{i(\varphi_{k_0} + \varepsilon\sin\varphi_{k_0} + U)}$ and $X_{N-k_0} = \bar{X}_{k_0}$. We get that

$$x_k = \frac{1}{N}\sum_{s=0}^{N-1} X_s e^{2\pi i s k/N} = \frac{1}{N}\sum_{s \neq k_0, N-k_0} Z_s e^{2\pi i s k/N} + \frac{1}{N}\left(X_{k_0}e^{2\pi i n k_0/N} + X_{N-k_0}e^{2\pi i(N-k_0)/N}\right).$$

Since

$$\frac{1}{N}\sum_{s \neq k_0, N-k_0} Z_s e^{2\pi i s k/N} = z_k - \frac{1}{N}\left(Z_{k_0}e^{2\pi i s k_0/N} + Z_{N-k_0}e^{2\pi i s(N-k_0)/N}\right),$$

the thesis follows. $\qquad\square$

*Remark* D.9. Note that the latent variables $\varphi_{k_0}^\mu$ and $U^\mu$ always enter through a non-linear function (the cosine). This is due to the modulation of the phases of the inputs in Fourier space, which are linked non-linearly to the inputs in pixel space.

## E. Hardness of learning the phase

In this section we prove that online SGD requires a long time - equivalently, a large number of samples - to distinguish inputs drawn from Gaussian white noise, $\mathbb{P}_0 = \mathcal{N}(0, \mathbb{1})$, and isotropic inputs drawn from the Fourier data model (3). In this setting, both distributions have identity covariance matrix, which implies that SGD cannot exploit information coming from low-order statistics of the data to perform classification.

We start by computing the coefficients $c_{ij}^\ell$ defined in Equation (17). We will show that they coincide with the coefficients of the Hermite expansion of the likelihood ratio between the distributions $\mathbb{P}$ and $\mathbb{P}_0$. These coefficients determine the low-order terms in the expansion of the population classification loss (6).

**Lemma E.1** (Coefficients of the likelihood ratio). *Consider the following coefficients, with expectation of the inputs $x$ taken with respect to $\mathbb{P}$:*

$$c_{ij}^\ell = \mathbb{E}_\mathbb{P}\left[h_i\left(\frac{v \cdot x}{\sigma_C}\right)h_j\left(\frac{u \cdot x}{\sigma_B}\right)\right], \tag{17}$$

*where $h_i$, $h_j$ are the $i$-th and $j$-th probabilist Hermite polynomials, for $i + j \leq 4$. We set $\sigma_C = \sqrt{v^\top \Sigma v}$ and $\sigma_B = \sqrt{u^\top \Sigma u}$. Then, we have that $c_{ij}^\ell = 0$ when $i + j < 4$. Moreover, $c_{13}^\ell = c_{31}^\ell = 0$,*

$$c_{22}^\ell = -\frac{1}{2} c_{40}^\ell \qquad and \qquad c_{40}^\ell = c_{04}^\ell = \frac{J_4(4\varepsilon)}{\lambda_{k_0}^2 N^2} \mathbb{E}[\rho_{k_0}^4].$$

*Proof.* Recall that the DFT vectors $u$ and $v$ are eigenvectors of $\Sigma$ for the same eigenvalue $\lambda_{k_0}$. Then, we obtain that $\sigma_C = \sigma_B = \sqrt{\lambda_{k_0}}$.

- Cases $i + j = 4$. For $i = 4, j = 0$, we have that

$$c_{40}^\ell = \mathbb{E}\left[ h_4\left( \frac{v \cdot x}{\sigma_C} \right)^4 \right] = \mathbb{E}\left[ \frac{(v \cdot x)^4}{\sigma_C^4} \right] - 6\,\mathbb{E}\left[ \frac{(v \cdot x)^2}{\sigma_C^2} \right] + 3.$$

Since $\mathbb{E}[(v \cdot x)^2] = \sigma_C^2$ and thanks to Lemma D.7, we can choose $\theta = \pi/2$ and get

$$c_{40}^\ell = \frac{J_4(4\varepsilon)}{\lambda_{k_0}^2 N^2} \mathbb{E}[\rho_{k_0}^4].$$

For $i = 0, j = 4$, we choose $\theta = 0$ and by the same reasoning as before we get $c_{04}^\ell = c_{40}^\ell$. Similarly,

$$c_{22}^\ell = \mathbb{E}\left[ h_2\left( \frac{v \cdot x}{\sigma_C} \right) h_2\left( \frac{u \cdot x}{\sigma_B} \right) \right] = \frac{1}{\lambda_{k_0}^2} \mathbb{E}[(v \cdot x)^2 (u \cdot x)^2] - \frac{1}{\lambda_{k_0}} \left[ \mathbb{E}[(v \cdot x)^2] + \mathbb{E}[(u \cdot x)^2] \right] + 1$$

$$= \frac{1}{\lambda_{k_0}^2} \mathbb{E}[(v \cdot x)^2 (u \cdot x)^2] - 1.$$

By exploiting the orthonormality of $u$ and $v$ and Lemma D.7, we have

$$\mathbb{E}[(v \cdot x)^2 (u \cdot x)^2] = \sum_{k,l,m,n=0}^{N-1} u_k u_l v_m v_n \mathbb{E}[x_k x_l x_m x_n] = \lambda_{k_0}^2 + T_4,$$

where

$$T_4 = \frac{2}{N^4} J_4(4\varepsilon) \mathbb{E}[\rho_{k_0}^4] \sum_{k,l,m,n=0}^{N-1} u_k u_l v_m v_n \cos \frac{2\pi k_0}{N}(k + l + n + m).$$

Define now the angle $\theta = 2\pi k_0/N$, and note that

$$\cos \theta r = \frac{e^{i\theta r} + e^{-i\theta r}}{2}, \quad \sin \theta r = \frac{e^{i\theta r} - e^{-i\theta r}}{2i}.$$

Then, exploiting the definition of $u$ and $v$, we can write $T_4$ as

$$T_4 = -\frac{2J_4(4\varepsilon)\mathbb{E}[\rho_{k_0}^4]}{32 N^4} \sum_{\substack{s_1, s_2 = \pm 1 \\ r_1, r_2 = \pm 1 \\ t = \pm 1}} r_1 r_2 \left( \sum_{k=0}^{N-1} e^{i\theta(s_1+t)k} \right) \left( \sum_{l=0}^{N-1} e^{i\theta(s_2+t)l} \right) \left( \sum_{n=0}^{N-1} e^{i\theta(r_1+t)n} \right) \left( \sum_{m=0}^{N-1} e^{i\theta(r_2+t)m} \right) \frac{4}{N^2}.$$

Since, for any $q \in \mathbb{Z}$, we have that

$$\sum_{q'=0}^{N-1} e^{i\theta q q'} = \begin{cases} N & \text{if } k_0 q \equiv_N 0, \\ 0 & \text{otherwise}, \end{cases}$$

the factors in $T_4$ are non vanishing if and only if $s_1 + t = s_2 + t = r_1 + t = r_2 + t = 0$, corresponding to the configurations

$$(s_1, s_2, r_1, r_2, t) = (1, 1, 1, 1, -1) \text{ and } (s_1, s_2, r_1, r_2, t) = (-1, -1, -1, -1, 1).$$

Each of the two non vanishing factors produces a contributions of $N^2$. Hence, we conclude that $T_4 = -J_4(4\varepsilon)\mathbb{E}[\rho_{k_0}^4]/(2N^2)$ and then $c_{22}^\ell = T_4/\lambda_{k_0}^2 = -J_4(4\varepsilon)\mathbb{E}[\rho_{k_0}^4]/(2N^2\lambda_{k_0}^2)$. With a similar computation, we can prove that $c_{31}^\ell = c_{13}^\ell = 0$. Indeed, we have

$$c_{31}^\ell = \frac{1}{\lambda_{k_0}^2}\mathbb{E}[(v \cdot x)^3(u \cdot x)] = 3(u^\top \Sigma u)(u^\top \Sigma v) + T_4',$$

where

$$T_4' = \frac{2}{N^4}J_4(4\varepsilon)\mathbb{E}[\rho_{k_0}^4]\sum_{k,l,m,m=0}^{N-1} u_k u_l u_m v_n \cos\frac{2\pi k_0}{N}(k+l+n+m)$$

$$= \frac{J_4(4\varepsilon)\mathbb{E}[\rho_{k_0}^4]}{2N^6}\sum_{\substack{s_1,s_2,s_3=\pm1 \\ r=\pm1, t=\pm1}} r\left(\sum_{k=0}^{N-1} e^{i\theta(s_1+t)k}\right)\left(\sum_{l=0}^{N-1} e^{i\theta(s_2+t)l}\right)\left(\sum_{m=0}^{N-1} e^{i\theta(s_3+t)m}\right)\left(\sum_{n=0}^{N-1} e^{i\theta(r+t)n}\right).$$

Since the only two non vanishing configurations are $(s_1, s_2, s_3, r, t) = (1, 1, 1, 1, -1)$ and $(s_1, s_2, s_3, r, t) = (-1, -1, -1, -1, 1)$, and each one contributes with $N^2$, it turns out that $T_4' = 0$. By recalling that $u$ and $v$ are orthogonal, we conclude that $c_{31}^\ell = 0$.

- Cases $i + j = 3$, $i + j = 1$ or $i, j = 0$. Because of Lemma D.5, it is clear that $c_{30}^\ell = c_{03}^\ell = c_{21}^\ell = c_{12}^\ell = 0$. Since the inputs have zero mean, $c_{10}^\ell = c_{01}^\ell = 0$. Clearly, the null term is $c_{00}^\ell = 1$.

- Cases $i + j = 2$. Since $\Sigma v = \lambda_{k_0} v$ and $\Sigma u = \lambda_{k_0} u$, we have that

$$c_{20}^\ell = \frac{1}{\sigma_C^2}\mathbb{E}[(v \cdot x)^2] - 1 = \frac{1}{\lambda_{k_0}}v^\top \Sigma v - 1 = 0,$$

$$c_{02}^\ell = \frac{1}{\sigma_B^2}\mathbb{E}[(u \cdot x)^2] - 1 = \frac{1}{\lambda_{k_0}}u^\top \Sigma v - 1 = 0,$$

$$c_{11}^\ell = \mathbb{E}\left[\left(\frac{v \cdot x}{\sigma_C}\right)\left(\frac{u \cdot x}{\sigma_B}\right)\right] = \frac{1}{\lambda_{k_0}}v^\top \Sigma u = 0.$$

$\square$

We now prove the main theorem of this section (Theorem 3.1 in the main text), which provides the sample complexities to recover the phase information in case of isotropic inputs.

*Theorem* 3. Sample $(x^\mu)_{\mu=1}^n \subseteq \mathbb{R}^N$ from the Fourier data model (3) with $\Sigma = \mathbb{1}$. Apply online SGD (5) to the correlation loss (1) and define the overlaps $\alpha_{u,n} = w_n \cdot u$ and $\alpha_{v,n} = w_n \cdot v$, where $(u, v)$ are the DFT phase vectors. Then, if $1/(N^2 \log^2 N) \ll \delta_N \ll 1/(N \log N)$ and $n \gg N^3 \log^2 N$, there is **weak recovery** for $(u, v)$ i.e., for some $\eta > 0$,

$$\lim_{N\to\infty} P(|\alpha_{u,n}| \geq \eta) = 1 \quad \text{and} \quad \lim_{N\to\infty} P(|\alpha_{v,n}| \geq \eta) = 1.$$

Conversely, when $\delta_N \ll 1/(N \log N)$ and $n \ll N^3$, there is **no recovery** for $(u, v)$ i.e., in probability,

$$|\alpha_{u,n}|, |\alpha_{v,n}| \xrightarrow[N\to+\infty]{} 0.$$

*Proof.* By taking the expectation in the correlation loss (1) with respect to the data distribution, the population correlation loss reads as

$$L(w) = 1 - \frac{1}{2}\mathbb{E}_\mathbb{P}[\sigma(w \cdot x)] + \frac{1}{2}\mathbb{E}_{\mathbb{P}_0}[\sigma(w \cdot z)]. \tag{18}$$

Since $w$ has unitary norm, $w \cdot z \sim \mathcal{N}(0, 1)$ and then $\mathbb{E}_{\mathbb{P}_0}[\sigma(w \cdot x)] = c_0^\sigma$, where $c_0^\sigma$ is the zeroth - order Hermite coefficient for the activation function $\sigma$. Moreover, note that the likelihood ratio $\ell = d\mathbb{P} \backslash d\mathbb{P}_0$ depends only on the projections $u \cdot x$ and $v \cdot x$, by construction of the Fourier data model. Indeed, the inputs sampled from the baseline distribution $\mathbb{P}_0$ are Gaussian distributed and translation-invariant, which implies that their (orthogonal) DFT coefficients $X_k$ are independent. Then, to

obtain the new inputs of the Fourier data model, we perform the modification of the phase of the mode $k_0$. Hence, if the densities in Fourier space are $p$ and $p_0$, we get the following factorization:

$$\ell(x) = \frac{d\mathbb{P}}{d\mathbb{P}_0}(x) = \frac{dp}{dp_0}(X) = \frac{dp(X_{k_0}, X_{N-k_0})}{dp_0(X_{k_0}, X_{N-k_0})} \prod_{k \neq k_0, N-k_0} \frac{dp(X_k, X_{N-k})}{dp_0(X_k, X_{N-k})} = \frac{dp(X_{k_0}, X_{N-k_0})}{dp_0(X_{k_0}, X_{N-k_0})}.$$

Also, we have that $X_{k_0} = \sum_{n=0}^{N-1} x_n/\sqrt{N}\, e^{-2\pi k_0 n/N} = \frac{1}{\sqrt{2}}(x \cdot u - i\, x \cdot v)$, where $u$ and $v$ are the DFT phase vectors. Since $X_{N-k_0} = \overline{X}_{k_0}$, the likelihood ratio $\ell$ depends only on the projections of the inputs along the DFT phase vectors. In what follows, with a slight abuse of notation, we will write $\ell = \ell(x) = \ell(x \cdot v, x \cdot u)$. Therefore, we have that

$$\mathbb{E}_{\mathbb{P}}[\sigma(w \cdot x)] = \mathbb{E}_{\mathbb{P}_0}[\sigma(w \cdot x)\ell(v \cdot x, u \cdot x)]. \tag{19}$$

We can now expand $\ell$ and $\sigma$ in Hermite polynomials and get that

$$\ell(v \cdot x, u \cdot x) = \sum_{i,j=0}^{+\infty} \frac{c_{ij}^\ell}{i!j!} h_i(v \cdot x) h_j(u \cdot x) \quad \text{and} \quad \sigma(w \cdot x) = \sum_{k=0}^{+\infty} \frac{c_k^\sigma}{k!} h_k(w \cdot x),$$

with coefficients

$$c_{ij}^\ell = \mathbb{E}_{\mathcal{N}(0,\mathbb{1}_2)}[\ell(Z_1, Z_2) h_i(Z_1) h_j(Z_2)] \quad \text{and} \quad c_k^\sigma = \mathbb{E}[\sigma(Z) h_k(Z)].$$

We now want to write $c_{ij}^\ell$ in a different form, such that it is possible to explicitly compute them. To do so, define the projection map $T : \mathbb{R}^N \to \mathbb{R}^2$ such as $T(x) = (Z_1, Z_2) = [(x \cdot v)/\sigma_C, (x \cdot u)/\sigma_B]$. Under $\mathbb{P}_0$, we get that $T(x) \sim \mathcal{N}(0, \mathbb{1}_2)$, since $u$ and $v$ are orthonormal. Then, $T_\# \mathbb{P}_0 = \mathcal{N}(0, \mathbb{1}_2)$ and

$$c_{ij}^\ell = \mathbb{E}_{\mathcal{N}(0,\mathbb{1}_2)}[\ell(Z_1, Z_2) h_i(Z_1) h_j(Z_2)] = \mathbb{E}_{T_\# \mathbb{P}_0}[\ell(Z_1, Z_2) h_i(Z_1) h_j(Z_2)]$$
$$= \mathbb{E}_{\mathbb{P}_0}\left[\ell(T(x))\, h_i\left(\frac{x \cdot u}{\sigma_C}\right) h_j\left(\frac{x \cdot u}{\sigma_B}\right)\right] = \mathbb{E}_{\mathbb{P}}\left[h_i\left(\frac{x \cdot v}{\sigma_C}\right) h_j\left(\frac{x \cdot u}{\sigma_B}\right)\right].$$

Note that the coefficients $c_{i,j}^\ell$ for $i + j \leq 4$ have been computed in Lemma E.1. From Equation (19), it holds that

$$\mathbb{E}_{\mathbb{P}}[\sigma(w \cdot x)] = \sum_{i,j,k=0}^{+\infty} \frac{c_{ij}^\ell c_k^\sigma}{i!j!k!} \mathbb{E}_{\mathbb{P}_0}[h_i(v \cdot x) h_j(u \cdot x) h_k(w \cdot x)]. \tag{20}$$

We can now decompose $w$ such that $w = \alpha_v v + \alpha_u u + w_\perp$, where $\alpha_v = v \cdot w$ and $\alpha_u = u \cdot w$. Recall that for independent normal Gaussian variables $Z_1, Z_2$ and coefficients $a, b > 0$ it holds that

$$h_k(aZ_1 + bZ_2) = \sum_{s=0}^{k} \binom{k}{s} a^s b^{k-s} h_s(Z_1) h_{k-s}(Z_2), \tag{21}$$

for any order $k \in \mathbb{N}$. Since the projections of the inputs along $u, v$ and $w_\perp$ are independent, we can apply formula Equation (21) twice and we get that

$$h_k(w \cdot x) = \sum_{a+b+c=k} \frac{k!}{a!b!c!} \alpha_v^a \alpha_u^b h_a(x \cdot v) h_b(x \cdot u) h_c(x \cdot w_\perp).$$

By plugging $h_k(w \cdot x)$ into Equation (20), we finally obtain that

$$\mathbb{E}_{\mathbb{P}}[\sigma(w \cdot x)] = \sum_{i,j,k=0}^{+\infty} \sum_{a+b+c=k} \frac{c_{ij}^\ell c_k^\sigma k!}{i!j!k!a!b!c!} \underbrace{\mathbb{E}_{\mathbb{P}_0}[h_i(v \cdot x) h_a(v \cdot x)]}_{i!\delta_{ia}} \underbrace{\mathbb{E}_{\mathbb{P}_0}[h_j(u \cdot x) h_b(u \cdot x)]}_{j!\delta_{jb}} \underbrace{\mathbb{E}_{\mathbb{P}_0}[h_c(w_\perp \cdot x)]}_{\delta_{c0}}$$
$$= \sum_{i+j=k}^{+\infty} \frac{c_{ij}^\ell c_k^\sigma}{i!j!} \alpha_v^i \alpha_u^j = \sum_{i,j=0}^{+\infty} \frac{c_{ij}^\ell c_{i+j}^\sigma}{i!j!} \alpha_v^i \alpha_u^j,$$

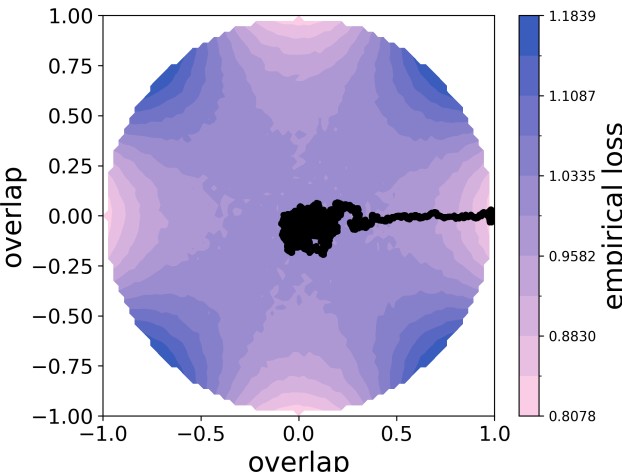

*Figure 9.* **Level sets of the empirical correlation loss.** Consider the empirical correlation loss (1) with inputs sampled from the Fourier data model (3) and $\Sigma = \mathbb{1}$. Here, the activation function $\sigma$ is the 4-th Hermite polynomial $\sigma(s) = h_4(s) = s^4 - 6s^2 + 3$. We plot the level sets of this loss, while its 3D version is shown in Figure 2 (right). As before, one run of SGD (•) is displayed: SGD wanders for a long time in the search phase before recovering the non-Gaussian signal. On the $x$- and $y$-axes we have the overlaps $w \cdot u$ and $w \cdot v$ of the weight vector $w$ with the DFT phase vectors $u$ and $v$. Note that the minima of the loss landscape are obtained at $w = \cos\theta\, u + \sin\theta\, v$ for the angles $\theta = 0, \pi/2, \pi, 3/2\pi$, which are the values assumed by the discrete latent variable $U$ in the Fourier data model. Note that we are demonstrating experimentally that, beyond the weak recovery proved in Theorem 3.1, one of the DFT vectors is recovered strongly.

where we have used the orthogonality property of the Hermite polynomials (Lemma C.14). Then, thanks to Lemma E.1 (note that here $\lambda_{k_0} = 1$) we are able to compute the coefficients of the likelihood ratio $c_{ij}^\ell$. We get that

$$L(w) = 1 - \frac{1}{2}\left[ c_1^\sigma \overbrace{(c_{10}^\ell \alpha_v + c_{01}^\ell \alpha_u)}^{0} + c_2^\sigma \overbrace{(c_{20}^\ell \alpha_v^2/2 + c_{11}^\ell \alpha_v \alpha_u + c_{02}^\ell \alpha_u^2/2)}^{0} \right.$$

$$+ c_3^\sigma \overbrace{(c_{30}^\ell \alpha_v^3/3! + c_{21}^\ell \alpha_v^2 \alpha_u/2 + c_{12}^\ell \alpha_v \alpha_u^2/2 + c_{03}^\ell \alpha_u^3/3!)}^{0}$$

$$+ c_4^\sigma \underbrace{(c_{40}^\ell \alpha_v^4/4! + c_{31}^\ell \alpha_v^3 \alpha_u/3! + c_{22}^\ell \alpha_v^2 \alpha_u^2/4 + c_{13}^\ell \alpha_v \alpha_u^3/3! + c_{04} \alpha_u^4/4!)}_{J_4(4\varepsilon)(\alpha_v^4 + \alpha_u^4)/4! - \frac{J_4(4\varepsilon)}{2}\alpha_v^2\alpha_u^2/4} + \text{others} \Bigg]$$

$$= 1 - \frac{c_4^\sigma J_4(4\varepsilon)}{48}(\alpha_v^4 + \alpha_u^4) + c_4^\sigma \frac{J_4(4\varepsilon)}{16}\alpha_v^2\alpha_u^2 + \text{others}.$$

Since the first non-vanishing terms in the expansion of the population correlation loss are of order 4 in the sum of the degrees of the overlaps $\alpha_u$ and $\alpha_v$, we can apply the main results from (Ben Arous et al., 2021), with information exponent $k = 4$, to get the thesis. Note that the population correlation loss is not simply monotonic in the overlaps, because it depends on both the DFT basis vectors $u$ and $v$, making the loss landscape more complex. To deal with that, it is possible to perform an analysis similar to the one in (Bardone & Goldt, 2024). $\qquad\square$

## F. Expansion of the population loss for non-isotropic inputs

We now focus on the setting in which the inputs are non-isotropic: both classes of inputs have a non-trivial circulant covariance $\Sigma$. In the next proposition, we expand the population correlation loss (6) in a form such that it is possible to explicitly identify the low-order terms in the overlaps.

*Proposition* 4. Consider $w \in \mathbb{R}^N$, $\alpha_u = w \cdot u$ and $\alpha_v = w \cdot v$, where $(u, v)$ are the DFT phase vectors. Let $\lambda_{k_0}$ be the $k_0$-th eigenvalue of a circulant matrix $\Sigma$ and sample the inputs $x^\mu$ from the Fourier data model (3) with circulant covariance $\Sigma$. Then, the population loss (6) is such that

$$L(w) = 1 - \lambda_{k_0}^2 \left( c_{04}\alpha_u^4 + c_{22}\alpha_u^2\alpha_v^2 + c_{04}\alpha_v^4 \right) \left[ c_4^\sigma + c_6^\sigma \frac{(\sigma_\Sigma^2 - 1)}{2} \right] + \text{h.o.t.}, \tag{22}$$

with $\sigma_\Sigma = \sqrt{w^\top \Sigma w}$. Here, $c_{04}, c_{22}$ and $c_{04}$ depend on the likelihood ratio between $\mathbb{P}$ and $\mathbb{P}_0$, $c_4^\sigma$ and $c_6^\sigma$ are the 4-th and 6-th Hermite coefficients of the activation $\sigma$.

*Proof.* The population correlation loss to be expanded is

$$L(w) = 1 - \frac{1}{2}\mathbb{E}_\mathbb{P}[\sigma(w \cdot x)] + \frac{1}{2}\mathbb{E}_{\mathbb{P}_0}[\sigma(w \cdot z)],$$

where we can write $\mathbb{E}_\mathbb{P}[\sigma(w \cdot x)] = \mathbb{E}_{\mathbb{P}_0}[\sigma(w \cdot x)\ell(u \cdot x, v \cdot x)]$. We start by expanding in Hermite polynomials the activation function $\sigma$. If the inputs $x$ are distributed according to the baseline distribution $\mathbb{P}_0$, by defining $a = w \cdot x$ and $\sigma_\Sigma = \sqrt{w^\top \Sigma w}$, we have that $A = a/\sigma_\Sigma \sim \mathcal{N}(0, 1)$. By Hermite expansion in the variable $w \cdot x = \sigma_\Sigma A$, it holds that

$$\sigma(\sigma_\Sigma A) = \sum_{n=0}^\infty \frac{c_n^\sigma}{n!} h_n(\sigma_\Sigma A),$$

where $c_n^\sigma$ and $h_n$ are the $n$-th order Hermite coefficient and Hermite polynomial, respectively. We can now apply a rescaling formula for Hermite polynomials to $h_n(\sigma_\Sigma A)$, which yields

$$h_n(\sigma_\Sigma A) = \sum_{m=0}^{\lfloor N/2 \rfloor} \frac{n!}{(n - 2m)!m!} \sigma_\Sigma^{n-2m} \left( \frac{\sigma_\Sigma^2 - 1}{2} \right)^m h_{n-2m}(A). \tag{23}$$

Define now

$$\rho_u = \frac{w^\top \Sigma u}{\sigma_\Sigma \sigma_B}, \rho_v = \frac{w^\top \Sigma v}{\sigma_\Sigma \sigma_C}, \rho_{uv} = \frac{v^\top \Sigma v}{\sigma_B \sigma_C},$$

where $\sigma_B = \sqrt{u^\top \Sigma u}$ and $\sigma_C = \sqrt{v^\top \Sigma v}$. Note that, since the DFT phase vectors $u$ and $v$ are orthogonal eigenvectors of $\Sigma$, we get that $\rho_{uv} = 0$. Then, we can write

$$A = \rho_v B + \rho_u C + \sqrt{1 - \rho_u^2 - \rho_v^2}\ Z,$$

where we have defined the rescaled variables $B$ and $C$ as $B = (x \cdot v)/\sigma_B$ and $C = (x \cdot u)/\sigma_C$, which are independent. If $Z \sim \mathcal{N}(0, 1)$ is independent of $B$ and $C$, we get that $B, C, Z$ are independent standard Gaussian random variables, and therefore i.i.d. Since $\rho_u^2 + \rho_v^2 + \tau^2 = 1$, if we define $\tau$ as $\tau = \sqrt{1 - \rho_u^2 - \rho_v^2}$, we can use the formulas for the sum of Hermite polynomials and conclude that

$$h_{n-2m} = \sum_{n-2m=i+j+k} \frac{(n - 2m)!}{i!j!k!} \rho_v^i \rho_u^j \tau^k h_i(C) h_j(B) h_k(Z). \tag{24}$$

Putting together Equation (23) and Equation (24), we end up with

$$\sigma(\sigma_\Sigma A) = \sum_{n=0}^\infty \sum_{m=0}^{\lfloor N/2 \rfloor} \sum_{n-2m=i+j+k}^\infty \frac{c_n^\sigma}{m!i!j!k!} \sigma_\Sigma^{n-2m} \left( \frac{\sigma_\Sigma^2 - 1}{2} \right)^m \rho_v^i \rho_u^j \tau^k h_i(C) h_j(B) h_k(Z). \tag{25}$$

Now that we have expanded the activation function $\sigma$, we want to similarly expand the likelihood ratio $\ell$ in Hermite polynomials. Define $\tilde{\ell}$ as the rescaled likelihood ratio such that $\ell(c, b) = \tilde{\ell}(c/\sigma_C, b/\sigma_B)$, with $c = v \cdot x$ and $b = u \cdot x$. We cannot directly expand $\ell$ in Hermite polynomials because we cannot compute its Hermite coefficients (which are $\mathbb{E}[\ell(C, B)h_i(C)h_j(B)]$), but we are actually able to compute the Hermite coefficients of its rescaled version $\tilde{\ell}$. Indeed, we have that

$$\ell(c, b) = \tilde{\ell}(C, B) = \sum_{i,j=0}^{+\infty} \frac{c_{ij}^{\tilde{\ell}}}{i!j!} h_i(C) h_j(B), \tag{26}$$

with coefficients given by

$$
\begin{aligned}
c_{ij}^{\tilde{\ell}} &= \mathbb{E}_{B,C}[\tilde{\ell}(B,C)h_i(C)h_j(B)] \\
&= \mathbb{E}_{B,C}[\ell(\sigma_B B, \sigma_C C)h_i(C)h_j(B)] \\
&= \mathbb{E}_{\mathbb{P}}\left[ h_i\left( \frac{v \cdot x}{\sigma_C} \right) h_j\left( \frac{u \cdot x}{\sigma_B} \right) \right],
\end{aligned}
$$

where we have used the definition of Hermite coefficients for $\tilde{\ell}$ in the the first equality and the definition of $\tilde{\ell}$ in the second one. In what follows, we will always denote them by $c_{ij}^{\tilde{\ell}}$. Note that we have already computed $c_{ij}^{\tilde{\ell}}$ in Lemma D.7, where we have called them simply $c_{ij}^\ell$. From now on, we will refer to them as $c_{ij}^\ell$. Hence,

$$
\mathbb{E}_{\mathbb{P}}[\sigma(w \cdot x)] = \mathbb{E}_{\mathbb{P}_0}[\sigma(w \cdot x)\ell(v \cdot x, u \cdot x)]
$$

$$
= \sum_{s,t,n=0}^{+\infty} \sum_{m=0}^{\lfloor n/2 \rfloor} \sum_{i+j+k=n-2m} \frac{c_{st}^\ell c_n^\sigma \sigma_\Sigma^{n-2m}}{s!t!m!i!j!k!} \left( \frac{\sigma_\Sigma^2 - 1}{2} \right)^m \rho_v^i \rho_u^j \tau^k \underbrace{\mathbb{E}_B[h_i(B)h_s(B)]}_{i!\delta_{is}} \underbrace{\mathbb{E}_C[h_j(C)h_t(C)]}_{j!\delta_{jt}} \mathbb{E}_Z[h_k(Z)]
$$

$$
= \sum_{n=0}^{+\infty} \sum_{m=0}^{\lfloor n/2 \rfloor} \sum_{i+j=n-2m} \frac{c_{ij}^\ell c_n^\sigma}{m!i!j!} \sigma_\Sigma^{n-2m} \left( \frac{\sigma_\Sigma^2 - 1}{2} \right)^m \rho_v^i \rho_u^j,
$$

where the last equality follows from the fact that the only non vanishing terms are the ones with $k = 0, i = s$ and $j = t$. If $\alpha_v = v \cdot w$ and $\alpha_u = u \cdot w$, we get that

$$
\rho_v^i = \frac{(w^\top \Sigma v)^i}{\sigma_\Sigma^i \sqrt{\lambda_{k_0}^i}} = \frac{\lambda_{k_0}^i \alpha_v^i}{\sigma_\Sigma^i \sqrt{\lambda_{k_0}^i}} = \frac{\lambda_{k_0}^{i/2} \alpha_v^j}{\sigma_\Sigma^i}
$$

and, analogously, $\rho_u^j = (\lambda_{k_0}^{j/2} \alpha_u^j)/\sigma_\Sigma^j$. Then, defining $n = i + j + 2m$, we get that

$$
\mathbb{E}_{\mathbb{P}}[\sigma(w \cdot x)] = \sum_{i,j,m=0}^{+\infty} \frac{c_{ij}^\ell c_{i+j+2m}^\sigma}{m!i!j!} \left( \frac{\sigma_\Sigma^2 - 1}{2} \right)^m \lambda_{k_0}^{(i+j)/2} \alpha_v^i \alpha_u^j \tag{27}
$$

$$
= \underbrace{\sum_{i,j=0,m\geq 0}^{+\infty} \frac{c_{2m}^\sigma}{m!} \left( \frac{\sigma_\Sigma^2 - 1}{2} \right)^m}_{\mathbb{E}_{\mathbb{P}_0}[\sigma(w \cdot x)]} + \sum_{\substack{m \geq 0, \\ i > 0 \text{ or } j > 0}} \frac{c_{ij}^\ell c_{i+j+2m}^\sigma}{m!i!j!} \left( \frac{\sigma_\Sigma^2 - 1}{2} \right)^m \lambda_{k_0}^{(i+j)/2} \alpha_v^i \alpha_u^j. \tag{28}
$$

Note that the first term in Equation (28) is exactly $\mathbb{E}_{\mathbb{P}_0}[\sigma(w \cdot x)]$. Indeed, we can take the expectation with respect to $B, C$ and $Z$ in Equation (25) and, as a result, only the addendum corresponding to $k = 0$ survives. Therefore, we conclude that

$$
L(w) = 1 - \frac{1}{2} \sum_{\substack{m \geq 0, \\ i > 0 \text{ or } j > 0}} \frac{c_{ij}^\ell c_{i+j+2m}^\sigma}{m!i!j!} \left( \frac{\sigma_\Sigma^2 - 1}{2} \right)^m \lambda_{k_0}^{(i+j)/2} \alpha_v^i \alpha_u^j.
$$

Since $\sigma$ is even, only the terms with $i + j + 2m \in 2\mathbb{N}$ do not vanish. We compute them up to sixth order in the total degree of the overlaps.

- **Order 2:** $i + j + 2m = 2$. We get that

$$
L_2 = \lambda_{k_0}\left[ c_2^\sigma \underbrace{\sum_{i=0}^2 \frac{\alpha_v^i \alpha_u^{2-i}}{i!(2-i)!} c_{i,2-i}^\ell}_{i+j=2, \, m=0} \right] = 0,
$$

since $c_{ij}^\ell = 0$ when $i + j = 2$ because of Lemma E.1.

- **Order 4:** $i + j + 2m = 4$. We get that

$$L_4 = \left[\underbrace{\lambda_{k_0}^2 c_4^\sigma \sum_{i=0}^4 \frac{\alpha_v^i \alpha_u^{4-i}}{i!(4-i)!} c_{i,4-i}^\ell}_{i+j=4,\ m=0} + \underbrace{\lambda_{k_0} c_6^\sigma \frac{(\sigma_\Sigma^2 - 1)}{2} \sum_{i=0}^2 \frac{\alpha_v^i \alpha_u^{2-i}}{i!(2-i)!} c_{i,2-i}^\ell}_{i+j=2,\ m=1}\right] = \lambda_{k_0}^2 c_4^\sigma \sum_{i=0}^4 \frac{\alpha_v^i \alpha_u^{4-i}}{i!(4-i)!} c_{i,4-i}^\ell.$$

- **Order 6:** $i + j + 2m = 6$. We get that

$$L_6 = \left[\underbrace{\lambda_{k_0}^3 c_6^\sigma \sum_{i=0}^4 \frac{\alpha_v^i \alpha_u^{6-i}}{i!(6-i)!} c_{i,6-i}^\ell}_{i+j=6,\ m=0} + \underbrace{\lambda_{k_0}^2 c_6^\sigma \frac{(\sigma_\Sigma^2 - 1)}{2} \sum_{i=0}^4 \frac{\alpha_v^i \alpha_u^{4-i}}{i!(4-i)!} c_{i,4-i}^\ell}_{i+j=4,\ m=1}\right].$$

Then, the population correlation loss reads as

$$L(w) = 1 - \lambda_{k_0}^2 \left[c_4^\sigma \sum_{i=0}^4 \frac{\alpha_v^i \alpha_u^{4-i}}{i!(4-i)!} c_{i,4-i}^\ell + c_6^\sigma \frac{(\sigma_\Sigma^2 - 1)}{2} \sum_{i=0}^4 \frac{\alpha_v^i \alpha_u^{4-i}}{i!(4-i)!} c_{i,4-i}^\ell\right] + \lambda_{k_0}^3 c_6^\sigma \sum_{i=0}^4 \frac{\alpha_v^i \alpha_u^{6-i}}{i!(6-i)!} c_{i,6-i}^\ell + \text{h.o.t.}$$

$$= 1 - \lambda_{k_0}^2 \left(\frac{c_{04}^\ell}{4!} \alpha_u^4 + \frac{c_{22}^\ell}{4} \alpha_u^2 \alpha_v^2 + \frac{c_{04}^\ell}{4!} \alpha_v^4\right) \left[c_4^\sigma + c_6^\sigma \frac{(\sigma_\Sigma^2 - 1)}{2}\right] + \text{h.o.t.}$$

where the last equality follows from the computations of the coefficients $c_{ij}^\ell$ in Lemma E.1. By defining $c_{04} = c_{04}^\ell/4!$, $c_{22} = c_{22}^\ell/4$ and $c_{40} = c_{40}^\ell/4!$, we get the thesis. We conclude by discussing briefly the order of the quadratic form associated to the covariance matrix $\Sigma$. We can write $\sigma_\Sigma^2 - 1$ as

$$\sigma_\Sigma^2 - 1 = (\lambda_{k_0} - 1)(\alpha_u^2 + \alpha_v^2) + \sum_{m=1}^{N-2} (\lambda_m - 1)(\alpha_{u_m}^2 + \alpha_{v_m}^2),$$

where we have denoted by $(u_m, v_m)_{m=1}^{N-1}$ the DFT basis eigenvectors of $\Sigma$, excluding the phase eigenvectors $u, v$. Since the degree of $\sigma_\Sigma^2 - 1$ is two in the overlaps, it induces a total contribution of order six. We include it anyway in the explicit expression of the loss because this contribution becomes dominant in case of eigenvalues scaling extensively with the dimension of the inputs - a case we will treat below. In conclusion, in "h.o.t." (higher-order terms), we can find terms of order six or more. $\qquad\square$

## G. Learning the phase at quasi-linear sample complexity with SGD

The goal of this section is to study the dynamics of online SGD on our classification task in a realistic non-isotropic setting. Specifically, we consider inputs drawn from the Fourier data model (3) with a non-trivial circulant covariance matrix $\Sigma$. We focus on the setting where a finite number of eigenvalues $\lambda_1, \ldots, \lambda_M$ of $\Sigma$ identify a principal subspace, and all the remaining eigenvalues are precisely equal to one. Additionally, we include $\lambda_{k_0}$, the eigenvalues corresponding to the DFT phase vectors, in the list of the non-trivial eigenvalues. As a result, the principal subspace of the inputs is spanned by the eigenvectors associated with $(\lambda_{k_0}, \lambda_1, \cdots, \lambda_M)$. In particular, we are going to analyse the case in which these non-trivial eigenvalues are extensive $O(N)$ with the dimension of the inputs; this is a realistic scenario, given the power-law decay observed in the spectrum of real images (cf. Figure 3).

Inspired by the setting proposed by Ben Arous et al. (2022), we consider the correlation loss Equation (1) plus a quartic penalization term $\beta \|w\|^4$, for $\beta > 0$, and online SGD without normalisation. More precisely, for any $t \geq 1$, the iteration reads as

$$w_t = w_{t-1} + \delta_N \nabla L(w, x_t, y_t)\big|_{w=w_{t-1}}, \tag{29}$$

for $\delta_N = O(1/N)$. Here, $L(w)$ and $L(w, Y)$ denote the population loss and the point-wise loss respectively, where $Y = (x_t, y_t)$ indicates the data point sampled at time $t$.

We study the dynamics of a collection of summary statistics $\boldsymbol{\alpha}$ during training. In particular, our summary statistics track the evolution of the weight vector projected onto the principal components of inputs drawn from the Fourier data model, including the DFT phase vectors.

## G.1. Literature review

We quickly recap the tools needed to address our classification problem on inputs drawn from the Fourier data model (3). These tools are borrowed from the results proven by Ben Arous et al. (2022) and Ben Arous et al. (2025b), where the authors analyse tensor PCA and classification on a Gaussian mixture model. Assume we have $H(w, Y) = L(w, Y) - L(w)$, where, as usual, the population correlation loss is $L(w) = \mathbb{E}_Y[L(w, Y)]$. Define also $V(w) = \mathbb{E}_Y[\nabla H(w) \otimes \nabla H(w)]$, which is essentially the covariance matrix for $\nabla H$ evaluated at the weight vector $w$. Consider a set of $S$ summary statistics $\boldsymbol{\alpha} = (\alpha_i)_{i=1}^S$, depending on the ambient dimension $N$, i.e. $\boldsymbol{\alpha} = \boldsymbol{\alpha}(N)$, and denote with $J = \nabla \alpha_i$ the Jacobian of the summary statistics. The following is a regularity assumption:

**Definition G.1.** We say that $(\boldsymbol{\alpha}, L)$ is "$\delta_N$-localizable" with localizing sequence $(E_K)_K$ if there is an exhaustion by compacts $(E_K)_K$ of $\mathbb{R}^S$ and constants $C_K$ independent of $N$ such that

1. $\max_i \sup_{w \in \boldsymbol{\alpha}^{-1}(E_K)} \left\| \nabla^2 \alpha_i \right\|_{\mathrm{op}} \leq C_K \, \delta_N^{-1/2}$ and $\max_i \sup_{w \in \boldsymbol{\alpha}^{-1}(E_K)} \left\| \nabla^3 \alpha_i \right\|_{\mathrm{op}} \leq C_K$.

2. $\sup_{w \in \boldsymbol{\alpha}^{-1}(E_K)} \left\| \nabla L(w) \right\| \leq C_K$ and $\sup_{w \in \boldsymbol{\alpha}^{-1}(E_K)} \mathbb{E}\left[ \|\nabla H\|^8 \right] \leq C_K \, \delta_N^{-4}$.

3. $\max_i \sup_{w \in \boldsymbol{\alpha}^{-1}(E_K)} \mathbb{E}\left[ \langle \nabla H, \nabla \alpha_i \rangle^4 \right] \leq C_K \delta_N^{-2}$, $\max_i \sup_{w \in \boldsymbol{\alpha}^{-1}(E_K)} \mathbb{E}\left[ \langle \nabla^2 \alpha_i, \nabla H \otimes \nabla H - V \rangle^2 \right] = o\left( \delta_N^{-3} \right)$.

Consider now the first and second-order differential operators

$$\mathcal{A}_N = \sum_i \partial_i L' \partial_i \quad \text{and} \quad \mathcal{L}_N = \frac{1}{2} \sum_{i,j} V_{ij} \partial_i \partial_j, \tag{30}$$

or, equivalently, $\mathcal{A}_N = \langle L', \nabla \rangle$ and $\mathcal{L}_N = \frac{1}{2} \langle V, \nabla^2 \rangle$.

**Definition G.2.** The summary statistics $\boldsymbol{\alpha}$ are "asymptotically closable" for learning rate $\delta_N$ if $(\boldsymbol{\alpha}, L)$ is $\delta_N$-localizable with localizing sequence $(E_K)_K$ and furthermore there exist locally Lipschitz functions $h : \mathbb{R}^S \to \mathbb{R}^S$ and $\Sigma : \mathbb{R}^S \times \mathbb{R}^{S \times S}$ such that

$$\sup_{w \in \boldsymbol{\alpha}^{-1}(E_K)} \left\| \left( -\mathcal{A}_N + \delta_N \mathcal{L}_N \right) \boldsymbol{\alpha} - h(\boldsymbol{\alpha}) \right\| \to 0, \quad \sup_{w \in \boldsymbol{\alpha}^{-1}(E_K)} \left\| \delta_N J_n V J_n^\mathsf{T} - \Sigma(\boldsymbol{\alpha}) \right\| \to 0. \tag{31}$$

We call $\Sigma$ **diffusion matrix** and $h$ **effective drift** for $\boldsymbol{\alpha}$. If $\mathcal{A}_N$ and $\delta_N \mathcal{L}_N$ admit themselves limits, i.e

$$\sup \left\| \mathcal{A}_N \boldsymbol{\alpha} - A_{\boldsymbol{\alpha}}(\boldsymbol{\alpha}) \right\| \to 0 \quad \text{and} \quad \sup \left\| \delta_N \mathcal{L}_N \boldsymbol{\alpha} - G(\boldsymbol{\alpha}) \right\| \to 0,$$

for some $A_{\boldsymbol{\alpha}}, G : \mathbb{R}^S \to \mathbb{R}^S$, where the supremum is taken over $w \in \boldsymbol{\alpha}^{-1}(E_K) \delta_N$, we call $A_{\boldsymbol{\alpha}}$ **population drift** and $G$ **population corrector**.

We present now the theorem providing the evolution of the summary statistics at linear time, originally Theorem 2.2. in (Ben Arous et al., 2022). The solution to the given SDE is called **effective dynamics**.

**Theorem G.3** (Effective dynamics). *Let $(w_t)_t \subseteq \mathbb{R}^N$ be the estimators given by online SGD (29) initialized from $w_0 \sim \mu_0(\mathbb{R}^N)$, with learning rate $\delta_N$, applied to the loss $L'(\cdot, \cdot)$. For the family of summary statistics $\boldsymbol{\alpha} = (\alpha_i)_{i=1}^S$, let $\boldsymbol{\alpha}_N(t)$ be the linear interpolation of $\left( \boldsymbol{\alpha}(w_{\lfloor t \delta_N^{-1} \rfloor}) \right)_t$. Suppose that $\boldsymbol{\alpha}$ are asymptotically closable with learning rate $\delta_N$, effective drift $h$ and diffusion matrix $\Sigma$, and that the pushforward of the initial data satisfies $\boldsymbol{\alpha}_\# \mu_0 \to \nu$ weakly for some measure $\nu = \nu(\mathbb{R}^k)$. Then $\boldsymbol{\alpha}_N(t) \to (\boldsymbol{\alpha}_t)_t$ weakly as $N \to \infty$, where $(\boldsymbol{\alpha}_t)_t$ solves the stochastic differential equation*

$$d\boldsymbol{\alpha}_t = h(\boldsymbol{\alpha}_t) \, dt + \Sigma(\boldsymbol{\alpha}_t)^{1/2} \, dB_t, \tag{32}$$

*initialized from $\nu$, and $(B_t)_t$ is a standard Brownian motion in $\mathbb{R}^S$.*

## G.2. Results for the Fourier data model

We want to apply Theorem G.3 to our classification task, in presence of a non-trivial covariance matrix for both classes of inputs $\mathbb{P}$ and $\mathbb{P}_0$. We assume that both classes share a principal subspace spanned by some leading principal components $(u^m, v^m)_{m=1}^M$ together with the DFT phase vectors $(u, v)$. We want to describe the dynamics of the overlap of the perceptron weight vector and these principal components. First of all, we formalise the definition of the summary statistics we deal with.

**Definition G.4** (Summary statistics). For a fixed integer $M \in \mathbb{N}$, assume that $(\lambda_m)_{m=1}^M$ are eigenvalues of the covariance matrix $\Sigma$ of inputs sampled from the Fourier data model (3), with $\lambda_m > 1$. In addition, assume also that the eigenvalue $\lambda_{k_0}$ corresponding to the DFT phase vectors is such that $\lambda_{k_0} > 1$. For simplicity, set all the other eigenvalues to one. We define the summary statistics $\boldsymbol{\alpha}$ as

$$\boldsymbol{\alpha} = (\alpha_u, \alpha_v, (\alpha_{u_m}, \alpha_{v_m})_{m=1}^M, \omega_\perp),$$

where $\alpha_u$ and $\alpha_v$ are the overlaps of the weight vector $w$ with the DFT phase vectors $u$ and $v$. The other statistics $\alpha_{u_1}, \alpha_{v_1}, \ldots, \alpha_{u_M}, \alpha_{v_M}$ are the overlaps of the weight vector with the DFT basis vectors associated to the set of the eigenvalues identifying the principal subspace $(\lambda_m)_{m=1}^M$. Moreover, $\omega_\perp = w \cdot w_\perp$, where $w_\perp$ is the projection of the weight vector $w$ onto the subspace orthogonal to that spanned by the whole set of principal components $(u, v, (u^m, v^m)_{m=1}^M)$.

Due to the definition of our (linear) statistics, we simply get that the corrector term is $\mathcal{L}_N \boldsymbol{\alpha} = 0$. The population drift is given by $A_{\boldsymbol{\alpha}} = \lim_{N \to \infty} \mathcal{A}_N \boldsymbol{\alpha}$, if this limit exists. To identify the effective dynamics for the summary statistics, one should also compute the diffusive matrix.

### G.2.1. NEAR-ISOTROPIC INPUTS

In the next lemma, we compute the population drift for the summary statistics, which specifies the population dynamics, when the non-trivial eigenvalues $\lambda_{k_0}, \lambda_1, \ldots, \lambda_M$ are $O(1)$.

**Lemma G.5** (Population drift). *Consider the population classification loss (6) and the summary statistics introduced in Definition G.4. Assume that all the non-trivial eigenvalues $(\lambda_{k_0}, (\lambda_m)_{m=1}^M)$ of the covariance matrix of the inputs are $O(1)$. Then, the population drift*

$$A_{\boldsymbol{\alpha}} = (A_u, A_v, (A_{u_m}, A_{v_m})_{m=1}^M, A_{\omega_\perp})$$

*is as follows: by defining $R = \|w\|^2$, the drifts for the DFT phase vectors are*

$$\begin{aligned}
A_u(\boldsymbol{\alpha}) &= \lim_{N \to +\infty} \mathcal{A}_N \alpha_u \\
&= \frac{1}{2} \lambda_{k_0}^2 \left[ \sum_{i=0,2} \frac{\alpha_v^i \alpha_u^{3-i} c_{i,4-i}^\ell}{i!(3-i)!} \left( c_4^\sigma + c_6^\sigma \frac{\sigma_\Sigma^2 - 1}{2} \right) + c_6^\sigma \sum_{i=0,2,4} \frac{\alpha_v^i \alpha_u^{4-i} c_{i,4-i}^\ell}{i!(4-i)!} (\lambda_{k_0} - 1) \alpha_u \right] + 4\beta R^2 \alpha_u,
\end{aligned}$$

$$\begin{aligned}
A_v(\boldsymbol{\alpha}) &= \lim_{N \to +\infty} \mathcal{A}_N \alpha_v \\
&= \frac{1}{2} \lambda_{k_0}^2 \left[ \sum_{i=1,3} \frac{\alpha_v^i \alpha_u^{3-i} c_{i+1,3-i}^\ell}{i!(3-i)!} \left( c_4^\sigma + c_6^\sigma \frac{\sigma_\Sigma^2 - 1}{2} \right) + c_6^\sigma \sum_{i=0,2,4} \frac{\alpha_v^i \alpha_u^{4-i} c_{i,4-i}^\ell}{i!(4-i)!} (\lambda_{k_0} - 1) \alpha_v \right] + 4\beta R^2 \alpha_v.
\end{aligned}$$

*For the other principal components $(u^m, v^m)_{m=1}^M$, we get that the drifts are*

$$A_{u_m}(\boldsymbol{\alpha}) = \lim_{N \to +\infty} \mathcal{A}_N \alpha_{u_m} = \frac{1}{2} c_6^\sigma \lambda_{k_0}^2 \sum_{i=0,2,4} \frac{\alpha_v^i \alpha_u^{4-i}}{i!(4-i)!} c_{i,4-i}^\ell (\lambda_m - 1) \alpha_{u_m} + 4\beta R^2 \alpha_{u_m}$$

*and same for $(v^m)_m$. For the orthogonal part $\omega_\perp$, we have that*

$$A_{\omega_\perp}(\boldsymbol{\alpha}) = \lim_{N \to +\infty} \mathcal{A}_N \omega_\perp = 4\beta R^2 \omega_\perp.$$

*Proof.* Since $\|w\|^4 = (\alpha_u^2 + \alpha_v^2 + \cdots + \alpha_{u_M} + \alpha_{v_M} + \omega_\perp^2)^2$, we get that the gradient of the population correlation loss can be written as

$$\nabla L(w) = (\mathcal{A}_N \alpha_u) u + (\mathcal{A}_v \alpha_v) v + \sum_{m=1}^M ((\mathcal{A}_N \alpha_{u_m}) u^m + (\mathcal{A}_N \alpha_{v_m}) v^m) + 4(R^2 r_\perp) w_\perp.$$

Hence,

$$\mathcal{A}_N \alpha_u = -\frac{1}{2} \lambda_{k_0}^2 \left[ \sum_{i=0}^3 \frac{\alpha_v^i \alpha_u^{3-i} c_{i,4-i}^\ell}{i!(3-i)!} \left( c_4^\sigma + c_6^\sigma \frac{\sigma_\Sigma^2 - 1}{2} \right) + c_6^\sigma \sum_{i=0}^4 \frac{\alpha_v^i \alpha_u^{4-i} c_{i,4-i}^\ell}{i!(4-i)!} (\lambda_{k_0} - 1) \alpha_u \right] + 4\beta R^2 \alpha_u,$$

$$\mathcal{A}_N \alpha_v = -\frac{1}{2}\lambda_{k_0}^2 \left[ \sum_{i=0}^{3} \frac{\alpha_v^i \alpha_u^{3-i} c_{i+1,3-i}^{\ell}}{i!(3-i)!} \left( c_4^{\sigma} + c_6^{\sigma} \frac{\sigma_{\Sigma}^2 - 1}{2} \right) + c_6^{\sigma} \sum_{i=0}^{4} \frac{\alpha_v^i \alpha_u^{4-i} c_{i,4-i}^{\ell}}{i!(4-i)!} (\lambda_{k_0} - 1) \alpha_v \right] + 4\beta R^2 \alpha_v,$$

$$\mathcal{A}_N \alpha_{u_m} = -\frac{1}{2} c_6^{\sigma} \lambda_{k_0}^2 \sum_{i=0}^{4} \frac{\alpha_v^i \alpha_u^{4-i}}{i!(4-i)!} (\lambda_m - 1) \alpha_{u_m} + 4\beta R^2 \alpha_{u_m},$$

$$\mathcal{A}_N \alpha_{v_m} = -\frac{1}{2} c_6^{\sigma} \lambda_{k_0}^2 \sum_{i=0}^{4} \frac{\alpha_v^i \alpha_u^{4-i}}{i!(4-i)!} (\lambda_m - 1) \alpha_{v_m} + 4\beta R^2 \alpha_{v_m}.$$

The population drift for $\omega_\perp$ is simply

$$\mathcal{A}_N \omega_\perp = -4R^2 \beta \omega_\perp.$$

We have assumed that $\lambda_{k_0}$ and the other eigenvalues do not scale with the ambient dimension, and so none of the right-hand-sides depend on $N$. Then, the drifts stay the same when taking the limit for $N \to +\infty$. The thesis follows from the observation that $c_{13}^{\ell} = c_{31}^{\ell} = 0$, because of Lemma E.1. $\qquad\square$

In view of the described population dynamics, we claim in Conjecture 5.2 that SGD cannot successfully perform classification at linear sample complexity when the leading eigenvalues of the covariance matrix of the inputs are non-extensive. More precisely, we expect (and provide an heuristic derivation in Remark G.6) that it is possible for SGD to exit the search phase only at cubic sample complexity, as in the setting treated in Theorem 3.1 for isotropic data points.

*Remark* G.6 (Cubic sample complexity for near-isotropic inputs). Consider the setting of Conjecture 5.2, i.e. when a finite number $O(1)$ of non-unit eigenvalues of the covariance matrix is $O(1)$. It is possible to heuristically derive, following Damian et al. (2023) and Ricci et al. (2025a), that online SGD requires a cubic sample complexity to escape the search phase. Indeed, the evolution of any summary statistic, e.g. $\alpha_v$, is given by

$$\dot{\alpha}_v = \frac{\mathbb{E}[g \cdot v]^2}{\alpha_v \mathbb{E}[\|g\|^2]},$$

where $g = \nabla L(w; x, y)$. Since $g$ is a random vector in $\mathbb{R}^N$ where each coordinate is $O(1)$, we have that $\mathbb{E}[\|g\|^2] \approx N$. Following the calculations in Lemma G.5, we get $\mathbb{E}[g \cdot v] \approx \alpha_v^3$. Then, $\dot{\alpha}_v = \alpha_v^5 / N$ which, initialised at $\alpha(0) = 1/\sqrt{N}$, implies that $\alpha_v = O(1)$ in approximately $N^3$ steps.

### G.2.2. Power-law decaying inputs

In this section we compute the population drifts when the top eigenvalues $((\lambda_m)_m, \lambda_{k_0})$ are extensive $O(N)$ with respect to the ambient dimension $N$.

The presence of a shared leading principal subspace for the covariance matrix of both classes has the effect of inducing an extensive signal-to-noise ratio, which allows a faster recovery - from cubic (as in the isotropic case) to quasi-linear. An extensive signal-to-noise ratio has been considered in Ben Arous et al. (2018) and it was indeed crucial to effectively solve the tensor PCA problem.

Note that, since this subspace is shared among the two classes of inputs, it is not immediately clear that it impacts the speed of solving the task, as it is not task-relevant. Nevertheless, we establish that it accelerates the learning of the information contained in higher-order cumulants of the inputs, namely in the phase.

Furthermore, since the eigenvalues are extensive, the previous approach breaks down, as we would have $A_\alpha = \infty$. What we can do is zooming close to the equator and study the effective dynamics for the rescaled summary statistics, as suggested by (Ben Arous et al., 2022).

Consider then a rescaling of the summary statistics in a microscopic neighborhood of zero, which captures the initial phase of their evolution from a random start of the weight vector. The new rescaled statistics are defined as

$$\boldsymbol{m} = (\sqrt{N}\alpha_u, \sqrt{N}\alpha_v, (\sqrt{N}\alpha_{u_m}, \sqrt{N}\alpha_{v_m})_{m=1}^{M}, \omega_\perp). \tag{33}$$

Note that the rescaled statistics are initialised at the non-vanishing measure

$$\nu = \lim_{N \to +\infty} \boldsymbol{m}_{\#}\mu_0 = \mathcal{N}(0,1) \otimes \cdots \otimes \mathcal{N}(0,1) \otimes \delta_1.$$

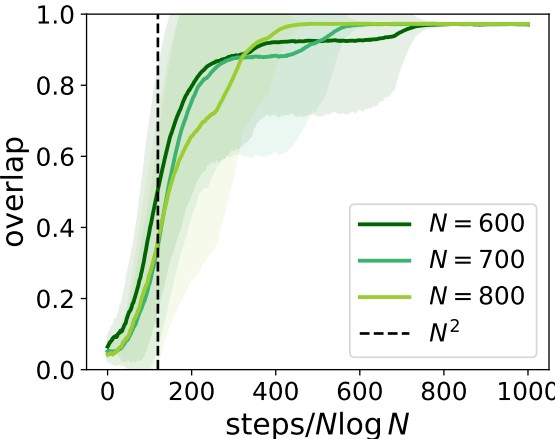

*Figure 10.* **Fast recovery of the phase information.** Consider the correlation loss (1) and non-isotropic inputs sampled from the Fourier data model (3), with circulant covariance matrix in which only the eigenvalue corresponding to the modified mode $k_0$ in the phase is extensive ($\lambda_{k_0} \approx \sqrt{N}$), the other ones are set equal to one. Recall that the covariance matrix is shared among the inputs of both classes. We run SGD to distinguish the two classes of inputs, in dimensions $N = 600, 700, 800$. SGD weakly recovers the subspace spanned by the DFT phase vectors at quasi-linear sample complexity - on the $y$- axis we have the norm of the projection of the weight vector in the subspace spanned by the DFT phase vectors. The reason why such fast recovery happens is explained in Conjecture 5.3. Note that the signal is recovered within $n \ll N^2$ steps, as happens in the case of information exponent $k^* = 2$ for single index models. We use the log cosh activation function.

In the next lemma, we compute the population drift $A_{\boldsymbol{m}}(\boldsymbol{m})$ for the rescaled statistics. Note that from the following calculations it is clear that this population drift would vanish in case of near-isotropic inputs. Conversely, $A_{\boldsymbol{m}}(\boldsymbol{m})$ is not identically zero when $\lambda_{k_0}$ is extensive. More precisely, the minimal scaling required for the induced signal-to-noise is $\lambda_{k_0}^2 \approx N$, which is the same required to solve tensor PCA for a fourth-order tensor with SGD at linear sample complexity, as proven in Ben Arous et al. (2018).

**Lemma G.7** (Rescaled population drift)**.** *Consider the correlation classification loss* (6) *and the rescaled summary statistics* (12)*. Assume that $\lambda_{k_0} \approx \sqrt{N}$ and $\sqrt{N} \lesssim \lambda_m \lesssim N$. Then, the population drift for the rescaled statistics is*

$$A_{\boldsymbol{m}} = (A_{m_u}, A_{m_v}, (A_{m_{u_M}}, A_{m_{v_M}})_{m=1}^M, A_{\omega_\perp})$$

*such that*

$$A_{m_u} = \lim_{N \to +\infty} \mathcal{A}_N m_u = \sum_{i=0,2} \frac{m_v^i m_u^{3-i} c_{i,4-i}^\ell}{2i!(3-i)!} c_4^\sigma + \frac{c_6^\sigma}{4} \sum_{m=1}^M (m_{u_m}^2 + m_{v_m}^2) + 4\beta R^2 m_u,$$

$$A_{m_v} = \lim_{N \to +\infty} \mathcal{A}_N m_v = \sum_{i=1,3} \frac{\alpha_v^i \alpha_u^{3-i} c_{i+1,3-i}^\ell}{2i!(3-i)!} c_4^\sigma + \frac{c_6^\sigma}{4} \sum_{m=1}^M (m_{u_m}^2 + m_{v_m}^2) + 4\beta R^2 m_v$$

*for the statistics $m_u$ and $m_v$ associated to the* DFT *phase vectors. Conversely, for any $m = 1, \ldots, M$, we get*

$$A_{m_{u_m}} = \lim_{N \to +\infty} \mathcal{A}_N m_{u_m} = \frac{c_6^\sigma}{2} \sum_{i=0,2,4} \frac{m_v^i m_u^{4-i}}{i!(4-i)!} c_{i,4-i}^\ell \, m_{u_m} + 4\beta R^2 m_{u_m}$$

*and same for $m_{v_m}$. The drift for $\omega_\perp$ stays the same as the previous case and implies that $\dot{\omega}_\perp = -4\beta R^2 \omega_\perp$.*

*Proof.* By definition of $\sigma_\Sigma$, we have that

$$\sigma_\Sigma^2 - 1 = (\lambda_{k_0} - 1)(\alpha_u^2 + \alpha_v^2) + \sum_{m=1}^M (\lambda_m - 1)(\alpha_{u_m}^2 + \alpha_{v_m}^2)$$

$$= \frac{(\sqrt{N} - 1)}{N}(m_u^2 + m_v^2) + \frac{(N - 1)}{N} \sum_{m=1}^M (m_{u_m}^2 + m_{v_m}^2).$$

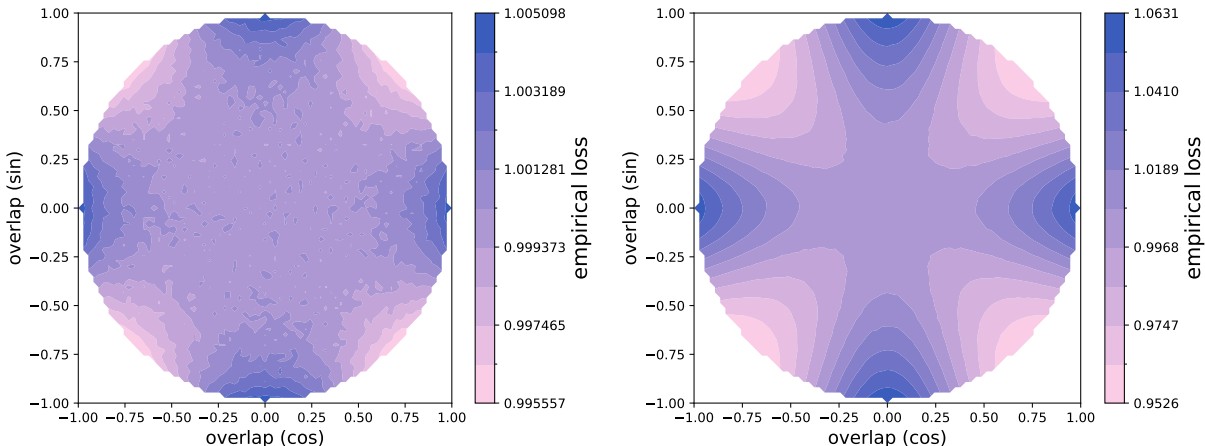

*Figure 11.* **Level sets of the empirical loss for isotropic (left) and power-law decaying (right) inputs sampled from the Fourier data model.** The acceleration due to the increase of the drift magnitude in the anisotropic setting corresponds to a less flat landscape (note the pink steeps are closer to the origin), which resolves in a faster escape from the search phase. On the right, we have $\lambda_{k_0} = \sqrt{N}$. In both figures, we use $N^3$ samples to empirically approximate the landscape, for $N = 200$. We use the $\log \cosh$ activation function.

Then, assuming that $\lambda_m^2 \approx N$, we get

$$\mathcal{A}_N m_u = \overbrace{\frac{\sqrt{N}}{2}N\left[\sum_{i=0,2}\frac{m_v^i m_u^{3-i}c_{i,4-i}^\ell}{\sqrt{N}^3 i!(3-i)!}\left(c_4^\sigma + \frac{c_6^\sigma(\sqrt{N}-1)}{2N}(m_u^2+m_v^2)+\overbrace{\frac{c_6^\sigma(N-1)}{2N}\sum_{m=1}^M(m_{u_m}^2+m_{v_m}^2)}^{\Theta(1)}\right)\right.}^{\Theta(1)}$$
$$\left.+ c_6^\sigma\sum_{i=0,2,4}\frac{m_v^i m_u^{4-i}c_{i,4-i}^\ell}{\sqrt{N}^4 i!(4-i)!}\frac{(\sqrt{N}-1)}{\sqrt{N}}m_u\right] + 4\beta R^2 m_u.$$

Taking the limit for $N \to +\infty$, the population drift for $m_u$ reads as

$$A_{m_u} = \sum_{i=0,2}\frac{m_v^i m_u^{3-i}c_{i,4-i}^\ell}{i!(3-i)!}c_4^\sigma + \frac{c_6^\sigma}{2}\sum_{m=1}^M(m_{u_m}^2+m_{v_m}^2) + 4\beta R^2 m_u.$$

By doing the same for the other principal components, we get that

$$\mathcal{A}_N m_v = \frac{\sqrt{N}}{2}N\left[\sum_{i=0,2}\frac{m_v^i m_v^{3-i}c_{i+1,3-i}^\ell}{\sqrt{N}^3 i!(3-i)!}\left(c_4^\sigma + \frac{c_6^\sigma(\sqrt{N}-1)}{2N}(m_u^2+m_v^2)+\frac{c_6^\sigma(N-1)}{2N}\sum_{m=1}^M(m_{v_m}^2+m_{v_m}^2)\right)\right.$$
$$\left.+ c_6^\sigma\sum_{i=0,2,4}\frac{m_v^i m_u^{4-i}c_{i,4-i}^\ell}{\sqrt{N}^4 i!(4-i)!}\frac{(\sqrt{N}-1)}{\sqrt{N}}m_v\right] + 4\beta R^2 m_v,$$
$$\mathcal{A}_N m_{u_m} = \underbrace{\frac{\sqrt{N}}{2}c_6^\sigma N\sum_{i=0,2,4}\frac{m_v^i m_u^{4-i}}{\sqrt{N^4}i!(4-i)!}c_{i,4-i}^\ell\frac{(N-1)}{\sqrt{N}}m_{u_m}}_{\Theta(1)} + 4\beta R^2 m_{u_m},$$

and same for $m_{v_m}$, for any $m = 1,\ldots,M$. By taking the limit for $N \to +\infty$, we obtain the thesis. Note that the calculations do not change for the larger range of scalings $\sqrt{N} \lesssim \lambda_m \ll N$. $\qquad\square$

*Remark* G.8. From Lemma G.7, the population dynamics of the rescaled summary statistics is given by

$$\dot{m}_u = c_1 m_u^3 + c_2 m_v^2 m_u + \sum_{m=1}^{M} c_3 (m_{u_m}^2 + m_{v_m}^2) + 4\beta \|w\|^4 m_u,$$

$$\dot{m}_{u_m} = c_4 \sum_{i=0,2,4} \frac{m_v^i m_u^{4-i}}{i!(4-i)!} c_{i,4-i}^\ell \, m_{u_m} + 4\beta \|w\|^4 m_{u_m}, \tag{34}$$

for suitable constants $c_1, \ldots, c_4 \in \mathbb{R}$, and similar for $m_v$ and $m_{v_m}$.

*Remark* G.9. Note that, if the leading eigenvalues in Lemma G.7 are near-isotropic, the population drift $A_{\boldsymbol{m}}$ for the rescaled statistics turns out to be identically zero, due to the limit taken for $N \to +\infty$. The same happens if $\lambda_{k_0}^2 = o(N)$.

