# OpenReview forum: "A Fourier perspective on the learning dynamics of neural networks: from sample complexities to mechanistic insights"
_ICML.cc/2026/Conference — ICML 2026 regular_

### Official Review · Reviewer_zTig · 2026-02-16

**Soundness:** 3
**Presentation:** 1
**Significance:** 1
**Originality:** 2
**Overall Recommendation:** 3
**Confidence:** 4

**Summary:**

Motivated by image classification and neural networks' tendency to learn patterns in increasing order of complexity ("simplicity bias"), the manuscript proposes a generative model that produces two classes of images with the same mean and covariance matrix but different fourth-order cumulants. After validating the simplicity bias hypothesis experimentally on a real-image dataset, showing that the network first learns the amplitude, then the phase, necessary to distinguish the images, the authors focus on the theoretical analysis of the proposed model. In the isotropic setting, they show that spherical SGD will require a sample complexity of order $O(N^3)$. They subsequently show that anisotropic noise can substantially accelerate learning. Their analysis is based on the population loss closed-form expression and conjectural extensions of B. Arous et al. (2022,2025) works in the anisotropic case.

**Compliance With Llm Reviewing Policy:**

Affirmed.

**Final Justification:**

I thank the authors for their rebuttal, which partially addressed my concerns. While the paper has clear merit (an interesting problem and model), I remain somewhat skeptical about the technical novelty. However, I have raised my score, as some of my concerns have been addressed and the overall consensus leans toward acceptance.

**Key Questions For Authors:**

1) In order to clarify the results in the anisotropic case, would it be possible to derive a clean analysis in the rank-one perturbed covariance matrix setting where only $\lambda_{k_0}$ is modified? This would i) isolate the effect of $\lambda_{k_0}$ without introducing multi-spikes or power-law covariance data, ii) make explicit the scaling required to see a difference with the isotropic case, and iii) allow a clearer comparison with results existing for the single-index model.

2) In the work of Moussav-Hosseini et al. (2023), the authors show that spherical SGD could fail dramatically in anisotropic settings. Why did you consider this algorithm as your baseline algorithm? I think a discussion to explain why the algorithm is more relevant in this context than the aforementioned work is needed.

3) How important is the choice of the optimization algorithm? I believe the numerical experiment reported in Figure 1 used a standard optimization method, such as mini-batch SGD or Adam, but the analyzed algorithm is spherical SGD applied to a specific loss. Would you expect the “amplitude-first, phase-later” phenomenon to be robust across optimizers/losses?

4) Could you clarify the claim that "The key difficulty in analysing the population correlation loss (6) in this setting is that it is not possible anymore to directly perform the expansion in Hermite polynomials [...]" (line 364)? To my understanding, one can still use Hermite polynomial decompositions in the anisotropic case, provided the input distribution is Gaussian. The main difference is that the formula becomes more complicated due to different scaling effects, making the resulting dynamical system harder to analyze.

**Limitations:**

yes

**Strengths And Weaknesses:**

Strengths:

1) The problem is very well-motivated, linking empirical observation in vision problems with theoretical questions about the simplicity bias in neural networks.

2) The proposed toy model isolates key properties in a clean way of the aforementioned problem while being analytically tractable.

Weaknesses:

1) The hardness of learning the signal when the baseline noise is isotropic is only assessed through the failure of spectral SGD. But there is no reason to believe that using this algorithm in combination with the correlation loss should be optimal in terms of sample complexity (cf. Ricci et al. (2025)). One needs a stronger argument to demonstrate that in the isotropic setting, the problem is intrinsically more difficult than in the anisotropic setting.

2) Section 5 is very confusing. Except for the closed-form expression of the loss, all the results are stated as conjectures, but presented as resulting from a rigorous analysis. The author should clarify what has been demonstrated and what is heuristic.

3) In Section 4 the authors informally emphasized the importance of the power-law spectrum without defining it. A common definition of power law spectrum is that the eigenvalues $\lambda_i\propto i^{-a}$ for some $a>0$. But this doesn't match the scaling used in Section 5. Moreover, in Section 5.2, they seem to reduce the analysis to an $M+1$-spike model.

4) In Section 5.2, line 397, the author defined a new loss function informally by adding truncation and regularization. I think this should be emphasized since it is not the same algorithm that has been used for isotropic and anisotropic settings.

5) The technical development closely follows the framework of Ben Arous et al., and it would be helpful for the authors to more clearly delineate what is genuinely new beyond adapting existing machinery to the present toy model.

6) Some related works are missing. Contrary to the author's claim (line 372), non-linear regression models with general anisotropic covariance (not the rank one spiked model) have been analyzed in Wortsman and Loureiro (2025) "Kernel ridge regression under power-law data: spectrum and generalization", Braun et al. (2025) "Fast Escape, Slow Convergence: Learning Dynamics of Phase Retrieval under Power-Law Data", etc. Also for multi-spiked model: B. Arous et al (2024) "Stochastic gradient descent in high dimensions for multi-spiked tensor PCA".

7) The algorithm used in Section 3 is spherical SGD, but it is often referred to as SGD; this is quite misleading.

---

> ### Author Rebuttal · Authors · 2026-03-31
>
> We thank the reviewer for their careful review of our paper.
>
> > Intrinsic hardness of learning phases
>
> We focus on online SGD to be as close as possible to the main algorithm used to train deep networks. We agree that online SGD is unlikely to be sample-optimal, but providing  statistical or computational limits for generic polynomial-time algorithms is beyond our scope. In any case, we expect that smoothing the landscape (Biroli et al. J Stat 2020, Damian et al. NeurIPS 2023, Ricci et al. ICML 2025) would close a potential gap between online SGD and computational thresholds.
>
> > Status of conjectures
>
> This is a good point. Obtaining sample complexities of online SGD rigoroulsy requires control of the effective volatility of the stochastic process (Theorem F.3). This has only been achieved for Gaussian additive models like single-index models or Gaussian mixtures (cf. Ben Arous et al. '22, '25). Our inputs are strictly non-Gaussian, and hence outside these cases. We rigorously derive the population dynamics of the summary statistics in Lemma F.5 and F.7, which reveal weak recovery at different time scales, for both penalised and non-pensalied losses, depending only on the statistics of the data (near-isotropic vs. power-law decay).
>
> > Definition of the power-laws
>
> Yes, we set $\lambda_m \propto m^{-\alpha}$, apologies for not stating it explicitly. We expand the loss in the general case of an extensive (with the dimension) number of non-trivial eigenvalues in Section 5.1. In Section 5.2, we focus on power-laws with exponent $\alpha > 1$, which is typically observed in natural images (Hyvärinen et al. '09). Since the effective dimension of the covariance in this case is $O(1)$ (see Cheng & Montanari '24 or Wortsman & Loureiro, arXiv:2510.04780), we consider an effective low-rank eigenstructure that simplifies the analysis and maintains the correct scalings.
>
> > Truncation and regularization for the loss
>
> We regularise the loss in Sec. 5.2 to match the setting of Ben Arous et al. '22. However, truncation and regulatisation do not change the results for the population dynamics in Conjectures 5.2 and 5.3. This can be seen by looking at the population drifts in Lemma F.5 and F.7.
>
> > Novelty w.r.t. Ben Arous et al.
>
> On the technical side, our focus is on the non-Gaussian structure of the inputs, which cannot be reduced to a Gaussian additive model (see Lemma C.7). In this setting, anisotropy has not been treated yet (see next point) and makes the dynamics highly non-trivial.
>
> > Related work on anisotropy
>
> Thank you for pointing out these references, which we will discuss in the revision. We study anisotropic non-Gaussian inputs. Wortsman and Loureiro ('25) study anisotropic covariances in _kernel ridge_ regression over _Gaussian_ inputs, but kernels struggle to learn single-index models efficiently, see Chapter 3 of Misiakiewicz and Montanari arXiv:2308.13431. Braun et al. (2025) study phase retrieval for _Gaussian_ inputs. Likewise, Ben Arous et al ('24) study a Gaussian additive model, which is a crucial simplification compared to our case, where non-Gaussianity and anisotropy fundamentally reshape the population dynamics and sample complexities.
>
> > Case of a rank-one perturbed covariance matrix
>
> If only $\lambda_{k_0} \approx \sqrt{N}$, then $\sigma_{\Sigma}^2- 1 \approx 1/\sqrt{N}$. From eq. 9,
> we get $L(w) \approx 1/N$ and an information exponent $k^* = 2$  (task can be solved at quasi-linear sample complexity). Our simulations in (https://figshare.com/s/d6d73683370bb9064aaa) support this conclusion. For $\lambda_{k_0} \approx N^{1/4}$, the information exponent instead becomes $k^*=3$, turning into quadratic sample complexity.
>
> > Limits of SGD on anisotropy
>
> It is true that spherical SGD often tails on inputs with high variability along one direction. Here, in contrast, spherical SGD does not fail even in presence of a (multi)spiked-covariance, which instead speeds up recovery of the signal.
>
> > Optimization algorithm
>
> Thank you for raising this interesting point. We have repeated our experiments with Resnet18 on an additional data set (CIFAR100) with both SGD and Adam, and find the same speedup as in fig 3.e, see  (https://figshare.com/s/0fd6b37b82fd6b0bc21e), for cross-entropy loss. In total, we have verified our predictions on three data sets (textures, ImageNet, CIFAR100) and two optimisers (mini-batch SGD and Adam).
>
> > Anisptropic expansion
>
> We take the expectation over the baseline anisotropic Gaussian distribution of the product between $\sigma$ and $\ell$ (cf. Appendix E). After expanding both, we first need to rescale (17) and resum (18) our formulas before applying the orthogonality properties (Lemma B.14); otherwise, one cannot write the expectation (so the loss) in terms of the relevant overlaps.
>
> > Summary
>
> We hope our replies have addressed your concerns. If so, we would appreciate it if you could revise your score; if not, we look forward to addressing any remaining issue during the discussion.

---

> > ### Author Rebuttal · Reviewer_zTig · 2026-04-03
> >
> > I thanks the authors for their clarifications. I still believe that the claimed novelty from non-Gaussianity is an overstated. To my understanding, the computation of the population loss from which derives the population dynamics analysis reduce to evaluate the correlation between Hermite polynomials of rescaled Gaussian r.v. (and this is not really novel). In that sense, in it looks more than an adaptation of the classical approach than a substantially new framework for dealing with non Gaussian inputs. The main challenge seems to be the analysis of the noise effect (it's here that the model clearly departs from additive like Gaussian/symmetric distribution and additive noise). However, this is precisely where the analysis remains incomplete.

---

> > > ### Author Response · Authors · 2026-04-06
> > >
> > > > I thanks the authors for their clarifications. I still believe that the claimed novelty from non-Gaussianity is an overstated. To my understanding, the computation of the population loss from which derives the population dynamics analysis reduce to evaluate the correlation between Hermite polynomials of rescaled Gaussian r.v. (and this is not really novel). In that sense, in it looks more than an adaptation of the classical approach than a substantially new framework for dealing with non Gaussian inputs. The main challenge seems to be the analysis of the noise effect (it's here that the model clearly departs from additive like Gaussian/symmetric distribution and additive noise). However, this is precisely where the analysis remains incomplete.
> > >
> > >
> > > We are happy to further clarify the significance of our work.
> > > At a technical level, we note that many theoretical papers rely on manipulations of Hermite expansions, which is not a mathematical novelty per se, but does not prevent them from making conceptual steps, see e.g. Dandi et al. (ICML 2024), Damian et al. (NeurIPS 2023), Bardone & Goldt (ICML 2024). We introduce the Fourier data model, which allows us to discover a new phenomenon, namely the speed-up in learning the Fourier phases of the inputs, in presence of a power-law decay in the spectrum. This prediction is only possible because of the non-Gaussianity of the input model! Importantly, we are able to verify this effect experimentally both on synthetic data (Figures 2 and 4) and real data (textures, ImageNet, CIFAR-100) with deep convolutional networks like Resnet18 and two-layer classifiers. Such robust empirical validation is often lacking in theoretical works and has been noted as a limitation of the existing literature.
> > >
> > > At a more detailed level, we note that non-Gaussianity and anisotropy of the inputs require computing their fourth-order statistics already at the population level (Lemma C.6), the coefficients of the likelihood ratio (Lemma D.1) as well as isolating the contribution of an extensive (with the dimension) number of spikes in the expansion of the population loss (cf. Proposition 5.1). We cannot e.g. have a likelihood ratio depending on the spikes, as done instead for the rank-one spiked covariance of Bardone & Goldt (2024) which is, to the best of our knowledge, the closest approach to ours. Moreover, we thank the reviewer again for having pointed out that online SGD often fails in presence of anisotropic covariances, cf. Moussav-Hosseini et al. (2023). For us, instead, online SGD does not fail and strongly recovers the signal (experiments in Figure 4) by exhibiting a rich dynamics: it first weakly recovers and then forgets the other principal components. In prior work, e.g., Ben Arous et al. (2024), a similar behavior emerges only under fine tuned signal-to-noise separation conditions, whereas it is induced by the scalings of the amplitudes in the Fourier domain.
> > >
> > > > Summary
> > >
> > > Since this is the last reply permitted by the rules for the author-reviewer discussion, we  hope that we were able to address your remaining concerns. If so, we would appreciate it if you could revise your score; in any case, we thank you for engaging with our work.

---

### Official Review · Reviewer_7fEi · 2026-03-06

**Soundness:** 3
**Presentation:** 4
**Significance:** 4
**Originality:** 4
**Overall Recommendation:** 5
**Confidence:** 4

**Summary:**

This paper investigates the "simplicity bias" in neural networks through a Fourier-based analytical framework, demonstrating that models trained with gradient-based methods sequentially exploit lower-order statistics (amplitudes) before transitioning to higher-order structural features (phases). By introducing a tractable synthetic data model for translation-invariant inputs, the authors rigorously establish that learning exclusively from phase information is a computationally hard task for online stochastic gradient descent, requiring cubic sample complexity ($n \gg N^3$) in high-dimensional isotropic settings.

Crucially, the study proves that the characteristic power-law decay of Fourier amplitudes found in natural images provides a mechanistic acceleration, reducing the required sample complexity to a quasi-linear regime even when amplitudes themselves are non-discriminative. These theoretical insights are corroborated by extensive simulations on texture datasets and ImageNet with deep convolutional architectures, providing a principled explanation for how neural networks efficiently navigate the statistical complexities of natural image distributions.

**Compliance With Llm Reviewing Policy:**

Affirmed.

**Final Justification:**

I believe the authors have adequately addressed my concerns in the rebuttal, and the overall quality of the paper is solid. Accordingly, I have decided to increase my score.

**Key Questions For Authors:**

1. The paper’s most significant theoretical claim—that power-law decaying amplitudes reduce the information exponent from k*=4 to k*=2 is currently presented as Conjectures 5.2 and 5.3. While the population drift analysis is compelling, it remains a first-order approximation that omits diffusive noise. Can the authors provide a more rigorous sketch or proof-outline showing that the diffusive terms do not dominate the dynamics in the high-dimensional limit ($N \rightarrow \infty$)? A formal verification that the "acceleration" is robust to the stochasticity of SGD would justify a higher Soundness score.

2. The empirical validation on ImageNet (Figure 3e) utilizes a "super-classed" subset of only 10 classes. However, the full 1000-class distribution involves a significantly more fragmented and lower-rank principal subspace for any given class pair. Does the observed mechanistic acceleration persist when the task-relevant modes are competing with a much larger set of non-relevant but extensive eigenvalues? Confirmation that the power-law benefit scales beyond simplified subsets would greatly enhance the Significance of the work.

3. The experiments show that networks learn amplitudes before phases. However, it remains unclear if this "amplitude-first" bias is strictly a property of the 1/f power spectrum or partially a byproduct of the architecture's initialization (e.g., the spectral bias toward low-frequency components). If the same experiment (Figure 1g) were performed on a network initialized with high-frequency "edge-detecting" kernels, would the sequential learning order be preserved? Clarifying the relative weight of data statistics vs. architectural bias would refine the Mechanistic Insight contribution.

4. The introduction of the roots-of-unity corrector $U^{\mu}$ is a clever technical innovation to ensure translation-invariance while enabling non-Gaussian statistics. Are the fourth-order cumulants and the resulting $k^*$=4 complexity specific to sampling $U^{\mu}$ from the fourth roots of unity? If $U^{\mu}$ were sampled from a continuous uniform distribution or a different discrete group, would the information exponent—and thus the learning hardness—change? This would clarify the scope of the Fourier Data Model as a general theoretical tool.

5.  Figure 2 (Right) illustrates a long "search phase" around the origin before the weights align with the DFT phase vectors in the isotropic cubic regime. In the power-law (non-isotropic) regime, does the acceleration purely increase the drift magnitude, or does it fundamentally eliminate the search phase by "tilting" the initial landscape? Understanding whether the power-law spectrum changes the computational nature of the optimization problem (from search to descent) would strengthen the paper's Originality.

**Limitations:**

The authors have not adequately discussed the technical limitations or potential negative societal impacts of their work, as the current impact statement is overly brief and lacks specific highlighting of consequences.

Constructive Suggestions for Improvement:

1. The authors should acknowledge that the transition from a cubic information exponent (k*=4) to a quasi-linear one (k*=2) in non-isotropic settings currently rests on conjectures (5.2 and 5.3). A discussion on the necessity of computing diffusive terms to fully formalize the stochastic stability of this acceleration would provide a more honest technical baseline.
2. The limitation of using a "super-classed" 10-class version of ImageNet should be addressed. The authors should discuss whether the mechanistic acceleration remains dominant when the principal subspace is more fragmented, as seen in the full 1000-class distribution.
3. The authors should discuss the extent to which the observed "amplitude-first" simplicity bias is contingent on the specific initialization or inductive biases of the chosen architectures.
4. While this is a theoretical work, the authors should briefly address the risks associated with simplicity bias in real-world deployments. For example, if a model relies primarily on pairwise correlations (amplitudes) during early training, it may fail to generalize to out-of-distribution samples where phase-encoded structural information (edges/shapes) is critical for safety.

**Strengths And Weaknesses:**

#### **1. Soundness**

The technical execution is generally rigorous, particularly in the formal derivation of the $O(N^3)$ sample complexity lower bound for phase recovery in isotropic high-dimensional settings. The use of the information exponent framework to characterize the "hardness" of learning from phases alone is theoretically sound and well-aligned with established literature in high-dimensional inference. The empirical validation, utilizing phase-swapped test sets and texture datasets, provides compelling evidence for the sequential learning of amplitudes followed by phases.

However, a significant portion of the non-isotropic analysis—arguably the paper's most impactful contribution—remains grounded in Conjectures 5.2 and 5.3. While the population drift analysis is persuasive, the absence of a rigorous treatment for diffusive terms and the resulting Ornstein-Uhlenbeck processes leaves a gap in the formal proof of acceleration. Additionally, the ImageNet experiments in Figure 3e are conducted on a "super-classed" version with only 10 classes, leaving open the question of whether this mechanistic acceleration holds at the same scale in the full 1000-class distribution.

#### **2. Presentation**

The submission is exceptionally clear and successfully bridges disparate fields, linking cognitive science observations regarding human perception to high-dimensional statistical physics. The narrative flow—from experimental discovery to synthetic modeling and theoretical bound derivation—is logical and easy to follow. Visualizations, particularly Figure 2’s illustration of cubic scaling and Figure 1’s perceptual swapping, effectively ground the abstract theory in observable phenomena.

A minor presentation concern involves Section 2.2, where the notation for the roots-of-unity corrector $U^\mu$ is quite dense. While the mathematical necessity of $U^\mu$ to maintain translation-invariance is clear, a more intuitive prose explanation of its functional role in the Fourier data model would enhance accessibility for a broader ML audience.

#### **3. Significance**

This work addresses a foundational question in deep learning: why neural networks can learn complex structural features (like edges) from natural data despite the theoretical "curse" of sample complexity in high dimensions. By identifying the **power-law decay (1/f) of Fourier amplitudes** as a functional accelerator for higher-order phase recovery, the authors provide a principled mechanistic explanation for the efficiency of CNNs and other architectures on natural image distributions. This insight could significantly influence future research into data-dependent learning dynamics and architectural initialization.

#### **4. Originality**

The framing of simplicity bias through the specific dichotomy of Fourier amplitudes vs. phases is highly original. It offers a much cleaner statistical separation of pairwise and higher-order correlations than previous frequency-based spectral bias studies. Technically, the introduction of a synthetic model that surgically manipulates phases while preserving circulant covariance structures—via theRoots of Unity corrector—is a creative and novel addition to the theoretical toolkit of high-dimensional inference.

---

> ### Author Rebuttal · Authors · 2026-03-31
>
> We thank the reviewer for their careful review of our paper, and we appreciate that they found the work to be "exceptionally clear" "technically rigorous," and "highly original".
>
> > On proving conjectures 5.2 and 5.3, and on diffusive noise
>
> Thanks for the question. We can state the conjectures in Section 5.2 rigorously at the level of the population dynamics (cf. Lemma F.5 and F.7). From there, one can observe that the blow-up for the summary statistics (10) happens at different time scales. This population dynamics reflects the information exponents computed in Section 5.1, based on the expansion of the loss in Proposition 5.1.
> We decided to take a step further towards rigorous claims by looking at the effective dynamics (see Theorem F.3 in our work, originally from Ben Arous et al. ('22)). Among the three ingredients needed by this theorem, namely the population drift, the population corrector and the effective volatility, we cannot estimate the effective volatility. Indeed, to the best of our knowledge, it has been only computed in simpler data models, like Gaussian additive models e.g. single-index models or Gaussian mixtures (cf. Ben Arous et al. '22, '24, '25), while we have built a genuinely non-Gaussian data model. Nevertheless, we can provide some empirical validation for the scalings we predict (Figure 3).
>
> > The limitation of using a "super-classed" 10-class version of ImageNet should be addressed.
>
> Thanks, this is a good point, for which we can provide some additional numerical evidence. We haven't performed the experiments on ImageNet for a large number of classes due to restrictions of time and computational resources. However, we see the speed-up effect due to the shared principal subspace (Figure 3) also on the full CIFAR-100 data set  (https://figshare.com/s/0fd6b37b82fd6b0bc21e) with both SGD and Adam optimisers.
>
> > Spectral bias
>
> We thank the reviewer for pointing out that the initialisation may have a role in the "amplitudes first, then phases" phenomenon. Nevertheless, our theoretical results establish that learning is strongly driven by the structure of the data, due to the information encoded in higher-order statistics of the inputs (i.e. in the Fourier phases) that is relevant for the given task. In particular, our results in Proposition 5.1, Conjecture 5.2  and 5.3 show that the speed-up in learning the phases (in the anisotropic case) is present even when the leading principal subspace does _not_ correspond to low-order frequencies. This distinguishes our contribution from previous works like Rahaman et al. '19. If we repeated the experiment in Figure 1g) with a network initialized using high-frequency, edge-detecting kernels, the network would be already close to convergence, and the sequential learning effect would disappear.
>
>
> > Roots-of-unity corrector
>
> We thank the reviewer for this question regarding the generality of the roots-of-unity corrector. This corrector allows us  manipulate the higher-order cumulants of the data while preserving the translation-invariance of the inputs in our Fourier model. Alternative constructions are possible as long as translation-invariance is preserved. In such cases, the same analyses, both rigorous and conjectural, can be done.
> For example, one can sample $U$ from the third roots of unity, leading to non-trivial third-order cumulants and a lower ($k^* = 3$) information exponent (corresponding to a quadratic weak recovery of the phases in the isotropic case).
>
> > "Tilting" the original landscape
>
> This is a very good point. The non-isotropic (power-law) regime indeed fundamentally changes the optimization landscape, which is precisely why the search phase is significantly reduced. We show in (https://figshare.com/s/8ce3500c2028efdc2c47) what the effect of the power-low decay on the level sets of the empirical loss function is. In the isotropic case, we see a large flat region around the origin which essentially extend the serach phase.
>
>
> > While this is a theoretical work, the authors should briefly address the risks associated with simplicity bias in real-world deployments.
>
> Thanks, we will address these issues.
>
> > Summary
>
> We hope our replies have addressed your concerns. If so, we would appreciate it if you could revise your score; if not, we look forward to addressing any remaining issue during the discussion.

---

> > ### Author Rebuttal · Reviewer_7fEi · 2026-04-02
> >
> > I think this work is interesting. However, I have the following concerns:
> >
> > **(1)** The theoretical analysis is primarily developed under linear or shallow model assumptions. It remains unclear to what extent the conclusions extend to modern deep neural networks with strong nonlinearity and expressive capacity. A more explicit discussion on the scope and applicability of the theory would strengthen the paper.
> >
> > **(2)** The paper frequently equates stronger signal (i.e., larger loss along certain directions) with faster learning by SGD. This implication is not fully justified. In practice, SGD dynamics depend not only on the structure of the loss landscape, but also on initialization, stochasticity, and optimization trajectories. As such, the current analysis appears to conflate statistical properties of the population loss with optimization behavior. Clarifying this distinction and moderating the corresponding claims would improve the rigor of the work.
> >
> > **(3)** More broadly, the paper emphasizes the role of data structure (e.g., spectral properties) in determining learning difficulty. While this perspective is valuable, it is generally understood that learning performance arises from the interplay between data distribution, model expressivity, and optimization. By fixing the latter two, the paper isolates the effect of data, but some of its conclusions appear stronger than what is justified by the analysis.
> >
> > **(4)** In summary, I encourage the authors to better delineate the scope of their theoretical results and to more carefully discuss the underlying assumptions and limitations, which would enhance the clarity and credibility of the paper.

---

> > > ### Author Response · Authors · 2026-04-06
> > >
> > > We thank the reviewer for these additional questions.
> > >
> > > > (1) Implications of our analysis of shallow models for deep neural networks
> > >
> > > Yes, we focus on shallow models for our theoretical analysis; however, this is a pretty standard setting for many theory papers at ICML and NeurIPS, including the recent outstanding-paper award winners at NeurIPS 2022 and 2025 by Ben Arous et al. and Bonnaire et al., respectivley. Within the theoretical literature on shallow models, our work makes a crucial step since we do _not_ assume that inputs are Gaussian; instead, our Fourier model reflects the basic fact that real data is non-Gaussian by manipulating the phases. This feature of the model introduces introduces additional non-linearities in the analysis (cf. Lemma C.7). We provide clear evidence that the concrete predictions of our theoretical analysis, like the speed-up of learning due to the power-law spectrum of images, are verified in deep convolutional neural networks; in particular, we find the speed-up in a Resnet18 trained on multiple datasets like textures, ImageNet and CIFAR100 (https://figshare.com/s/0fd6b37b82fd6b0bc21e), using multiple optimisers (mini-batch SGD and Adam), and multiple losses (mean-squared and cross entropy).
> > >
> > > > (2) The paper frequently equates stronger signal (i.e., larger loss along certain directions) with faster learning by SGD. This implication is not fully justified. In practice, SGD dynamics depend not only on the structure of the loss landscape, but also on initialization, stochasticity, and optimization trajectories. As such, the current analysis appears to conflate statistical properties of the population loss with optimization behavior. Clarifying this distinction and moderating the corresponding claims would improve the rigor of the work.
> > >
> > > We agree with the reviewer that an analysis of the landscape by itself is insufficient to establish the difficulty of learning, which is why our results are for the _dynamics_ of learning (Theorem 3.1, Proposition 5.1 and Conjectures 5.2 and 5.3) and do not rely directly or only on the properties of the landscape. For example, we prove rigorously in Theorem 3.1 that online SGD, randomly initialised in high dimensions, gets stuck in a long search phase (requiring more that $O(N^3)$ steps), which is prohibitive in large dimensions for practical applications. As an empirical observation, we note in Figure 2 (right) that the loss landscape has a flat region around the origin. We contrast this setting with the power-law decay case: at the population level, we prove (cf. Lemma F.7) that there is a faster recovery of the signal.
> > > Then, we empirically observe that the loss landscape appears more tilted along informative directions (this aligns with the reviewer’s previous remark on the modification of the landscape, for which we have provided the numerics https://figshare.com/s/8ce3500c2028efdc2c47). We emphasise, however, that the dynamics is justified via Lemma F.7, not based on the shape of the loss landscape, which is a consequence.
> > >
> > > > (3) More broadly, the paper emphasizes the role of data structure (e.g., spectral properties) in determining learning difficulty. While this perspective is valuable, it is generally understood that learning performance arises from the interplay between data distribution, model expressivity, and optimization. By fixing the latter two, the paper isolates the effect of data, but some of its conclusions appear stronger than what is justified by the analysis.
> > >
> > > We agree that the performance of neural networks is the result of the interplay between data structure, network architecture and optimisation. While our focus is on the data structure, our results come about by taking the architecture of the network and the optimisation dynamics directly into account: indeed, our results describe the learning _dynamics_ of the neural network. While different choices of architectures would change the precise form of our results, our analysis helps isolate the effect of the data distribution in a controlled setting. We then establish that the captured effect is relevant in a variety of scenarios; for example, we show that the speed-up predicted theoretically in the power-law regime is consistent across different datasets (textures, ImageNet, CIFAR-100 (cf. Figure 3 and https://figshare.com/s/0fd6b37b82fd6b0bc21e)), losses (mean-squared and cross entropy), optimisers (mini-batches SGD and Adam) and architectures (two-layer classifier and ResNet18).
> > >
> > > > Summary
> > >
> > > Since this is the last reply permitted by the rules for the author-reviewer discussion, we  hope that we were able to address your remaining concerns. If so, we would appreciate it if you could revise your score; in any case, we thank you for engaging with our work.

---

### Official Review · Reviewer_8FNT · 2026-03-12

**Soundness:** 3
**Presentation:** 3
**Significance:** 3
**Originality:** 2
**Overall Recommendation:** 5
**Confidence:** 4

**Summary:**

The manuscript studies the sample complexity and dynamics of retrieving phase versus amplitude information in translationally invariant datasets. They propose a toy data model, where the difference between classes resides in phase information which also accommodates a power law distribution of Fourier amplitudes. They prove using, information exponent approaches, that learning this phase information in the uniform amplitude case has cubic sample complexity. They then analyze theoretically the case where the spectrum is dominated by O(1) strong modes and the mode with relevant information is sub-extensive (and the rest are small). By analyzing the information exponent and the dynamics around initialization, they show that due to high-order correlations between the relevant mode and the large modes, one gets a form of assisted learning. Specifically, the network first learns to emphasize the subspace of the O(1) strong modes, then learns the phase information, and then these dominant but irrelevant modes decay. They provide numerical evidence showing such mechanism is at play on datasets with power law spectrum. Specifically, for real datasets (Fig. 3) they washout the amplitude information between the two classes, but show that the existence of non-informative power law amplitudes speeds up learning compared to a flattened dataset. For the toy dataset (Fig. 4.) they show that principle components along the dominant directions behave as expected.

**Compliance With Llm Reviewing Policy:**

Affirmed.

**Key Questions For Authors:**

None.

**Limitations:**

See weaknesses

**Strengths And Weaknesses:**

Strengths:

This works addresses interesting and subtle aspects about what makes image data easily learnable given that, in standard images, most information lays in the harder-to-extract phase information. They provide a detailed and non-trivial mechanism for how irrelevant but dominant modes facilitate learning, which is an extension of that of Ben Arous (2025), to richer datasets which combine power law spectrum and non-Gaussianities.

Weaknesses:

The weaknesses I found were mainly at the presentation level.

1. Around Eq. (8,9) the scaling seems to rely on having O(1) dominant (\lambda_m = O(N)) modes, however that’s not apparent in the text at this point.
2. In a related manner, in the paragraph containing Eq. (9), the sentence gets cutoff before they detail the assumptions on \lambda_m
3. By casually reading their section 5.1., which doesn’t say much about dynamics, one can be left with the notion that the entirely story is about powerlaws yielding an O(1) effective dimension and hence no show big separation between linear and cubic sample complexity. Though this is discussed a bit in words in the next section, and in the appendix — it is not so well separated from that simpler based-line case of a reduced effective dimension. The authors can potentially make this point stand out more clearly, earlier on, but explaining that it is the c^{\sigma}_6 coupling that allows this, and not the effective dimension alone.
4. There is a gap which is good to clarify, between the stated architecture having two trainable layers, and the analytics being carried with one trainable layer.

---

> ### Author Rebuttal · Authors · 2026-03-31
>
> We thank the reviewer for their careful reading of our paper.
>
> > Around Eq. (8,9) the scaling seems to rely on having $O(1)$ dominant (\lambda_m = O(N)) modes, however that’s not apparent in the text at this point.
>
> We thank the reviewer for this comment, which allows us to clarify the generality of our setting in Section 5.1. Indeed, we perform the expansion of the population loss in Proposition 5.1 for a general circulant covariance matrix with an extensive (scaling with the dimension) number of non-trivial eigenvalues, corresponding to an extensive number of DFT eigenvectors. With this expansion, we treat full-rank anisotropy in the context of non-Gaussian inputs.
> However, to identify the information exponents in the paragrahs "near-isotropic inputs" and "power-law decaying inputs", and in Section 5.2, we specialise to the case of having $O(1)$ dominant modes in order to keep the analysis simpler, since it is already quite involved. This more restrictive setting is motivated by the effective dimension of the covariance matrix of natural images (see discussion below and with reviewer zTig).
>
>
> > In a related manner, in the paragraph containing Eq. (9), the sentence gets cutoff before they detail the assumptions on \lambda_m
>
> Thanks, we will complete it in the revision, we ask for $\lambda_m \approx N$. We will expand on the fact that, for spectra exhibiting a power-law-decay (defined as $\lambda_m \propto m^{-\alpha}$), we can consider an effective low-rank eigenstructure: a finite number $O(1)$ of modes having an extensive $\lambda_m = O(N)$, whereas the tail of the spectrum stays $O(1)$. This is appropriate in view of different measures of dimensionality, like the effective dimension or the Inverse Partecipation Ratio (IPR), of the eigenvalues concentrating on an $O(1)$ quantity for power-law decays with exponent $\alpha>1$ (see Cheng & Montanari, Annals of Statistics ('24) or Wortsman & Loureiro, arXiv:2510.04780). This happens to be the case of natural images (see e.g. Hyvärinen et al. (2009)).
>
>
> > By casually reading their section 5.1., which doesn’t say much about dynamics, one can be left with the notion that the entirely story is about powerlaws yielding an $O(1)$ effective dimension and hence no show big separation between linear and cubic sample complexity.
>
> This is a very good observation. Indeed, the effective low-dimensional eigenstructure does not reduce the dynamics to a finite exploration, as it is often the result of commonly used in practice preprocessing precedure (like PCA). We will emphasise in the revision that the effect of the presence of the power-law decay in the spectrum is to induce and extensive (with the dimension) signal-to-noise ratio in the loss (cf. Proposition 5.1) which soemhow "couples" the signal and the quadratic form appearing due to the covariance. Also, in our setting the principal subspace are shared between classes, so they do not help with classification by themselves.
>
> > There is a gap which is good to clarify, between the stated architecture having two trainable layers, and the analytics being carried with one trainable layer.
>
> Thanks, we will discuss this in a section dedicated to some limitations of this work. Note however that our theory is able to capture some rebust behaviours accross various architectures (two-layer classifiers, convolutional deep neworks), datasets (textures, ImageNet, CIFAR-100) and training algorithms (mini-batch SGD, Adam). Cf. Fig 3 and the new figure at (https://figshare.com/s/0fd6b37b82fd6b0bc21e), for which we use a cross-entropy loss.

---

> > ### Author Rebuttal · Reviewer_8FNT · 2026-04-02
> >
> > The authors have addressed all my concerns satisfactorily. I think the results are novel and solid theoretically. My score remains 5.

---

### Official Review · Reviewer_E5kn · 2026-03-12

**Soundness:** 3
**Presentation:** 2
**Significance:** 3
**Originality:** 3
**Overall Recommendation:** 5
**Confidence:** 3

**Summary:**

The authors study the impact of Fourier properties on the sample efficiency of a neural network. They start by showing empirically that neural networks learns amplitude information before phase information. After that they derive analytical results with a perceptron and a data model they introduce. They two main theoretical results are that with isotropic data more than N^3 samples are necessary to recover the discriminative component whereas in some non-isotropic scenario a quasi-linear complexity can be obtained.

**Compliance With Llm Reviewing Policy:**

Affirmed.

**Final Justification:**

The authors provided a clear rebuttal.

**Key Questions For Authors:**

One of the key results of the paper is the impact of non-isotropy on sample efficiency. However I observed that Th. 3.1 relies on non penalized loss whereas Conjectures 5.2 and 5.3 relies on penalized loss. What is impact of penalized loss on your results and analysis? You say for near-isotropic inputs that "We therefore expect SGD to require cubic sample complexity to recover the signal." (line 382), do we expect this for penalized and non-penalized loss? What do you mean by we "expect"? Is it an intuition or can we prove it? (Conjecture 5.2 only mention that sample complexity is not quasi-linear, no mention of cubic complexity)
- Line 401, should "finite number of large eigenvalues" be understood as M=O(1) (independent of N)
- In Conjectures 5.2 and 5.3, what do you mean by "recover"? In Theorem 3.1 only weak recovery is defined
- Some typos: line 123 (right), line 373(right): if lambda_m what?, line 421 (right)
- The statemnt of the paragrpah "Power-law decaying inputs" saying that lambda_k0 is one of the top eigenvalues but not the leading one was confusing during my first read. The thing is that in section 5.1 lambda_m is defined as all eigenvalues excpept the k0 one whereas in section 5.2 lambda_m is define as a few large eigenvalues. It seems that the paragrpah "Power-law decaying inputs" should be in section 5.2

**Limitations:**

There is no clear mention of the limitations of the results. While theoretical results even in simplified setups can be valuable to aim towards a better understanding of how neural networks learn in practical setups, it is important to assess the limitations of the results. The first key question can be seen as an example of limitation not mentioned in the paper.

**Strengths And Weaknesses:**

Strengths
- The authors give valuable insights about the role that the "power law" property of natural images might have on learning efficiency. It's valuable to know what properties of "real" data makes learning efficient.
- The analysis is rigorous and relies on methods developed by other recent work (Ben Arous et al)

Weaknesses
- The experimental argument that "is demonstrating that neural networks sequentially learn to exploit first amplitude, then phase information." is weak and the connection with the theoretical results is not clear. The theorems and conjecture from the theoretical part do not give insights on the learning dynamics (directly) but on sample complexity.
- Regarding the presentation, there are a few typos and parts that might be reordered or reformulated (see Key questions)
- minor weakness:it seems the paper fails to discuss earlier work on implicit bias/learning dynamics in CNNs, which is relevant because the Fourier perspective is used there as well. See, a.o.,

------ Gunasekar, Suriya, et al. "Implicit bias of gradient descent on linear convolutional networks." Advances in neural information processing systems 31 (2018).

---- Pinson, Hannah, Joeri Lenaerts, and Vincent Ginis. "Linear cnns discover the statistical structure of the dataset using only the most dominant frequencies." International Conference on Machine Learning. PMLR, 2023.

---

> ### Author Rebuttal · Authors · 2026-03-31
>
> We thank the reviewer for their careful reading of the manuscript.
>
> > Relation between theoretical results and experiments
>
> We thank the reviewer for this question. Our theory and experiments are closely related. We theoretically describe the learning dynamics of "online" SGD, where a new sample is used to compute the gradient at each step of SGD. Therefore, the sample complexity equals the number of steps. This is a pretty standard setup for theoretical analyses (cf. the paper by Ben Arous et al. which won the best-paper award at NeurIPS in 2022) and our predictions crucially match our simulations on three different data sets (textures, ImageNet, and CIFAR100, see the new experiment (https://figshare.com/s/0fd6b37b82fd6b0bc21e)) with both fully-connected and convolutional architectures (see Figure 1 & 3).
>
> > Further related works
>
> Thank you for highlighting these papers. We will cite them and comment on the connection with our Fourier perspective. Note that linear networks cannot capture higher-order correlations of the inputs, so they are not able to learn the phase information that is the focus of this paper.
>
> > Penalised vs. non-penalised loss and cubic sample complexity for near-isotropic inputs
>
> This is a very good point also addressed by reviewer zTig. We use a penalized loss in Section 5.2 to match the conventions of Ben Arous et al. (2022, 2025). The existence of the penalty does not have an impact on the predictions for the time scales over which the phases are learnt. Those are identical for both penalized and non-penalized losses, and depend only on the input statistics (i.e. whether the spectrum decays is near-isotropic or decays with a power-law). In the near-isotropic case, we predict SGD to require a cubic sample complexity to (weakly) recover the signal for both losses, while a power-law spectrum accelerates recovery as formalised in Conjecture 5.3. This result is not just an intuition; it is based on the information exponent of the loss that we compute after eq. 9. We can also provide a heuristic derivation following Damian et al. (NeurIPS '23) and Ricci et al. (ICML '25): the evolution of, e.g., $\alpha_v$, is given by
> $\dot\alpha_v = \dfrac{\mathbb{E}[g \cdot v]^2}{\alpha_v\mathbb{E}[\|g\|^2]},$
> where $g := \nabla L(w; x,y)$.
> Since $g$ is a random vector in $\mathbb{R}^N$ where each coordinate is O(1), we have $\mathbb{E}[\|g\|^2] \approx N$. For both losses, following the rigorous calculations in Lemma F.5, we get $\mathbb{E}[g \cdot v] \approx \alpha_v^3$. Then, $\dot\alpha_v \approx \alpha_v^5/N$ which, initialised at $\alpha_v(0)= 1/\sqrt{N}$, implies that $\alpha_v = O(1)$ in approximately $N^3$ steps.
>
>
> > Finite number of eigenvalues as M=O(1)
>
> Yes, we will specify M = O(1). This choice is a consequence of the power-law decay of the eigenvalues of the covariance; if the spectrum decays as $\lambda_m \propto m^{-\alpha}$ for $\alpha > 1$, the intrinsic dimension of the covariance matrix is constant with respect to the input dimension, which is consistent with an O(1) number of large eigenvalues (see for example Wortsman and Loureiro (2025) arXiv:2510.04780, and our reply to zTig).
>
> > In Conjectures 5.2 and 5.3, what do you mean by "recover"?
>
> We refer to weak recovery, we will specify it.
>
> > Typos
>
> Thanks, we will fix them. We want $\lambda_m \approx N$.
>
> > $\lambda_{k_0}$ top eigenvalue but not leading
>
> You are correct: $\lambda_{k_0}$ is among the large eigenvalues, but possibly not the largest one. This is a key property of our Fourier data model because it means that simple spectral methods like PCA cannot identify the signal reliably, which is a limitation of some toy models in the literature.
>
> > $\lambda_m$ definitions
>
> We slightly overloaded the notation $\lambda_m$, thank you for pointing out that this can be confusing. We generically denote the $m$-th eigenvalue of the empirical covariance matrix as $\lambda_m$; in Section 5.1, where we give a general expression for the population loss, all of the eigenvalues appear. From there, we can treat specific cases like spectra with a finite number of non-trivial eigenvalues, or spectra that decay like a power-law, $\lambda_m \propto m^{-\alpha}$, where again only the leading eigenvalues are relevant for the dynamics, see our comment above. This is why it might appear that the definition of $\lambda_m$ changed. We will clarify this in the revised manuscript.
>
> > Towards a dedicated limitiation section
>
> At the moment, we discuss some limitations, e.g. the parity of the activation function as a technical assumption (line 233). We will collect them, and add an extended discussion of what one would need to turn the conjectures into theorems by computing the effective volatilities (cf. Theorem F.3), in a dedicated section in the revision.
>
>
> We hope our replies have addressed your concerns. If so, we would appreciate it if you could revise your score; if not, we look forward to addressing any remaining issue during the discussion.

---

> > ### Author Rebuttal · Reviewer_E5kn · 2026-04-03
> >
> > I would like to thank the reviewers for their clarification, and I will raise my score.

---

### Decision · Program_Chairs · 2026-04-30

**Decision:**

Accept (regular)

**Comment:**

The authors leverage the Fourier transform to understand the learning dynamics of neural networks.

The paper provides a rigorous derivation and a mechanistic interpretation. The results are recognized by reviewers to be novel and solid theoretically. The rebuttal period was helpful to resolve most of the reviewer concerns. The response to the most skeptical reviewer about novelty is appropriate and this alone cannot justify rejection.

Camera-ready version:
- The novelty of this paper must be clarified against existing simplicity bias and Gaussian-analysis papers.
- The presentation of the paper must be improved (assumptions, proof roadmap)
- Clarify what is proved and what is a mechanistic interpretation.